# Dimension-free Score Matching and Time Bootstrapping for Diffusion Models

**Syamantak Kumar**[*]
University of Texas at Austin
syamantak@utexas.edu

**Dheeraj Nagaraj**
Google DeepMind
dheerajnagaraj@google.com

**Purnamrita Sarkar**
University of Texas at Austin
purna.sarkar@utexas.edu

## Abstract

Diffusion models generate samples by estimating the score function of the target distribution at various noise levels. The model is trained using samples drawn from the target distribution, progressively adding noise. Previous sample complexity bounds have a polynomial dependence on the dimension $d$, apart from $\log(|\mathcal{H}|)$, where $\mathcal{H}$ is the hypothesis class. In this work, we establish the first (nearly) dimension-free sample complexity bounds, modulo any dependence due to $\log(|\mathcal{H}|)$, for learning these score functions, achieving a double exponential improvement in dimension over prior results. A key aspect of our analysis is to use a single function approximator to jointly estimate scores across noise levels, a critical feature in practice which enables generalization across timesteps. We introduce a novel martingale-based error decomposition and sharp variance bounds, enabling efficient learning from dependent data generated by Markov processes, which may be of independent interest. Building on these insights, we propose Bootstrapped Score Matching (BSM), a variance reduction technique that utilizes previously learned scores to improve accuracy at higher noise levels. These results provide crucial insights into the efficiency and effectiveness of diffusion models for generative modeling.

## 1 Introduction

Score-based diffusion models [43, 17] are generative models that have transformed image and video generation [38, 40, 37, 36], enabling foundation models to produce photorealistic and stylized images from text prompts. Their adaptability extends diverse domains such as audio [24, 10], text [13, 15, 29, 49], molecule [18, 19], and layout generation [21, 28]. They generate additional samples given $m$ i.i.d. samples from a target distribution ($\pi$) using a trained neural network that learns the score function $\pi$ at different noise levels. In contrast, the classical Markov Chain Monte Carlo (MCMC) methods which seek to sample from a distribution given access to underlying density function. While MCMC methods can be slow for multi-modal distributions, diffusion models can sample efficiently with minimal assumptions, provided the score functions are learned accurately [6, 1].

Given $m$ i.i.d. samples from the target distribution, the *first step* (called the forward process) obtains noised samples from a noising Markov process converging to the Gaussian distribution at various noise levels. The *second step* estimates score functions of the distribution at each noise level using Denoising Score Matching (DSM) [51]. This approach relies on learning from *dependent data* from

---

[*]Work partly done during internship at Google DeepMind

39th Conference on Neural Information Processing Systems (NeurIPS 2025).

multiple trajectories of a Markov process in contrast to learning with i.i.d. data in traditional settings. Prior works [3, 14] provide theoretical guarantees for score function approximation separately at each noise level using the same samples. However, in practice, a *single function approximator* is commonly used at all noise levels, which is considered by [16]. [4] show that despite the problem of distribution estimation suffering from the curse of dimensionality [5, 35], the existence of low-dimensional structures allows neural networks to learn the score functions. All of these existing bounds exhibit polynomial dependence on the dimension, $d$. We establish that under *suitable smoothness conditions* for a given function class, score matching with a single function approximator jointly across all timesteps achieves a nearly **dimension-free sample complexity** that depends on the smoothness parameter and grows only as $\log \log(d)$.

## 1.1 Our Contributions

**(1)** We analyze the sample complexity of *denoising score matching* across noise levels using a single function approximator, achieving a **double-exponential reduction in dimension dependence**.

**(2)** We present a **novel martingale decomposition** of the error, which allows us to bound the error despite being composed of samples from multiple trajectories of *dependent data*.

**(3)** We use second-order Tweedie-type formulae to obtain a **sharp bound on the error variance**, crucial for establishing almost **dimension-free** convergence rates.

**(4)** Inspired by the above, we present **Bootstrapped Score Matching**. Here the score at a given noise level is learned by bootstrapping to the learned score function at a lower noise level, achieving variance reduction. This shows improved performance compared to DSM in simple empirical studies.

## 1.2 Related work

**Score matching and diffusion models:** Score Matching was introduced in the context of statistical estimation in [20] with an algorithm now called Implicit Score Matching (ISM). Diffusion models are trained using Denoising Score Matching (DSM) introduced in [51], and is based on Tweedie's formula. Several algorithms have been introduced since, such as Sliced Score Matching [44] and Target Score Matching [7]. Prior works have analyzed the complexity of Denoising Score Matching in various settings [5, 35, 14, 3, 12]. We consider the setting in [14, 3], where the score functions of the given distribution can be accurately approximated by a function approximator class (ex: neural networks), instead of the worst case non-parametric analysis in [12]. These bounds can then be used with the discretization analyses in [1, 6, 27] to guarantee the quality of the generated samples.

**Learning from dependent data:** Learning with data from a markov trajectory has been explored in literature in the context of system identification, time series forecasting and reinforcement learning [8, 42, 34, 25, 48, 53, 2, 26, 46] Many of these works analyze the rates of convergence with data derived from a mixing Markov chain, when the number of data points available is much higher than the mixing time, $\tau_{\mathsf{mix}}$. In our context, the Markov chain contains $\tilde{O}(\tau_{\mathsf{mix}})$ data points created by progressively noising samples from the target distributions, where $\tilde{O}$ hides logarithmic factors. This is similar to the setting in [48], which considered linear regression and linear system identification.

# 2 Problem setup and preliminaries

**Notation:** We use $[n]$ to denote $\{i \in \mathbb{N} \mid i \leq n\}$. $\mathbf{I} \in \mathbb{R}^{d \times d}$ represents the $d$-dimensional identity matrix. We use $\mathcal{N}(\mu, \boldsymbol{\Sigma})$ to denote the multivariate normal distribution with specified mean, $\mu$ and covariance matrix $\boldsymbol{\Sigma}$. $\|.\|_2$ denotes the $\ell_2$ euclidean norm for vectors and $\|.\|_{\mathsf{op}}$ denotes the operator norm for matrices. $\mathbb{E}[X]$ denotes the expectation of the random variable $X$ and $\mathsf{Cov}(X)$ denotes its covariance matrix. For $a, b \in \mathbb{R}$, we write $a \lesssim b$ if and only if there exists an absolute constant $C > 0$ such that $a \leq Cb$. $\tilde{O}, \tilde{\Omega}$ represent order notations with logarithmic factors. We also define a coarser notion of subGaussianity used subsequently in our proof sketch,

**Definition 1** (($\beta^2, K$)-subGaussianity)**.** *A mean-zero random variable $Y$ is said to be $(\beta^2, K)$-subGaussian if it satisfies $\mathbb{P}(|Y| > A) \leq e^K \exp(-\frac{A^2}{2\beta^2})$.*

**Ornstein-Uhlenbeck process:** Consider a target distribution $\pi$ over $\mathbb{R}^d$. Suppose $x_0 \sim \pi$ and $x_t$ solve the following Stochastic Differential Equation (SDE):

$$dx_t = -x_t dt + \sqrt{2}dB_t \,, \tag{1}$$

where $B_t$ is the standard Brownian Motion over $\mathbb{R}^d$. An application of Ito's formula demonstrates that $x_t = x_0 e^{-t} + z_t$ where $z_t \sim \mathcal{N}(0, \sigma_t^2 \mathbf{I})$ is independent of $x_0$ and $\sigma_t := \sqrt{1 - e^{-2t}}$. This is the forward noising process, which progressively noises the initial sample into a standard Gaussian vector. Ito's formula also relates $x_t, x_{t'}$ for any timesteps $t > t' \geq 0$ to obtain, $x_t = x_{t'} e^{-(t-t')} + z_{t,t'}$ where $z_{t,t'} \sim \mathcal{N}(0, \sigma_{t-t'}^2 \mathbf{I})$ is independent of $x_{t'}$ and $\sigma_{t-t'} := \sqrt{1 - e^{-2(t-t')}}$. For $t \in [0, T]$, let $p_t$ be the probability density function of $x_t$. Given $\bar{x}_0 \sim p_T$ and a standard $\mathbb{R}^d$ Brownian motion $\bar{B}$, then the denoising process is:

$$d\bar{x}_t = \bar{x}_t dt + 2\nabla \log p_{T-t}(\bar{x}_t)dt + \sqrt{2}d\bar{B}_t \,. \tag{2}$$

It is the time reversal of the noising process which implies $\bar{x}_T \sim \pi$ [45].

**Score matching:** Given i.i.d. data points $x^{(1)}, \ldots, x^{(m)}$ from the target distribution $\pi$, diffusion models learn the score function $s(t, x) : \mathbb{R}^+ \times \mathbb{R}^d \to \mathbb{R}^d$ defined as $s(t, x) \equiv s_t(x) := \nabla \log p_t(x)$ via denoising score matching (DSM). Tweedie's formula states that

$$s(t, x_t) = \mathbb{E}\left[\frac{-z_t}{\sigma_t^2} \Big| x_t\right]. \tag{3}$$

Let $\mathcal{H}$ be a finite class of functions which map $\mathbb{R}^+ \times \mathbb{R}^d$ to $\mathbb{R}^d$ with functions $(t, x) \to f(t, x) \equiv f_t(x)$. Let $\mathcal{T} = \{t_1, \ldots, t_N\} \subseteq [0, T]$. Let $x_t^{(i)}$ denote the solution of Equation (1) at time $t$ with $x_0^{(i)} = x^{(i)}$ and define $z_t^{(i)} := x_t^{(i)} - e^{-t}x^{(i)}$. We consider the joint DSM objective to be:

$$\hat{\mathcal{L}}(f) := \frac{1}{mN} \sum_{i=1}^m \sum_{t \in \mathcal{T}} \Big\| f(t, x_t^{(i)}) + \frac{z_t^{(i)}}{\sigma_t^2} \Big\|_2^2. \tag{4}$$

Thus, optimizing (4) is a regression task with noisy labels. The two sources of error are: i) By (3), while $-\mathbb{E}[z_t/\sigma_t^2 | x_t] = s(t, x_t)$, $z_t/\sigma_t^2$ is noisy conditioned on $x_t$ and ii) $x_t \sim p_t$ itself is random. The empirical risk minimizer is defined as $\hat{f} = \arg\inf_{f \in \mathcal{H}} \hat{\mathcal{L}}(f)$. The results established in [1] states that the error in sampling arising from using the estimated score function $\hat{f}$ is given by:

$$\epsilon_{\text{score}}^2(\hat{f}) := \sum_{i=2}^N \gamma_i \mathbb{E}_{x \sim p_{t_i}}\left[\|\hat{f}(t_i, x) - s(t_i, x)\|_2^2\right], \text{ where } \gamma_i := t_i - t_{i-1} \tag{5}$$

Our goal is to bound this error. For simplicity, we consider $t_i = i\Delta$ for some step size $\Delta \in (0, 1)$.

## 3 Main results

We operate under the following smoothness assumption on the function class, $\mathcal{H}$.

**Assumption 1** (Smoothness of function class)**.** *Let the true score function, $s \in \mathcal{H}$.*

    *0. $\nabla \log p_t(\cdot)$ is continuously differentiable for every $t \in \mathbb{R}^+$.*

    *1. Lipschitzness : For all $t \in \mathcal{T}, x_1, x_2 \in \mathbb{R}^d$, $f \in \mathcal{H}$: $\|f(t, x_1) - f(t, x_2)\|_2 \leq L\|x_1 - x_2\|$*

    *2. Local Time Regularity : There exists a set $B_{\delta, t}$ such that $p_t(B_{\delta, t}) \geq 1 - \delta$, $\forall t \geq t' \in \mathcal{T}, x \in B_{\delta, t}, \forall f \in \mathcal{H}$*

$$\|e^{-(t-t')}f(t, x) - f(t', e^{(t-t')}x)\|_2 \leq e^{t-t'}L\sqrt{8(t - t')\log(\tfrac{2}{\delta})}$$

Assumption (1) is a standard Lipschitz continuity assumption followed in the literature (see e.g. [3])[2]. Assumption (2) assumes Hölder continuity with respect to the time variable. This is a natural assumption because Lemma 10 shows that Assumption 1-1 implies Assumption 1-2 for the true score

---

[2]We note that our results generalize to a time-varying Lipschitz parameter $L(t)$. This lets us replace the worst case Lipschitz constant by its time-averaged variant.

function, $s(t, x)$. We also note that the assumption $s \in \mathcal{H}$ can be relaxed to assume that $\exists \bar{s} \in \mathcal{H}$ with sufficiently small $\ell_2$ error, similar to [14]. While we assume a finite function class, $\mathcal{H}$, we can extend it to infinite classes by a standard covering argument in learning theory or consider $\mathcal{H}$ to be the finite class of floating point quantized models such as neural networks.

Equation (1) demonstrates that $x_t$ forms a Markov chain, leading to the noise random variables, $z_t$, being dependent. Additionally, (1) is typically iterated for $T = \tilde{O}(\tau_{\text{mix}})$ timesteps, until $p_T$ is close to a gaussian distribution. This setup falls outside the scope of conventional analyses of learning from dependent data (see Section 1.2). Such analyses usually assume a significantly larger number of datapoints, where datapoints separated by $\tau_{\text{mix}}$ in time are approximately independent, and the convergence rates align with their i.i.d. counterparts, adjusted for an effective sample size reduced by a factor of $\tau_{\text{mix}}$. In contrast, our setting involves substantially fewer datapoints. To address this challenge, we propose a novel martingale decomposition (stated in Lemma 2 and proved in Lemma 20) of the error and establish sharp concentration bounds to account for these dependencies.

Recall the DSM objective in (4). As explained before, there are two sources of noise: (1) due to $-z_t/\sigma_t^2$ conditioned on $x_t$, (2) due to $x_t \sim p_t$. We demonstrate the effect of fluctuations in $z_t | x_t$ in Theorem 1 and then deal with the random fluctuations due to $x_t$ in Theorem 2.

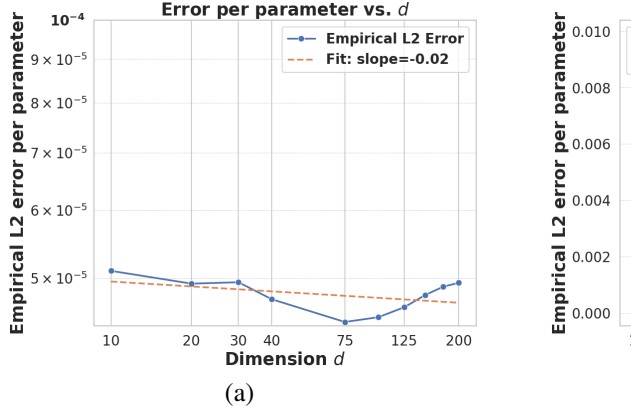
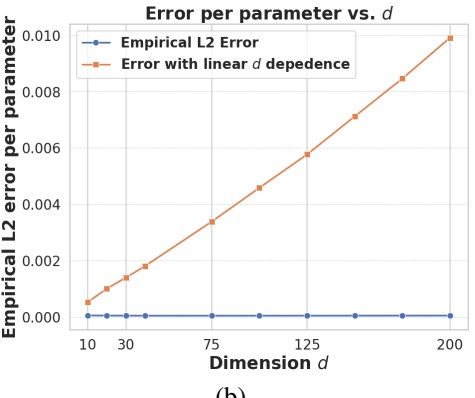

          (a)                                          (b)

Figure 1: (a) Empirical L2 error ((4)), scaled inversely by $\log(|\mathcal{H}|) \log\log(d)$, on a $\log - \log$ scale. A linear fit to the points shows a nearly zero slope, consistent with our $\log\log d$ dimension dependence. (b) Comparison of scaled empirical L2 error, vs. the scaled error if there were a linear dimension dependence as in prior works. As discussed subsequently, all previous works provide scaled error bounds with atleast a linear dependence.

Our first result in Theorem 1 provides a dimension-free bound on the empirical squared error, wherein we show how to control the noise due to $z_t$, conditioned on the data, $x_t$.

**Theorem 1** (Empirical $L_2$ Bound). *Let Assumption 1 hold. Fix $\delta \in (0, 1)$. For all $j \in [N]$, let $t_j := \Delta j$ and $\gamma_j := \Delta$. Let $B := C \log\left((L+1) dmN \log\left(\frac{1}{\delta}\right) / \Delta\right)$ for an absolute constant $C > 0$, and let $\Delta \log^3(\frac{1}{\Delta}) d^3 \log^3(2d) \log^3\left(\frac{2Nm}{\delta}\right) \log^3\left(\frac{B|\mathcal{H}|}{\delta}\right) \le 1$ and $N\Delta \le C \log(\frac{1}{\Delta})$. Then for*

$$m \gtrsim \frac{(L+1)^2}{\epsilon^2} \log\left(\frac{B|\mathcal{H}|}{\delta}\right) N\Delta$$

*with probability at least $1 - \delta$,*

$$\sum_{i \in [m], j \in [N]} \frac{\gamma_j \left\| \hat{f}\left(t_j, x_{t_j}^{(i)}\right) - s\left(t_j, x_{t_j}^{(i)}\right) \right\|_2^2}{m} \lesssim \epsilon^2$$

**Remark 1.** *All prior works such as [3, 14, 12] have at least a linear dimension dependence apart from the $\log(|\mathcal{H}|)$. In contrast, modulo $\log(|\mathcal{H}|)$, our bounds are nearly dimension-free ( $\log\log(d)$ dependence due to $B$), and instead depend on the smoothness parameter $L$. Therefore, we bring down the complexity from $\text{poly}(d) \log(|\mathcal{H}|)$ to $\log(|\mathcal{H}|)$ which is meaningful in high dimensions.*

*To put our results in context, prior work [12] shows that score matching suffers from exponential dependence on $d$ in the worst case. In contrast, we show that when the target distribution admits a*

*suitable class of estimators with mild smoothness assumptions, score estimation becomes sample-efficient. This closes the gap between theoretical results and empirical findings in diffusion models, where global optimization reliably learns accurate score functions for natural images, despite the worst-case guarantees for training neural networks.*

*In Figure 1 (see Appendix Section F for details of experimental setup), we train a 2-layer neural network with a fixed hidden dimension and sample size, to sample from $\mathcal{N}(0, \Sigma)$, and measure the L2 error across timesteps, described in (4), scaled inversely by $\log(|\mathcal{H}|) \log\log(d)$. Here we use the fact that $\log(|\mathcal{H}|)$ scales linearly with the number of parameters for neural networks (with quantization). Figure 1 is consistent with the dimension-free bounds in Theorem 1.*

**Remark 2.** *Theorem 1 requires the time-discretization $\Delta = \widetilde{O}(1/d^3)$. However, this is not a problem during training due to the use of stochastic optimization algorithms which obtain minibatches of datapoints at randomly sampled times. We illustrate this further by distinguishing between the statistical and optimization questions underlying our results.*

***Statistical question***: *The standard loss formulation in both theory and practice has time discretization $\Delta = 0$, corresponding to an integral/expectation over time. See Equation (17) in [17] and Equation (7) in [43]. We consider a fine approximation of the integral with $\Delta = O(1/d^3)$ due to purely technical reasons since a $\log\log(1/\Delta)$ term appears in the error bound. Note that $\Delta$ can be made much smaller than $O(1/d^3)$ due to dependence. In fact, due to assumption 1, the difference between the integral (i.e, $\Delta = 0$) and our loss is $O(\sqrt{\Delta})$. Therefore, our framework allows for considering the standard integral formulation of the loss as well.*

***Optimization question***: *The integral or the large sum in the loss function is computationally intractable to optimize directly. Hence stochastic optimization algorithms such as SGD or Adam are used. Here random batches of datapoints are drawn to evaluate the stochastic gradients. When the times are sampled randomly, we obtain unbiased estimators for the gradients of this loss, even when (or when it is very small). Therefore, practical training can be performed efficiently even when $\Delta = O(1/d^3)$.*

*We further note that this does not adversely affect inference since Theorem 3 shows that one can use larger step sizes during inference without deteriorating quality.*

**Remark 3.** *The parameter $B$ arises from our martingale-based concentration analysis, which involves subGaussian random variables whose subGaussianity parameters are themselves random. We show that with high probability, these parameters are uniformly $O(\exp(B))$ via a union-bound, leading to the $\log(B)$ factor. Refer to Lemma 17 in the Appendix for the detailed argument.*

Theorem 1 is the first step in proving the expectation bound in Theorem 2 and may be of independent interest. Theorem 2 deals with the noise arising from the data $x_t \sim p_t$. Our next assumption, called 'hypercontractivity', controls the 4th-moment of the error bound with respect to the 2nd-moment, which can be used to prove the generalization of the score function in $L^2$ error. This is a mild assumption, standard in statistics and learning theory under heavy tails [30, 23, 33].

**Assumption 2.** *Let $\kappa > 0$ be a fixed constant. Then, for every $f \in \mathcal{H}$ and $x_t \sim p_t$, we have:*

$$\mathbb{E}[\|f(t, x_t) - s(t, x_t)\|^4]^{\frac{1}{4}} \leq \kappa \mathbb{E}[\|f(t, x_t) - s(t, x_t)\|^2]^{\frac{1}{2}}$$

$\kappa^4$ can be bounded (up to multiplicative constants) by the kurtosis of $f(t, x_t) - s(t, x_t)$. Assumption 2 follows from the smoothness and strong convexity of neural networks in the parameter space (not $x_t$), as shown in Lemma 40 in the Appendix. Recent work [32, 52] shows that near the global minimizer of the population loss, many smooth non-convex losses exhibit local strong convexity.

Under Assumptions 1 and 2, we state our main result in Theorem 2. In this result, we use Theorem 1 and handle the noise due to $x_t \sim p_t$ in the DSM objective.

**Theorem 2** ($L_2$ Error Bound). *Let Assumptions 1 and 2 hold. Fix $\delta \in (0, 1)$. For all $j \in [N]$, let $t_j := \Delta j$ and $\gamma_j := \Delta$. Let $B := C \log\left((L+1)\,dmN \log\left(\frac{1}{\delta}\right)/\Delta\right)$ for an absolute constant $C > 0$, and let $\Delta \log^3(\frac{1}{\Delta})d^3 \log^3(2d) \log^3\left(\frac{2Nm}{\delta}\right) \log^3\left(\frac{B|\mathcal{H}|}{\delta}\right) \leq 1$ and $N\Delta \leq C \log(\frac{1}{\Delta})$. If*

$$m \gtrsim \kappa^2 \max\left\{\log\left(\frac{N}{\delta}\right), \frac{(L+1)^2 N\Delta}{\epsilon^2} \log\left(\frac{B|\mathcal{H}|}{\delta}\right)\right\}$$

*then with probability at least $1 - \delta$,*

$$\sum_{j \in [N]} \gamma_j \mathbb{E}_{x_{t_j}} \left[ \left\| \hat{f}\left(t_j, x_{t_j}\right) - s\left(t_j, x_{t_j}\right) \right\|_2^2 \right] \lesssim \epsilon^2$$

**Remark 4.** *In addition to the sample complexity of Theorem 1, the sample complexity for the generalization bound in Theorem 2 additionally has a factor of $\kappa^2$ due to the local strong convexity assumption formalized in Lemma 40.*

We note that Theorem 2 considers small values of time discretization $\Delta$, which is not an issue during training (see Remark 2). However, we can accelerate inference by using a larger timestep-size to discretize the diffusion process, as shown in Theorem 3 and proved in Theorem 5.

**Theorem 3** (Fast Inference). *Under the same assumptions as Theorem 2, partition the timesteps $\{t_j = \Delta j\}_{j \in [N]}$ into $k$ disjoint subsets $S_1, S_2, \ldots, S_k$, where each subset $S_i$ contains timesteps of the form $t_j = \Delta(i + nk)$ for $n \in \mathbb{N}$. Define $\gamma_j' := k\Delta$ for all $j$ in any subset $S_i$. Then, there exists at least one subset $S_i$ such that, with probability at least $1 - \delta$.*

$$\sum_{j \in S_i} \gamma_j' \mathbb{E}_{x_{t_j}} \left[ \left\| \hat{f}(t_j, x_{t_j}) - s(t_j, x_{t_j}) \right\|_2^2 \right] \lesssim \epsilon^2,$$

The subsets $S_i$ allow for a much coarser discretization with differences being $k\Delta$ instead of $\Delta$. While the error due to discretization of the SDE might become worse, as shown by the bounds in [1], Theorem 3 demonstrates that the score estimation error does not degrade.

**Comparison with prior work:** [3] and [14] analyze the DSM objective (4) by independently bounding the error at each timestep and applying a union bound over all $t \in \mathcal{T}$. [3] assume bounded support over an $\ell_2$ ball of radius $R$ and $L$-Lipschitz score functions, and derive (up to logarithmic factors) the following per-timestep bound using Rademacher complexity:

$$\mathbb{E}_{x_t} \left[ \left\| \hat{f}\left(t, x_t\right) - s\left(t, x_t\right) \right\|_2^2 \right] \lesssim \frac{\epsilon^2}{\sigma_t^2} \tag{6}$$

where $\mathcal{R}_n(\mathcal{H})$ is the Rademacher complexity of $\mathcal{H}$. When using a uniform step size $\Delta$, this leads to sample complexity scaling as $\frac{1}{\text{poly}(\Delta)}$, with at least linear dependence on $d$, especially since $R = O(\sqrt{d})$ in practice.

[14] improve the dependence of sample complexity for Wasserstein error, removing smoothness assumptions on the score function and relaxing the $\ell_2$ error. They show that learning $f \in \mathcal{H}$ satisfying the relaxed criterion for each $t$ suffices for sampling, with a per-timestep sample complexity of $m \gtrsim d \log \left( \frac{|\mathcal{H}|}{\delta} \right) / \epsilon^2$, again using a union bound over time. Their bound avoids dependence on $\frac{1}{\sigma_t}$, but retains linear scaling with $d$.

[16] study gradient descent for optimizing (4) when $s$ is $L$-Lipschitz and the target distribution is within an $\ell_2$ ball of radius $R$. They model time as an input and use a kernel regression perspective to jointly learn across timesteps. While conceptually similar to our approach, their sample complexity (Theorem 3.12) still exhibits polynomial dependence on $d$.

[12] analyze the non-parametric setting under the assumption that the score belongs to a Hölder class with $\beta$-smoothness. This leads to exponential sample complexity in $d$ unless $\beta = \Omega(d)$, aligning with worst-case lower bounds. In contrast, our work avoids this curse of dimensionality by making more pragmatic assumptions inspired by empirical diffusion model performance: (a) $\mathcal{H}$ only needs to approximate the target's score function; (b) mild time regularity; and (c) second-order differentiability of the log-density. These assumptions, also common in prior work [50, 3, 6, 35], enable nearly dimension-free generalization bounds.

## 4   Technical results

In this section we describe our proof techniques and key technical results. Figure 2 summarizes the key results in this section and how they work together to lead to Theorem 1.

For ease of exposition, we introduce some additional notation. For all timesteps $\{t_j\}_{j \in [N]}$, wherever its clear from context, we denote $\sigma_{t_j} \equiv \sigma_j$ and for all samples $i \in [m]$, we denote $x_{t_j}^{(i)} \equiv x_j^{(i)}$ and

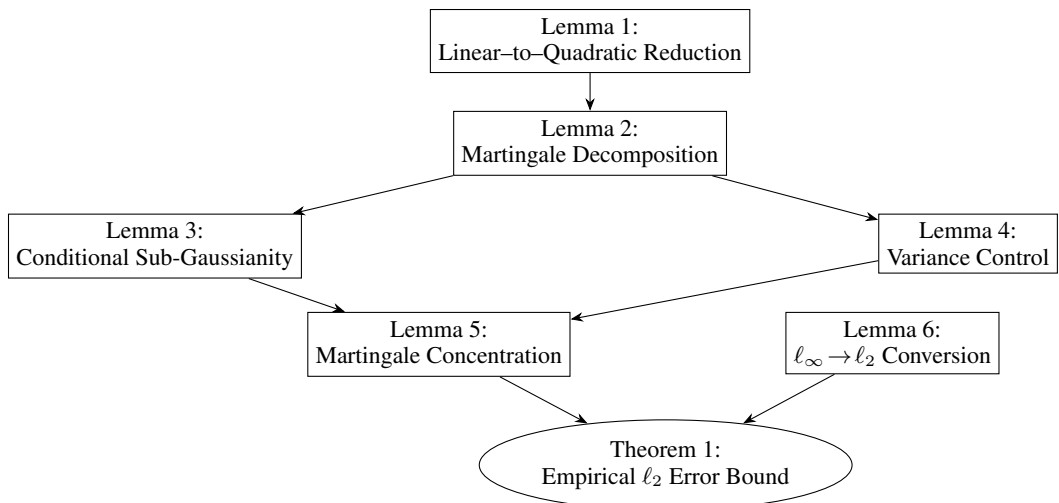

Figure 2: Dependency graph of the key lemmas leading to Theorem 1.

$z_{t_j}^{(i)} \equiv z_j^{(i)}$. For all appropriately defined function $f, g$, we denote the empirical expectations as $\widehat{\mathbb{E}}_{x_j}[f(x_j)] := \frac{1}{m} \sum_{i \in [m]} f(x_j^{(i)})$ and $\widehat{\mathbb{E}}_{x_j, z_j}[g(x_j, z_j)] := \frac{1}{m} \sum_{i \in [m]} g(x_j^{(i)}, z_j^{(i)})$. For any random variable $y$, we denote the conditional expectation as $\mathbb{E}_{i,j}[y] := \mathbb{E}[y|x_j^{(i)}]$.

We start by bounding the empirical squared error in terms of a linear form in Lemma 1. This relates the empirical error, $\hat{\mathcal{L}}$ of the minimizer $\hat{f}$ with the true score function, $s$. While we assume $s \in \mathcal{H}$ for simplicity, it can be relaxed to assume that $\exists s \in \mathcal{H}$ with sufficiently small $\ell_2$ error, similar to [14].

**Lemma 1.** *For $f \in \mathcal{H}$,*

$$\mathcal{L}(f) := \sum_{j \in [N]} \gamma_j \widehat{\mathbb{E}}_{x_j} \left[ \left\| f(t_j, x_j) - s(t_j, x_j) \right\|_2^2 \right],$$

$$H^f := \sum_{j \in [N]} \gamma_j \widehat{\mathbb{E}}_{x_j, z_j} \left[ \left\langle f(t_j, x_j) - s(t_j, x_j), \frac{-z_j}{\sigma_j^2} - s(t_j, x_j) \right\rangle \right].$$

*Let $\hat{\mathcal{L}}$ be as defined in* (4)*. If $s \in \mathcal{H}$ then for $\hat{f} = \arg\inf_{f \in \mathcal{H}} \hat{\mathcal{L}}(f)$, we have*

$$\mathcal{L}(\hat{f}) \leq H^{\hat{f}}, \tag{7}$$

Let $\hat{f}$ be the minimizer of $\hat{\mathcal{L}}(f)$. Lemma 1 (proved in Lemma 35) bounds $\mathcal{L}(\hat{f})$, the loss of $\hat{f}$ against the true and unknown score function with $H^{\hat{f}}$. We will show a high probability bound on $H^{\hat{f}}$ defined in (7) to control $\mathcal{L}(\hat{f})$. Interestingly, as shown in Lemma 2 (and proved in Lemma 20), for a fixed $f$ it is possible to decompose $H^f$ as a martingale difference sequence.

The martingale difference decomposition of $H^f$, exploiting the Markovian structure of (1), has terms of the form $Q_i := \langle G_i, Y_i - \mathbb{E}[Y_i | \mathcal{F}_{i-1}] \rangle$ adapted to the filtration $\{\mathcal{F}_i\}_{i \in [n]}$, where $G_i$ is a $\mathcal{F}_{i-1}$ measurable random variable. The proof primarily uses the fact that for $t_1 \leq t_2 \leq t_3$, $\mathbb{E}[x_{t_1} | x_{t_2}, x_{t_3}] = \mathbb{E}[x_{t_1} | x_{t_2}]$ due to the Markov property.

**Lemma 2.** *Let $\zeta := \frac{s-f}{m}$ for any $f \in \mathcal{H}$. Define*

$$\bar{G}_i := \sum_{j=1}^{N} \frac{\gamma_j e^{-(t_j - t_1)} \zeta\left(t_j, x_j^{(i)}\right)}{\sigma_j^2}, \quad G_{i,k} := \sum_{j=N-k+2}^{N} \frac{\gamma_j e^{-t_j} \zeta\left(t_j, x_j^{(i)}\right)}{\sigma_j^2}$$

*and define $R_{i,k}$ as*

$$R_{i,k} := \begin{cases} \langle G_{i,k+1}, \mathbb{E}_{i,N-k+1}[x_0^{(i)}] - \mathbb{E}_{i,N-k}[x_0^{(i)}] \rangle, & \text{for } k \in [N-1], \\ \langle \bar{G}_i, z_1^{(i)} - \mathbb{E}_{i,1}[z_1^{(i)}] \rangle, & \text{for } k = N. \end{cases}$$

and $R_{i,k} = 0$ for $k = 0$. Define $t_0 = 0$. Consider the filtration defined by the sequence of $\sigma$-algebras, $\mathcal{F}_{i,k} := \sigma(\{x_j^{(j)} : 1 \leq j < i, j \in [N]\} \cup \{x_j^{(i)} : j \geq N - k\})$, for $i \in [m]$ and $k \in \{0, \ldots, N\}$, satisfying the total ordering $\{(i_1, j_1) < (i_2, j_2) \text{ iff } i_1 < i_2 \text{ or } i_1 = i_2, j_1 < j_2\}$. Then,

1. For $k \in [N-1]$, $G_{i,k+1}$ is measurable with respect to $\mathcal{F}_{i,k-1}$, and $\bar{G}_i$ is $\mathcal{F}_{N-1}$-measurable.

2. For $i \in [m], k \in \{0\} \cup [N]$, $\{R_{i,k}\}_{(i,k)}$ forms a martingale difference sequence with respect to the filtration above.

3. $H^f = \sum_{i \in [m]} \sum_{k \in [N]} R_{i,k}$, where $H^f$ is defined in Lemma 1

In the above Lemma 2 (and proved in Lemma 20), $R_{ik}$ denotes the martingale difference sequence arising from the Doob decomposition (see e.g. [9]). Our aim is to bound $H^{\hat{f}}$ by bound $H^f$ uniformly for every $f$, using martingale concentration. In the next lemma, we show that conditioned on $\mathcal{F}_{i-1}$, $Q_i$ is subGaussian. To gain intuition into how subGaussianity comes into play in our context, we note that Lemma F.3. in [14] shows that the score function, $s(t, x_t)$, is $1/\sigma_t$-subGaussian. We develop a more fine-grained argument exploiting the smoothness of the score function to show subGaussianity (Definition 1) for our sequence. The proof is provided in Lemma 30.

**Lemma 3.** *Fix $\delta \in (0, 1)$. Consider $R_{i,k}$ and $\mathcal{F}_{i,k}$ as defined in Lemma 2 and let $\Delta := t_{N-k+1} - t_{N-k}$. Under Assumption 1, following the definition in Definition 1, conditioned on $\mathcal{F}_{i,k-1}$, $R_{i,k}$ is $(\beta_{i,k}^2 \|G_{i,k}\|^2, W_{i,k})$-subGaussian where $\beta_{i,k}, W_{i,k}$ are $\mathcal{F}_{i,k-1}$ measurable random variables such that $W_{i,k} \leq \log\left(\frac{2}{\delta}\right)$ with probability at-least $1 - \delta$ and*

$$\beta_{i,k} := \begin{cases} 8(L+1)e^{t_{N-k+1}}\sqrt{\Delta d}, & k \in [N-1], \\ 4\sqrt{\Delta d}, & k = N \end{cases}$$

However, the subGaussianity parameters in Lemma 3, depend polynomially on the data dimension, $d$ along with $G_i$ and the step size, $\Delta$. Solely relying on this leads to a dimension-dependent bound. To further refine our analysis and show a dimension-free bound, we evaluate the variance of $Q_i$ conditioned on $\mathcal{F}_{i-1}$. As shown in the next Lemma (Lemma 4) (and proved in Lemma 29), the variance depends only on the smoothness parameter, $L$, along with $G_i$ and $\Delta$.

**Lemma 4** (Variance bound for martingale difference sequence). *Consider the martingale difference sequence $R_{i,k}$ and the predictable sequence $G_{i,k+1}$ with respect to the filtration $\mathcal{F}_{i,k}$ from Lemma 21. Define $\Delta := t_{N-k+1} - t_{N-k}$. Then, $\mathbb{E}\left[R_{i,k}^2 | \mathcal{F}_{i,k-1}\right] \leq \nu_{i,k}^2$ where*

$$\nu_{i,k}^2 = \begin{cases} C(L\Delta^2 + \Delta + L^2\Delta)e^{2t_{N-k+1}}\|G_{i,k+1}\|^2, & \text{if } k \in [1, N-1], \\ C(L\Delta^2 + \Delta)\|\bar{G}_i\|^2, & \text{if } k = N. \end{cases}$$

*where $C > 0$ is an absolute constant and $\nu_{i,k}^2 = 0$ for $k = 0$.*

The proof of Lemma 4 is involved when $h_t(x) := \nabla^2 \log(p_t)(x)$ is not assumed to be Lipschtiz in $x$. Starting with the martingale difference sequence defined in Lemma 2, an application of the second-order tweedie's formula (see Lemma 22), reduces the problem to bounding the operator norm $\text{Cov}(s(t', x_{t'})|x_t)$ for $t - t' = \Delta > 0$, i.e, the conditional covariance matrix of the score function given the future. Exploiting the smoothness assumption on the score function, an application of the mean value theorem reduces our problem to bounding the operator norm of:

$$\mathbb{E}\left[h_{t'}(y_{t'})(x_{t'} - \tilde{x}_{t'})(x_{t'} - \tilde{x}_{t'})^\top h_{t'}(y_{t'})^\top | x_t\right], t' < t$$

for $x_{t'}, \tilde{x}_{t'}$ i.i.d conditioned on $x_t$ and $y_{t'} = \lambda x_{t'} + (1 - \lambda)\tilde{x}_{t'}$, $\lambda \in (0, 1)$. Notice that $y_{t'}|x_t$ is dependent on $x_{t'}, \tilde{x}_{t'}|x_t$, which does not allow the use of Tweedie's second-order formula (Lemma 22) to bound $\mathbb{E}\left[(x_{t'} - \tilde{x}_{t'})(x_{t'} - \tilde{x}_{t'})^\top | x_t\right]$ and derive variance bounds that are dimension-free. To approximately allow this argument, we decompose $h_{t'}(y_{t'})$ into two components:

$$h_{t'}(y_{t'}) = h_{t',\epsilon}(y_{t'}) + (h_{t'}(y_{t'}) - h_{t',\epsilon}(y_{t'})).$$

The first term, $h_{t',\epsilon}(y_{t'})$, represents a hessian after being smoothed with an appropriately chosen distribution, which we show satisfies Lipschitz continuity. This allows us to approximate $h_{t',\epsilon}(y_{t'}) \approx$

$h_{t',\epsilon}(e^{\Delta}x_t)$ and bound the variance with Tweedie's second order formula. The second term, which represents the deviation between the original and mollified Hessians, is bound using Lusin's theorem (Lemma 27) to provide approximate uniform continuity for $h_{t'}$, as developed further in Lemma 28.

Putting together Lemma 3 and Lemma 4, we provide a general concentration tool for martingale difference sequences with bounded variance and subGaussianity in Lemma 5. We follow a similar proof strategy via a supermartingale argument as in the proof Freedman's inequality (see for e.g. [47]), but diverge in dealing with subGaussianity instead of almost surely bounded random variables.

**Lemma 5.** *Let $M_n = \sum_{i=1}^n \langle G_i, Y_i - \mathbb{E}[Y_i|\mathcal{F}_{i-1}]\rangle$, $M_0 = 1$ and the filtration $\{\mathcal{F}_i\}_{i\in[n]}$ be such that $G_i$ is $\mathcal{F}_{i-1}$ measurable and*

*1. $\langle G_i, Y_i - \mathbb{E}[Y_i|\mathcal{F}_{i-1}]\rangle$ is $(\beta_i^2\|G_i\|^2, K_i)$ sub-Gaussian conditioned on $\mathcal{F}_{i-1}$ (where $\beta_i, K_i$ are random variables measurable with respect to $\mathcal{F}_{i-1}$)*

*2. $\mathsf{var}(\langle G_i, Y_i - \mathbb{E}[Y_i|\mathcal{F}_{i-1}]\rangle|\mathcal{F}_{i-1}) \leq \nu_i^2\|G_i\|^2$ and define $J_i := \max(1, \frac{1}{K_i}\log\frac{\beta_i^2 K_i}{\nu_i^2})$.*

*Pick a $\lambda > 0$ and let $\mathcal{A}_i(\lambda) = \{\lambda J_i\|G_i\|\beta_i\sqrt{K_i} \leq c_0\}$ for some small enough universal constant $c_0$. Then, there exists a universal constant $C > 0$ such that*

$$\forall v > 0, \ \mathbb{P}(\{\lambda M_n > C\lambda^2\sum_{i=1}^n \nu_i^2\|G_i\|^2 + v\} \cap_{i=1}^n \mathcal{A}_i(\lambda)) \leq \exp(-v)$$

Observe that the concentration result developed in Lemma 5 (and proved in Lemma 16) has two parts. Optimizing over the choice of $\lambda$, it can be shown that the bound on $M_n$ depends on two terms: (1) an $\ell_2$ term, $\sum_{i\in[n]} \nu_i^2\|G_i\|^2$ and (2) an $\ell_\infty$ term, $\sup_{i\in[n]} J_i\|G_i\|\beta_i\sqrt{K_i}$. When applied in our context, these two terms in turn depend on norms, $\|f - s\|_2$ and $\|f - s\|_\infty$. This is where the time-regularity assumption in Assumption 1 plays a crucial role in our analysis. Specifically, it enables us to bridge the $\ell_\infty$ and $\ell_2$ norm bounds derived from the martingale concentration results in Lemma 5. The proof of Lemma 6 leverages this assumption to relate $\|f(t + k\Delta, x_{t+k\Delta})\|_2$ to $\|f(t, x_t)\|_2$, as shown by:

$$\|f(t + k\Delta, x_{t+k\Delta})\|_2 - e^{k\Delta}\|f(t, x_t)\|_2 \geq -\tilde{\Omega}(L\sqrt{dk\Delta}).$$

Exploiting this property over a carefully selected range of $k$ values allows us to relate $\ell_\infty$ and $\ell_2$ norm bounds as we show in the following Lemma.

**Lemma 6.** *Under Assumption 1, with probability $1 - \delta$, for a universal constant $C > 0$ the following holds uniformly for every $f \in \mathcal{H}$:*

$$\left[\sup_{\substack{i\in[m]\\j\in[N]}} \|f(t_j, x_j) - s(t_j, x_j)\|_2\right]^2 \leq C\Delta^{\frac{1}{3}}\left[\sum_{\substack{i\in[m]\\j\in[N]}} \|f(t_j, x_j) - s(t_j, x_j)\|_2^2\right] + CL^2 d\Delta^{\frac{2}{3}}\log\left(\frac{Nm}{\delta}\right)$$

Lemma 6 establishes that the simultaneous analysis of all timesteps uses the smoothness across time. In the absence of this approach, the smoothness assumption in the $x_t$-space would lack dependence on $\Delta$ and could grow as large as the Lipschitz constant $L$. This is essential for establishing nearly dimension-independent bounds. The proof of this result can be found in Lemma 39 in the Appendix.

## 5 Bootstrapped score matching

In Section 4, we used time regularity and could prove nearly $d$-independent bounds. Learning with the same function class across timesteps and Assumption 1 was critical to our proof.

We now attempt to exploit the dependence across timesteps explicitly and reduce variance in estimation. Using the Markovian nature of (1), we show that for any $t' < t$ and $\alpha_t \in \mathbb{R}$, $s(t, x_t) = \mathbb{E}[\tilde{y}_t|x_t]$ for $\tilde{y}_t := -\frac{z_t}{\sigma_t^2} - \alpha_t(s(t', x_{t'}) - \frac{-z_{t'}}{\sigma_{t'}^2})$. This shows that $\tilde{y}_t$ can also be used to construct a learning target for the score function. This is in contrast to the target $y_t := -\frac{z_t}{\sigma_t^2}$ used in (4). The advantage of $\tilde{y}_t$ over $y_t$ is in the lower variance of $\tilde{y}_t$, as shown in Lemma 7 (proved in Lemmas 42, 43).

**Lemma 7** (Bootstrap Properties). *Let $\tilde{r}_t := \tilde{y}_t - s(t, x_t)$. For $t' < t$, let $\Delta := t - t'$ and $\alpha_t := \frac{e^{-\Delta}\sigma_{t'}^2}{\sigma_t^2}$. Then, under Assumption 1, we have $\mathbb{E}[\tilde{r}_t|x_t] = 0$ and $\left\|\mathbb{E}[\tilde{r}_t\tilde{r}_t^\top|x_t]\right\|_{\mathrm{op}} = O\left(\frac{(L^2+1)\Delta}{\sigma_t^4}\right)$.*

To compare with $y_t = \frac{-z_t}{\sigma_t^2}$, we note that an application of the second order tweedie's formula along with Assumption 1 shows the variance $\left\|\mathbb{E}[(y_t - s(t, x_t))(y_t - s(t, x_t))^\top | x_t]\right\|_{\text{op}}$ to be of the order $O(\frac{L+1}{\sigma_t^2})$. Therefore, although both $y_t$ and $\tilde{y}_t$ are unbiased, the variance of $\tilde{y}_t$ has an additional step size ($\Delta$) factor in the numerator (see Lemma 7)

The BSM algorithm (described in detail in Appendix G) operates sequentially over a discretized time horizon $0 = t_0 < t_1 < \cdots < t_N = T$ and builds upon the principles of DSM while introducing a novel bootstrapping mechanism to mitigate the increasing variance of the DSM loss in later timesteps. Given a dataset $D = \{x_0^{(i)}\}_{i \in [m]}$ sampled from the data distribution, the perturbed samples at timestep $t_k$ are generated as $x_{t_k}^{(i)} = x_0^{(i)} e^{-t_k} + z_{t_k}^{(i)}$, $z_{t_k}^{(i)} \sim \mathcal{N}(0, \sigma_{t_k}^2 I)$ where $\sigma_{t_k}^2 = 1 - e^{-2t_k}$. The task at each timestep $t_k$ is to estimate an approximate score function $\hat{s}_{t_k}(x)$ to optimize $\mathbb{E}_{x_{t_k}}[\|s(t_k, x) - \hat{s}_{t_k}(x)\|_2^2]$. For the initial timesteps $t_k$ with $k \leq k_0$, the algorithm employs DSM. The score function $\hat{s}_{t_k}$ is obtained by solving $\hat{s}_{t_k} = \arg\min_{f \in \mathcal{H}_k} \sum_{i \in [m]} \frac{1}{m} \|f(t_k, x_{t_k}^{(i)}) - \frac{-z_{t_k}^{(i)}}{\sigma_{t_k}^2}\|_2^2$. For later timesteps $t_k$ with $k > k_0$, the algorithm transitions to BSM. At each timestep, the algorithm constructs bootstrapped targets $\tilde{y}_{t_k}^{(i)}$ by combining the DSM target $\frac{-z_{t_k}^{(i)}}{\sigma_{t_k}^2}$ with the previously estimated score $\hat{s}_{t_{k-1}}$. Specifically, the targets are defined as:

$$\tilde{y}_{t_k}^{(i)} = (1 - \alpha_k) \underbrace{\frac{-z_{t_k}^{(i)}}{\sigma_{t_k}^2}}_{\text{Unbiased Target}} + \alpha_k \underbrace{\left( \frac{-z_{t_k}^{(i)}}{\sigma_{t_k}^2} + \left( \hat{s}_{t_{k-1}}(x_{t_{k-1}}^{(i)}) - \frac{-z_{t_{k-1}}^{(i)}}{\sigma_{t_{k-1}}^2} \right) \right)}_{\text{Biased Target}}$$

where $\alpha_k = e^{-\gamma_k} \sqrt{\frac{1 - e^{-2t_{k-1}}}{1 - e^{-2t_k}}}$, with $\gamma_k = t_k - t_{k-1}$. Given access to the true score function, $s(t_{k-1}, .)$, then $\tilde{y}_{t_k}^{(i)}$ would form an unbiased target with lower variance, as shown in Lemma 7. However, since we only have access to the estimated score function, $\hat{s}_{t_{k-1}}$ at the previous timestep, $\tilde{y}_{t_k}^{(i)}$ is a biased target, and the parameter $\alpha_k$ weighs between the biased and unbiased targets. The score function, $\hat{s}_{t_k}$, is then learned as: $\hat{s}_{t_k} \leftarrow \arg\min_{f \in \mathcal{H}_k} \sum_{i \in [m]} \frac{1}{m} \|f(t_k, x_{t_k}^{(i)}) - \tilde{y}_{t_k}^{(i)}\|_2^2$.

# 6 Conclusion

To our knowledge, this is *the first work* to establish (nearly) dimension-free sample complexity bounds for learning score functions across noise levels. We show that a mild assumption of time-regularity can significantly improve over previous bounds which have polynomial dependence on $d$. We achieve this with a novel martingale-based analysis with sharp variance bounds, addressing the complexities of learning from dependent data generated by multiple Markov process trajectories. Furthermore, we introduce the Bootstrapped Score Matching (BSM) method, which effectively leverages temporal information to reduce variance and enhance the learning of score functions. While we provide theoretical insights into the training of diffusion models, several open questions still remain. One potential direction is extending our framework to flow-matching models where such bounds could yield further insights. Additionally, while BSM is a compelling algorithm, establishing rigorous theoretical and empirical performance guarantees is an open problem which we leave for future work.

## Acknowledgments and Disclosure of Funding

We gratefully acknowledge NSF grants 2217069, 2019844, and DMS 2109155. Additionally, part of this work was carried out while Syamantak was an intern at Google DeepMind.

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

## A    Utility Results

**Definition 2** (norm subGaussian)**.** *We will call a random vector $X \in \mathbb{R}^d$ to be $\sigma$ norm subGaussian if $\mathbb{E}X = 0$ and*

$$\mathbb{E}\exp(\tfrac{\|X\|^2}{\sigma^2}) \leq 2 \,.$$

**Definition 3.** *We will call a random vector $X \in \mathbb{R}^d$ to be $\sigma$ subGaussian if $\mathbb{E}X = 0$ and for every $v \in \mathbb{R}^d$ and $\lambda \in \mathbb{R}$ we have:*

$$\mathbb{E}\exp(\lambda\langle v, X\rangle) \leq \exp(\tfrac{\lambda^2\|v\|^2\sigma^2}{2}) \,.$$

**Lemma 8.** *Let $X \sim \mathcal{N}\left(0, \sigma^2\mathbf{I}\right)$. Then, $X$ is $2\sigma$ norm subGaussian.*

*Proof.* Consider the random variable $y := \frac{\|X\|_2^2}{\sigma^2}$. Then, $y \sim \chi(d)$ follows the chi-squared distribution with $d$ degrees of freedom. Therefore, for any $t < \frac{1}{2}$,

$$\mathbb{E}\left[\exp\left(t\frac{\|X\|_2^2}{\sigma^2}\right)\right] = (1 - 2t)^{-\frac{d}{2}}$$

Setting $t = \frac{1}{4d}$, we have

$$\mathbb{E}\left[\exp\left(\frac{\|X\|_2^2}{(2\sigma)^2}\right)\right] = \left(1 - \frac{1}{2d}\right)^{-\frac{d}{2}} = \left(\left(1 - \frac{1}{2d}\right)^{-2d}\right)^{\frac{1}{4}} \leq 2$$

$\square$

**Lemma 9.** *For all $t > 0$, $x_1, x_2 \in \mathbb{R}^d$, consider any function $u : \mathbb{R}^d \rightarrow \mathbb{R}^d$ satisfying $\|u\left(x_1\right) - u\left(x_2\right)\|_2 \leq S\|x_1 - x_2\|_2$, where $S > 0$ is a fixed constant. For timesteps $0 \leq t' < t$, consider the random variable*

$$q_{t,t'} := u\left(x_{t'}\right) - \mathbb{E}\left[u\left(x_{t'}\right)|x_t\right]$$

*where $x_t$ is defined in* (1)*. Then, $q_{t,t'}$ is $\phi\sqrt{d}$ norm subGaussian for*

$$\phi := 4Se^\Delta\sqrt{1 - e^{-2\Delta}}$$

*where $\Delta := t - t'$.*

*Proof.* We first note that

$$\mathbb{E}_{x_t,x_{t'}}\left[q_{t,t'}\right] = \mathbb{E}_{x_{t'}}\left[u_{t'}\left(x_{t'}\right)\right] - \mathbb{E}_{x_t}\left[\mathbb{E}\left[u_{t'}\left(x_{t'}\right)|x_t\right]\right] = 0$$

Using Lemma 1 from [22], we show that $\mathbb{E}_{x_{t'},x_t}\left[\exp\left(\frac{\|q_{t,t'}\|_2^2}{\phi^2 d}\right)\right] \le 2$. Let $x'_{t'}$ be an iid copy of $x_{t'}$, conditioned on $x_t$. Then, we have,

$$
\begin{aligned}
\mathbb{E}_{x_{t'},x_t}\left[\exp\left(\frac{\|q_{t,t'}\|_2^2}{\phi^2 d}\right)\right] &= \mathbb{E}_{x_t}\left[\mathbb{E}_{x_{t'}}\left[\exp\left(\frac{\|q_{t,t'}\|_2^2}{\phi^2 d}\right)\Big| x_t\right]\right] \\
&= \mathbb{E}_{x_t}\left[\mathbb{E}_{x_{t'}}\left[\exp\left(\frac{\|u(x_{t'}) - \mathbb{E}_{x_{t'}}[u(x_{t'})|x_t]\|_2^2}{\phi^2 d}\right)\Big| x_t\right]\right] \\
&= \mathbb{E}_{x_t}\left[\mathbb{E}_{x_{t'}}\left[\exp\left(\frac{\|u(x_{t'}) - \mathbb{E}_{x'_{t'}}[u(x'_{t'})|x_t]\|_2^2}{\phi^2 d}\right)\Big| x_t\right]\right] \quad (8) \\
&= \mathbb{E}_{x_t}\left[\mathbb{E}_{x_{t'}}\left[\exp\left(\frac{\|\mathbb{E}_{x'_{t'}}[u(x_{t'}) - u(x'_{t'})|x_t]\|_2^2}{\phi^2 d}\right)\Big| x_t\right]\right] \\
&\le \mathbb{E}_{x_t}\left[\mathbb{E}_{x_{t'}}\left[\exp\left(\frac{\mathbb{E}_{x'_{t'}}\left[\|u(x_{t'}) - u(x'_{t'})\|_2^2|x_t\right]}{\phi^2 d}\right)\Big| x_t\right]\right] \\
&\le \mathbb{E}_{x_t}\left[\mathbb{E}_{x_{t'},x'_{t'}}\left[\exp\left(\frac{\|u(x_{t'}) - u(x'_{t'})\|_2^2}{\phi^2 d}\right)\Big| x_t\right]\right] \\
&\le \mathbb{E}_{x_t}\left[\mathbb{E}_{x_{t'},x'_{t'}}\left[\exp\left(\frac{S^2\|x_{t'} - x'_{t'}\|_2^2}{\phi^2 d}\right)\Big| x_t\right]\right] \quad (9)
\end{aligned}
$$

Note that using (1), $x_t = e^{-\Delta}x_{t'} + w_{t,t'} = e^{-\Delta}x'_{t'} + w'_{t,t'}$, for $w_{t,t'}, w'_{t,t'} \sim \mathcal{N}\left(0, \sigma_{t-t'}^2 \mathbf{I}_d\right)$. Therefore, from (9),

$$
\begin{aligned}
\mathbb{E}_{x_{t'},x_t}\left[\exp\left(\frac{\|q_{t,t'}\|_2^2}{\phi^2 d}\right)\right] &\le \mathbb{E}_{x_t}\left[\mathbb{E}_{w_{t,t'},w'_{t,t'}}\left[\exp\left(\frac{S^2 e^{2\Delta}\|w_{t,t'} - w'_{t,t'}\|_2^2}{\phi^2 d}\right)\Big| x_t\right]\right] \\
&= \mathbb{E}_{w_{t'},w'_{t'}}\left[\exp\left(\frac{S^2 e^{2\Delta}\|w_{t,t'} - w'_{t,t'}\|_2^2}{\phi^2 d}\right)\right] \\
&\le \mathbb{E}_{w_{t'},w'_{t'}}\left[\exp\left(\frac{2S^2 e^{2\Delta}(\|w_{t,t'}\|_2^2 + \|w'_{t,t'}\|_2^2)}{\phi^2 d}\right)\right] \\
&\le \frac{1}{2}\mathbb{E}_{w_{t'},w'_{t'}}\left[\exp\left(\frac{4S^2 e^{2\Delta}\|w_{t,t'}\|_2^2}{\phi^2 d}\right)\right] + \frac{1}{2}\mathbb{E}_{w_{t'},w'_{t'}}\left[\exp\left(\frac{4S^2 e^{2\Delta}\|w'_{t,t'}\|_2^2}{\phi^2 d}\right)\right] \\
&\le 2
\end{aligned}
$$

where the last inequality follows since $w_{t,t'}, w'_{t,t'} \sim \mathcal{N}\left(0, \sigma_{t-t'}^2 \mathbf{I}_d\right)$ marginally (but not necessarily conditionally). $\square$

**Lemma 10.** *Fix $\delta > 0$. Let $t > t'$. Then, under Assumption 1-(1), with probability at least $1 - \delta$ over $x_t$,*

$$
\left\|e^{-(t-t')}s(t,x_t) - s(t', e^{(t-t')}x_t)\right\|_2 \le e^{(t-t')}L\sqrt{8d(t-t')\log\left(\frac{2}{\delta}\right)}
$$

*where $\sigma_{t-t'}^2 := 1 - e^{-2(t-t')} \le 2(t - t')$.*

*Proof.* Using Corollary 2.4 from [7],

$$
s(t, x_t) = e^{t-t'}\mathbb{E}[s(t', x'_t)|x_t] \quad (10)
$$

Using (1),

$$x_t = e^{-(t-t')}x_{t'} + z_{t,t'}, \text{ for } z_{t,t'} \sim \mathcal{N}\left(0, \sigma_{t-t'}^2 \mathbf{I}\right) \tag{11}$$

Where $z_{t,t'}$ is independent of $x_{t'}$. Let $y_{t,t'} := e^{-(t-t')}s(t, x_t) - s(t', e^{(t-t')}x_t)$. Then,

$$
\begin{aligned}
\|y_{t,t'}\| &= \|e^{-(t-t')}s_t(x_t) - s(t', e^{(t-t')}x_t)\| \\
&= \|\mathbb{E}\left[s\left(t', x_t'\right)|x_t\right] - s(t', e^{(t-t')}x_t)\| \\
&= \|\mathbb{E}\left[s_{t'}\left(e^{(t-t')}(x_t - z_{t,t'})\right) - s(t', e^{(t-t')}x_t)|x_t\right]\| \\
&\leq e^{t-t'}L\mathbb{E}\left[\|z_{t,t'}\|_2 |x_t\right]
\end{aligned}
$$

Note that since $z_{t,t'} \sim \mathcal{N}\left(0, \sigma_{t-t'}^2 \mathbf{I}\right)$,

$$\mathbb{E}\left[\exp\left(\frac{\|z_{t,t'}\|_2^2}{4\sigma_{t-t'}^2 d}\right)\right] \leq 2, \text{ using Lemma 8}$$

Therefore, with probability at least $1 - \delta$ over $x_t$:

$$\mathbb{E}\left[\exp\left(\frac{\|z_{t,t'}\|_2^2}{4\sigma_{t-t'}^2 d}\right)\Big|x_t\right] \leq \frac{2}{\delta}, \text{ using Markov's inequality}$$

Using Jensen's inequality,

$$\exp\left(\frac{\mathbb{E}\left[\|z_{t,t'}\|_2^2 |x_t\right]}{4\sigma_{t-t'}^2 d}\right) \leq \mathbb{E}\left[\exp\left(\frac{\|z_{t,t'}\|_2^2}{4\sigma_{t-t'}^2 d}\right)\Big|x_t\right] \leq \frac{2}{\delta}$$

The result then follows by taking log on both sides. $\qquad\square$

**Lemma 11.** *Let $w_{t,t'} := z_{t,t'} + \sigma_{t-t'}^2 s(t, x_t)$ for $t > t' \geq 0$. Then, $w_{t,t'}$ is $\nu_{t,t'}\sqrt{d}$ norm subGaussian for $\nu_{t,t'} := 4\sigma_{t-t'}$.*

*Proof.* Notice that $x_t = e^{t'-t}x_{t'} + z_{t,t'}$ Using Tweedie's formula, $s(t, x_t) = -\mathbb{E}\left[\frac{z_{t,t'}}{\sigma_{t-t'}^2}\Big|x_t\right]$. Therefore,

$$e^{t-t'}\sigma_{t-t'}^2 s_t(x_t) + e^{t-t'}x_t = \mathbb{E}[x_{t'}|x_t] \implies w_{t,t'} = -e^{t'-t}x_t + \mathbb{E}[e^{t'-t}x_{t'}|x_t]$$

Applying Lemma 9 with $u(x) = -e^{t'-t}x$ (which is $e^{t'-t}$ Lipschitz), we conclude the result. $\qquad\square$

**Lemma 12.** *Suppose Assumption 1-(1) holds. Let $v_{t,t'} := \mathbb{E}[x_0|x_t] - \mathbb{E}[x_0|x_{t'}]$ for $t > t' \geq 0$. Then, $v_{t,t'}$ is $\rho_{t,t'}\sqrt{d}$ norm subGaussian for*

$$\rho_{t,t'} := 8(L+1)e^t\sigma_{t-t'}$$

*Proof.* Using Tweedie's formula, for all $t > 0$,

$$\mathbb{E}[x_0|x_t] = \mathbb{E}\left[e^t(x_t - z_t)|x_t\right] = e^t x_t + e^t\mathbb{E}[-z_t|x_t] = e^t\left(x_t + \sigma_t^2 s(t, x_t)\right)$$

Note that $x_{t'} = e^{t-t'}(x_t - z_{t,t'})$. Furthermore, note that

$$\mathbb{E}\left[z_{t,t'}|x_t\right] = -\sigma_{t-t'}^2 s(t, x_t), \quad \mathbb{E}\left[s_{t'}(x_{t'})|x_t\right] = e^{-(t-t')}s(t, x_t)$$

Therefore, we have

$$v_{t,t'} = \underbrace{e^t\left(z_{t,t'} + \sigma_{t-t'}^2 s(t, x_t)\right)}_{:=T_1} - \underbrace{e^{t'}\sigma_{t'}^2\left(s_{t'}(x_{t'}) - e^{-(t-t')}s(t, x_t)\right)}_{:=T_2}$$

Using Lemma 11, $T_1$ is $4e^t\sigma_{t-t'}\sqrt{d}$ norm subGaussian. Using Lemma 9, $T_2$ is $4Le^{t-t'}e^{t'}\sigma_{t'}^2\sigma_{t-t'}\sqrt{d} = 4Le^t\sigma_{t'}^2\sigma_{t-t'}\sqrt{d}$ norm subGaussian. Therefore, the result follows using the sum of subGaussian random variables. $\qquad\square$

**Lemma 13.** *Let $\Delta > 0$ and $\Delta < c_0$ for some universal constant $c_0$. Then,*

1. $\sum_{k=1}^{N} \sum_{j=k}^{N} \frac{e^{2(k-j)\Delta}}{(1-e^{-2\Delta j})^2} \leq \frac{1}{1-e^{-2\Delta}} \left( N + \frac{1}{1-e^{-2\Delta}} \right)$

2. $\sum_{j=1}^{N} \frac{e^{-2\Delta(j-1)}}{(1-e^{-2\Delta j})^2} \leq \frac{2}{(1-e^{-2\Delta})^2}$

3. $\sum_{j=1}^{N} \frac{e^{-\Delta(j-1)}}{1-e^{-2\Delta j}} \leq \frac{e^{-\Delta}}{1-e^{-2\Delta}} + \frac{\log(\frac{1}{\Delta})}{2\Delta}$

*Proof.* Let us start with the first bound. We have,

$$\sum_{k=1}^{N} \sum_{j=k}^{N} \frac{e^{2(k-j)\Delta}}{(1-e^{-2\Delta j})^2} = \sum_{j=1}^{N} \sum_{k=1}^{j} \frac{e^{2(k-j)\Delta}}{(1-e^{-2\Delta j})^2}$$

$$= \sum_{j=1}^{N} \frac{1}{(1-e^{-2\Delta j})^2} \sum_{k=1}^{j} e^{2(k-j)\Delta}$$

$$= \sum_{j=1}^{N} \frac{1}{(1-e^{-2\Delta j})^2} \frac{e^{2\Delta}}{e^{2\Delta}-1} \left(1 - e^{-2\Delta j}\right)$$

$$= \frac{e^{2\Delta}}{e^{2\Delta}-1} \sum_{j=1}^{N} \frac{1}{1-e^{-2\Delta j}}$$

Consider the function $f(x) := \frac{1}{1-e^{-2\Delta x}}$. Then, $f(x)$ is positive, convex and decreasing. Therefore,

$$\sum_{j=1}^{N} \frac{1}{1-e^{-2\Delta j}} \leq f(1) + \int_{1}^{N} \frac{1}{1-e^{-2\Delta x}} dx$$

$$= \frac{1}{1-e^{-2\Delta}} + \frac{1}{2\Delta} \ln \left(e^{2\Delta x}-1\right) \Big|_{1}^{N}$$

$$\leq \frac{1}{1-e^{-2\Delta}} + \frac{1}{2\Delta} \ln \left(e^{2\Delta N}-1\right)$$

$$\leq N + \frac{1}{1-e^{-2\Delta}}$$

which completes the first result. Now for the second result,

$$\sum_{j=1}^{N} \frac{e^{-2\Delta(j-1)}}{(1-e^{-2\Delta j})^2} = e^{2\Delta} \sum_{j=1}^{N} \frac{e^{-2\Delta j}}{(1-e^{-2\Delta j})^2}$$

Consider the function, $g(x) := \frac{e^{-2\Delta x}}{(1-e^{-2\Delta x})^2}$. For $x > 0$, $g(x)$ is a positive, decreasing and convex function. Therefore,

$$\sum_{j=1}^{N} \frac{e^{-2\Delta j}}{(1-e^{-2\Delta j})^2} \leq g(1) + \int_{1}^{N} g(x)\, dx$$

$$= \frac{e^{-2\Delta}}{(1-e^{-2\Delta})^2} + \int_{1}^{N} \frac{e^{-2\Delta x}}{(1-e^{-2\Delta x})^2} dx$$

$$= \frac{e^{-2\Delta}}{(1-e^{-2\Delta})^2} + \frac{1}{2\Delta(1-e^{-2\Delta x})} \Big|_{1}^{N}$$

$$\leq \frac{e^{-2\Delta}}{(1-e^{-2\Delta})^2} + \frac{1}{2\Delta(1-e^{-2\Delta N})}$$

$$\leq \frac{2e^{-2\Delta}}{(1-e^{-2\Delta})^2}$$

which completes the proof. Finally for the third result, consider the function $h(x) := \frac{e^{-\Delta x}}{1-e^{-2\Delta x}}$. For $x > 0$, $h(x)$ is a positive, decreasing and convex function. Therefore,

$$
\begin{aligned}
\sum_{j=1}^{N} \frac{e^{-\Delta j}}{1 - e^{-2\Delta j}} &\leq h(1) + \int_1^N h(x)\, dx \\
&= \frac{e^{-\Delta}}{1 - e^{-2\Delta}} + \int_1^N \frac{e^{-\Delta x}}{1 - e^{-2\Delta x}} dx \\
&= \frac{e^{-\Delta}}{1 - e^{-2\Delta}} + \frac{1}{2\Delta} \log\left(\tanh\left(\Delta x\right)\right) \Big|_1^N \\
&\leq \frac{e^{-\Delta}}{1 - e^{-2\Delta}} - \frac{\log\left(\tanh\left(\Delta\right)\right)}{2\Delta} \\
&\leq \frac{e^{-\Delta}}{1 - e^{-2\Delta}} - \frac{\log\left(1 - e^{-2\Delta}\right)}{2\Delta} \\
&\leq \frac{e^{-\Delta}}{1 - e^{-2\Delta}} + \frac{\log(\frac{1}{\Delta})}{2\Delta}
\end{aligned}
$$

$\square$

## B   Martingale Concentration

**Lemma 14.** *Let $Y$ be a $\left(\beta^2, K\right)$-subGaussian random variable following definition 1, with $(K \geq 1)$. Then, for any integer $k > 0$ and some universal constant $C > 0$:*

$$
\mathbb{E}\left[Y^{2k}\right] \leq C^k K^k \beta^{2k} + C^k k! \beta^{2k}
$$

*Proof.* By Definition 1, for any $A > 0$,

$$
\mathbb{P}(|Y| > A) \leq e^K \exp\left(-\frac{A^2}{2\beta^2}\right).
$$

Using the tail-integration representation of moments, we have

$$
\mathbb{E}[|Y|^{2k}] = \int_0^\infty \mathbb{P}\left(|Y|^{2k} > t\right) dt = \int_0^\infty \mathbb{P}\left(|Y| > t^{1/(2k)}\right) dt.
$$

Make the change of variables $t = x^{2k}$ so that $dt = 2k\, x^{2k-1} dx$. Then

$$
\mathbb{E}[|Y|^{2k}] = \int_0^\infty 2k\, x^{2k-1}\, \mathbb{P}(|Y| > x)\, dx \leq 2k \int_0^\infty x^{2k-1}\, \min(1, e^K \exp\left(-\frac{x^2}{2\beta^2}\right))\, dx.
$$

Let $x_0 = \sqrt{2\beta^2 K}$

$$
\begin{aligned}
\mathbb{E}[|Y|^{2k}] &\leq 2k \int_0^{x_0} x^{2k-1} dx + 2k \int_{x_0}^\infty x^{2k-1} e^K e^{-\frac{x^2}{2\beta^2}} dx \\
&= (2\beta^2 K)^k + 2k \int_{x_0}^\infty x^{2k-1} e^K e^{-\frac{x^2}{2\beta^2}} dx \\
&\leq (2\beta^2 K)^k + 2k \int_{x_0}^\infty x^{2k-1} e^{-\frac{(x-x_0)^2}{2\beta^2}} dx \\
&\leq (2\beta^2 K)^k + 2^{2k-1} k \int_{x_0}^\infty (x_0^{2k-1} + (x - x_0)^{2k-1}) e^{-\frac{(x-x_0)^2}{2\beta^2}} dx
\end{aligned}
$$

In the second step we have used the fact that whenever $x \geq x_0$, we must have $K - \frac{x^2}{2\beta^2} \leq -\frac{(x-x_0)^2}{2\beta^2}$.
In the third step we have used the fact that $x^{2k-1} \leq 2^{2k-2}[(x - x_0)^{2k-1} + x_0^{2k-1}]$ whenever $x \geq x_0$.

A standard Gamma-function integral yields

$$\int_0^\infty x^{2k-1} \exp\left(-\frac{x^2}{2\beta^2}\right) dx = \frac{1}{2}(2\beta^2)^k \Gamma(k),$$

and for integer $k$, $\Gamma(k) = (k-1)!$. Substituting this to the equation above, we conclude that for some universal constant $C_1$, we have:

$$\mathbb{E}[|Y|^{2k}] \leq (2\beta^2 K)^k + C_1^k(k\beta^{2k}K^{k-1/2} + \beta^{2k}k!)$$

We then conclude the result using the fact that $K \geq 1$ and $k \leq 2^k$.

$\square$

**Lemma 15.** *Let $Y$ be a $(\beta^2, K)$-subGaussian random variable following definition 1, such that $K \geq 1$, $\mathbb{E}[Y] = 0$ and $\mathbb{E}[Y^2] \leq \nu^2$. Then, for a sufficiently small universal constant $c_0 > 0$ such that, $\lambda\beta \leq c_0$, and any arbitrary $A > 0$, we have:*

$$\mathbb{E}\exp(\lambda^2 Y^2) \leq 1 + \lambda^2\nu^2\exp(\lambda^2 A^2) + C\lambda^4\beta^4 K^2 \exp(\tfrac{K}{2} - \tfrac{A^2}{4\beta^2} + C\lambda^2\beta^2 K)$$

*Proof.* For some $\lambda > 0$, consider:

$$\mathbb{E}\left[\exp(\lambda^2 Y^2)\right] = 1 + \lambda^2\nu^2 + \sum_{k \geq 2} \frac{\lambda^{2k}\mathbb{E}\left[Y^{2k}\right]}{k!} \tag{12}$$

Now, using Lemma 14, consider

$$
\begin{aligned}
\mathbb{E}\left[Y^{2k}\right] &= \mathbb{E}\left[Y^{2k}\mathbb{1}(|Y| > A)\right] + \mathbb{E}\left[Y^{2k}\mathbb{1}(|Y| \leq A)\right] \\
&\leq \sqrt{\mathbb{E}[Y^{4k}]}\sqrt{\mathbb{P}(|Y| > A)} + \mathbb{E}\left[Y^2\right]A^{2k-2} \\
&= \sqrt{\mathbb{E}[Y^{4k}]}\sqrt{\mathbb{P}(|Y| > A)} + \nu^2 A^{2k-2} \\
&\leq \sqrt{C^{2k}\beta^{4k}(2k)! + C^{2k}\beta^{4k}K^{2k}}\exp(\tfrac{K}{2} - \tfrac{A^2}{4\beta^2}) + \nu^2 A^{2k-2} \\
&\leq \left((2C)^k k!\beta^{2k} + C^k\beta^{2k}K^k\right)\exp(\tfrac{K}{2} - \tfrac{A^2}{4\beta^2}) + \nu^2 A^{2k-2} \tag{13}
\end{aligned}
$$

Here, we have used the fact that $(2k)! \leq 4^k(k!)^2$. Plugging this back in Equation (12), we conclude that whenever $\lambda\beta \leq c_0$ for some small enough constant $c_0$, we have:

$$\mathbb{E}\left[\exp(\lambda^2 Y^2)\right] \leq 1 + \lambda^2\nu^2\exp(\lambda^2 A^2) + C\lambda^4\beta^4 K^2 \exp(\tfrac{K}{2} - \tfrac{A^2}{4\beta^2} + C\lambda^2\beta^2 K) \tag{14}$$

$\square$

**Theorem 4.** *Let $Y$ be a $(\beta^2, K)$-subGaussian random variable following definition 1, such that $K \geq 1$, $\mathbb{E}[Y] = 0$ and $\mathbb{E}[Y^2] \leq \nu^2$. Set $A \geq \beta\sqrt{4\log(\frac{\beta K}{\nu})} + \beta\sqrt{2K}$ and $\lambda \leq \frac{c_0}{A}$ for some small enough constant $c_0 > 0$. Then, there exists a constant $C$ such that:*

$$\mathbb{E}\left[\exp(\lambda^2 Y^2)\right] \leq 1 + C\lambda^2\nu^2$$

*Proof.* The result follows from Lemma 15 substituting the values of $\lambda$ and $A$. $\square$

**Lemma 16.** *Let $M_n = \sum_{i=1}^n \langle G_i, Y_i - \mathbb{E}[Y_i|\mathcal{F}_{i-1}]\rangle$, $M_0 = 1$ and define the filtration $\{\mathcal{F}_i\}_{i\in[n]}$ such that:*

1. *$G_i$ is $\mathcal{F}_{i-1}$ measurable.*

2. *$\langle G_i, Y_i - \mathbb{E}[Y_i|\mathcal{F}_{i-1}]\rangle$ is $(\beta_i^2\|G_i\|^2, K_i)$ sub-Gaussian conditioned on $\mathcal{F}_{i-1}$ (where $\beta_i, K_i$ are random variables measurable with respect to $\mathcal{F}_{i-1}$)*

3. $\mathsf{var}(\langle G_i, Y_i - \mathbb{E}[Y_i|\mathcal{F}_{i-1}]\rangle|\mathcal{F}_{i-1}) \leq \nu_i^2\|G_i\|^2$ *and define* $J_i := \max(1, \frac{1}{K_i}\log\frac{\beta_i^2 K_i}{\nu_i^2})$.

*Pick a* $\lambda > 0$ *and let* $\mathcal{A}_i(\lambda) = \{\lambda J_i\|G_i\|\beta_i\sqrt{K_i} \leq c_0\}$ *for some small enough universal constant* $c_0$. *Then, there exists a universal constant* $C > 0$ *such that:*

1. $\exp(\lambda M_n - C\lambda^2\sum_{i=1}^n \nu_i^2\|G_i\|^2)\prod_{i=1}^n \mathbb{1}(\mathcal{A}_i(\lambda))$ *is a super-martingale with respect to the filtration* $\mathcal{F}_i$

2. $\forall \alpha > 0, \ \mathbb{P}(\{\lambda M_n > C\lambda^2\sum_{i=1}^n\nu_i^2\|G_i\|^2 + \alpha\} \cap_{i=1}^n \mathcal{A}_i(\lambda)) \leq \exp(-\alpha)$

*Proof.* Let $L_n := \exp(\lambda M_n - C\lambda^2\sum_{i=1}^n\nu_i^2\|G_i\|^2)\prod_{i=1}^n \mathbb{1}(\mathcal{A}_i(\lambda))$. Then we have,

$$\mathbb{E}\left[L_n\Big|\mathcal{F}_{n-1}\right] = L_{n-1}\mathbb{E}\left[\exp\left(\lambda\langle G_n, Y_n - \mathbb{E}[Y_n|\mathcal{F}_{n-1}]\rangle - C\lambda^2\nu_n^2\|G_n\|^2\right)\mathbb{1}\left(\mathcal{A}_n(\lambda)\right)\Big|\mathcal{F}_{n-1}\right]$$

$$= L_{n-1}\exp\left(-C\lambda^2\nu_n^2\|G_n\|^2\right)\mathbb{E}\left[\exp\left(\lambda\langle G_n, Y_n - \mathbb{E}[Y_n|\mathcal{F}_{n-1}]\rangle\right)\mathbb{1}\left(\{J_n\lambda\|G_n\|\beta_n\sqrt{K_n} \leq c_0\}\right)\Big|\mathcal{F}_{n-1}\right]$$

$$\leq L_{n-1}\exp\left(-C\lambda^2\nu_n^2\|G_n\|^2\right)\exp\left(C\lambda^2\nu_n^2\|G_n\|^2\right) \text{ using Theorem 4 and the definition of } \mathcal{A}_n(\lambda)$$

$$\leq L_{n-1}$$

The second result follows from a standard Chernoff bound argument. $\qquad\square$

**Lemma 17.** *Under the setting of Lemma 16, let* $\lambda^* := \sqrt{\frac{\alpha}{\sum_{i=1}^n\nu_i^2\|G_i\|^2}}$ *and* $\lambda_{\min} := \frac{c_0}{\sup_i J_i\|G_i\|\beta_i\sqrt{K_i}}$. *Let* $B \in \mathbb{N}$ *be arbitrary and consider the event:* $\mathcal{B} = \{e^{-B} \leq \min(\lambda^*, \lambda_{\min}) \leq \max(\lambda^*, \lambda_{\min}) \leq e^B\}$. *Then, for some universal constant* $C_1 > 0$ *and any* $\alpha > 0$,

$$\mathbb{P}\left(\{M_n > C_1\lambda^*\sum_{i=1}^n\nu_i^2\|G_i\|^2 + C_1\tfrac{\alpha}{\lambda_{\min}}\} \cap \mathcal{B}\right) \leq (2B+1)e^{-\alpha}$$

*Proof.* We apply union bound over $\lambda \in \Lambda_B := \{e^{-B}, e^{-B+1}, \ldots, e^B\}$. Using Lemma 16 along with a union bound,

$$\mathbb{P}(\cup_{\lambda\in\Lambda_B}\{\lambda M_n > C\lambda^2\sum_{i=1}^n\nu_i^2\|G_i\|^2 + \alpha\} \cap_{i=1}^n \mathcal{A}_i(\lambda)) \leq (2B+1)\exp(-\alpha)$$

Consider the following events:

1. Event 1: $\mathcal{E}_1 := \{\max(\lambda^*, \lambda_{\min}) > e^B\}$

2. Event 2: $\mathcal{E}_2 := \{\min(\lambda^*, \lambda_{\min}) < e^{-B}\}$

3. Event 3: $\mathcal{E}_3 := \{e^{-B} \leq \lambda^* < \lambda_{\min} \leq e^B\}$

4. Event 4: $\mathcal{E}_4 := \{e^{-B} \leq \lambda_{\min} < \lambda^* \leq e^B\}$

In the event $\mathcal{E}_4$, almost surely there exists a random $\bar{\lambda} \in \Lambda_B$ such that $\bar{\lambda}/\lambda_{\min} \in [\frac{1}{e}, e]$ and such that the event $\cap_{i=1}^n \mathcal{A}_i(\bar{\lambda})$ holds. Thus, we have:

$$\{M_n > Ce\lambda^*\sum_i\nu_i^2\|G_i\|^2 + \frac{e\alpha}{\lambda_{\min}}\} \cap \mathcal{E}_4 \subseteq \{M_n > Ce\lambda_{\min}\sum_i\nu_i^2\|G_i\|^2 + \frac{e\alpha}{\lambda_{\min}}\} \cap \mathcal{E}_4$$

$$\subseteq \{M_n > C\bar{\lambda}\sum_i\nu_i^2\|G_i\|^2 + \frac{\alpha}{\bar{\lambda}}\} \cap \mathcal{E}_4 = \{M_n > C\bar{\lambda}\sum_i\nu_i^2\|G_i\|^2 + \frac{\alpha}{\bar{\lambda}}\} \cap \mathcal{E}_4 \cap_{i=1}^n \mathcal{A}_i(\bar{\lambda})$$

$$\subseteq \mathcal{E}_4 \cap \left(\cup_{\lambda\in\Lambda_B}\{\lambda M_n > C\lambda^2\sum_{i=1}^n\nu_i^2\|G_i\|^2 + \alpha\} \cap_{i=1}^n \mathcal{A}_i(\lambda)\right) \tag{15}$$

Similarly, under the event $\mathcal{E}_3$, there exists a random $\bar{\lambda}^* \in \Lambda_B$ such that: $\bar{\lambda}^*/\lambda^* \in [\frac{1}{e}, e]$, such that the event $\cap_i \mathcal{A}_i(\bar{\lambda}^*)$ holds. Therefore, we must have:

$$\{M_n > Ce\lambda^* \sum_i \nu_i^2 \|G_i\|^2 + \frac{e\alpha}{\lambda^*}\} \cap \mathcal{E}_3 \subseteq \{M_n > C\bar{\lambda}^* \sum_i \nu_i^2 \|G_i\|^2 + \frac{\alpha}{\bar{\lambda}^*}\} \cap \mathcal{E}_3$$

$$= \{M_n > C\bar{\lambda}^* \sum_i \nu_i^2 \|G_i\|^2 + \frac{\alpha}{\bar{\lambda}^*}\} \cap \mathcal{E}_3 \cap_{i=1}^n \mathcal{A}_i(\bar{\lambda}^*)$$

$$\subseteq \mathcal{E}_3 \cap \left( \cup_{\lambda \in \Lambda_B} \{\lambda M_n > C\lambda^2 \sum_{i=1}^n \nu_i^2 \|G_i\|^2 + \alpha\} \cap_{i=1}^n \mathcal{A}_i(\lambda) \right) \tag{16}$$

Notice that $\lambda^*$ is chosen such that

$$Ce\lambda^* \sum_i \nu_i^2 \|G_i\|^2 + \frac{e\alpha}{\lambda^*} = e(C+1)\sqrt{\alpha(\sum_i \nu_i^2 \|G_i\|^2)}$$

$$= e(C+1)\lambda^* \sum_i \nu_i^2 \|G_i\|^2 \tag{17}$$

$$\leq e(C+1)\lambda^* \sum_i \nu_i^2 \|G_i\|^2 + \frac{e\alpha}{\lambda_{\min}} \tag{18}$$

Combining these equations, we conclude that for some constant $C_1 > 0$, we must have

$$\{M_n > C_1(\lambda^* \sum_{i=1}^n \nu_i^2 \|G_i\|^2 + \frac{\alpha}{\lambda_{\min}})\} \cap (\mathcal{E}_3 \cup \mathcal{E}_4) \subseteq \left( \cup_{\lambda \in \Lambda_B} \{\lambda M_n > C\lambda^2 \sum_{i=1}^n \nu_i^2 \|G_i\|^2 + \alpha\} \cap_{i=1}^n \mathcal{A}_i(\lambda) \right) \cap (\mathcal{E}_3 \cup \mathcal{E}_4)$$

Noting that $\mathcal{B} = \mathcal{E}_3 \cup \mathcal{E}_4$, we conclude the result.

$$\square$$

## C  Martingale Decomposition and Variance Calculation

In this section, we will consider the quantity similar to $H^f$ in Lemma 35, decompose it into a sum of martingale difference sequence, and then bounds its variance using the Tweedie's formula. In this section, assume that we are given $\zeta : \mathbb{R}^+ \times \mathbb{R}^d \to \mathbb{R}^d$ and consider the quantity:

$$H := \sum_{t \in \mathcal{T}, i \in [m]} \frac{\gamma_t}{\sigma_t^2} \langle \zeta(t, x_t^{(i)}), z_t^{(i)} - \mathbb{E}[z_t^{(i)}|x_t^{(i)}] \rangle$$

We suppose that $\zeta(t, x_t^{(i)})$ has a finite second moment. Where $\gamma_t > 0$ is some sequence. When $\zeta = \frac{s-f}{m}$, this yields us $H^f$ as we show in Lemma 19. We define the sigma algebras: $\sigma$-algebra $\mathcal{F}_j = \sigma(x_t^{(i)} : 1 \leq i \leq m, t \geq t_{N-j+1})$ for $j \in [N]$ and $\mathcal{F}_0$ is the trivial $\sigma$-algebra. We want to filter $H$ through the filtration $\mathcal{F}_j$ to obtain a martingale decomposition. To this end, define:

$$H_j := \mathbb{E}[H|\mathcal{F}_j] \; ; j \in \{0, \dots, N\} \tag{19}$$

**Lemma 18.**    *1. If $t \leq t_{N-j+1}$, then*

$$\mathbb{E}[\langle \zeta(t, x_t^{(i)}), z_t^{(i)} - \mathbb{E}[z_t^{(i)}|x_t^{(i)}] \rangle | \mathcal{F}_j] = 0$$

2. *If $t > t_{N-j+1}$, then*

$$\mathbb{E}[\langle \zeta(t, x_t^{(i)}), z_t^{(i)} - \mathbb{E}[z_t^{(i)}|x_t^{(i)}] \rangle | \mathcal{F}_j] = e^{-t} \langle \zeta(t, x_t^{(i)}), \mathbb{E}[x_0^{(i)}|x_t^{(i)}] - \mathbb{E}[x_0^{(i)}|x_{t_{N-j+1}}^{(i)}] \rangle$$

*Proof.* 1. Using the fact that $x_t^{(i)}$ forms a Markov process and that $(x_s^{(i)})_{s\geq 0}$, $(x_s^{(j)})_{s\geq 0}$ are independent when $i \neq j$, we have via the Markov property:

$$\mathbb{E}[\langle \zeta(t, x_t^{(i)}), z_t^{(i)} - \mathbb{E}[z_t^{(i)}|x_t^{(i)}]\rangle | \mathcal{F}_j] = \mathbb{E}[\langle \zeta(t, x_t^{(i)}), z_t^{(i)} - \mathbb{E}[z_t^{(i)}|x_t^{(i)}]\rangle | x_{t_{N-j+1}}^{(i)}]$$
$$= \mathbb{E}\left[\mathbb{E}[\langle \zeta(t, x_t^{(i)}), z_t^{(i)} - \mathbb{E}[z_t^{(i)}|x_t^{(i)}]\rangle | x_t^{(i)}, x_{t_{N-j+1}}^{(i)}]\big| x_{t_{N-j+1}}^{(i)}\right] \tag{20}$$

In the second step, we have used the tower property of the conditional expectation. Now, $z_t^{(i)} = x_t^{(i)} - e^{-t}x_0^{(i)}$. By the Markov Property, we have: $\mathbb{E}[x_0^{(i)}|x_t^{(i)}, x_{t_{j-N+1}}^{(i)}] = \mathbb{E}[x_0^{(i)}|x_t^{(i)}]$. Plugging this in, we have:

$$\mathbb{E}[\langle \zeta(t, x_t^{(i)}), z_t^{(i)} - \mathbb{E}[z_t^{(i)}|x_t^{(i)}]\rangle | \mathcal{F}_j] = \mathbb{E}\left[\mathbb{E}[\langle \zeta(t, x_t^{(i)}), z_t^{(i)} - \mathbb{E}[z_t^{(i)}|x_t^{(i)}]\rangle | x_t^{(i)}]\big| x_{t_{N-j+1}}^{(i)}\right]$$
$$= 0 \tag{21}$$

2. Notice that $z_t^{(i)} = x_t^{(i)} - e^{-t}x_0^{(i)}$. Clearly, $x_t^{(i)}$ is measurable with respect to $\mathcal{F}_j$. Therefore,

$$\mathbb{E}[\langle \zeta(t, x_t^{(i)}), z_t^{(i)} - \mathbb{E}[z_t^{(i)}|x_t^{(i)}]\rangle | \mathcal{F}_j] = -e^{-t}\langle \zeta(t, x_t^{(i)}), \mathbb{E}[x_0^{(i)}|\mathcal{F}_j] - \mathbb{E}[x_0^{(i)}|x_t^{(i)}]\rangle$$

Now, consider the fact that $x_0^{(i)}, x_{t_1}^{(i)}, \dots$ is a Markov chain. Therefore, the Markov property states that $x_0^{(i)}|x_s^{(i)} : s \geq \tau$ has the same law as $x_0^{(i)}|x_\tau^{(i)}$. Therefore, we must have: $\mathbb{E}[x_0^{(i)}|\mathcal{F}_j] = \mathbb{E}[x_0^{(i)}|x_{t_{j-N+1}}^{(i)}]$. Plugging this into the display equation above, we conclude the result.

$\square$

We connect the quantity $H$ defined above to the quantity $H^f$ related to the excess risk.

**Lemma 19.** *Let* $y_t^{(i)} := -\frac{z_t^{(i)}}{\sigma_t^2}$, $f \in \mathcal{H}$ *and*

$$H^f := \sum_{i\in[m], j\in[N]} \frac{\gamma_j \left\langle f\left(t_j, x_{t_j}^{(i)}\right) - s\left(t_j, x_{t_j}^{(i)}\right), y_{t_j}^{(i)} - s\left(t_j, x_{t_j}^{(i)}\right)\right\rangle}{m}$$

*Suppose we pick $\zeta = \frac{s-f}{m}$ in the definition of $H$. Then,*

$$H^f = (H - H_N) + \sum_{k=2}^{N} (H_k - H_{k-1})$$

*such that*

$$H - H_N = \sum_{i=1}^{m} \sum_{j=1}^{N} \frac{e^{-(t_j-t_1)}\gamma_j}{\sigma_{t_j}^2} \langle \zeta(t_j, x_{t_j}^{(i)}), z_{t_1}^{(i)} - \mathbb{E}[z_{t_1}^{(i)}|x_{t_1}^{(i)}]\rangle$$

$$H_k - H_{k-1} = \sum_{i=1}^{m} \sum_{j=N-k+2}^{N} \frac{e^{-t_j}\gamma_j}{\sigma_{t_j}^2} \langle \zeta(t_j, x_{t_j}^{(i)}), \mathbb{E}[x_0^{(i)}|x_{t_{N-k+2}}^{(i)}] - \mathbb{E}[x_0^{(i)}|x_{t_{N-k+1}}^{(i)}]\rangle$$

*Proof.* By Tweedie's formula, notice that $y_t^i - s(t, x_t^{(i)}) = \frac{\mathbb{E}[z_t^{(i)}|x_t^{(i)}] - z_t^{(i)}}{\sigma_t^2}$. This shows us that $H^f = H$ when we pick $\zeta = \frac{s-f}{m}$. The proof follows due to Lemma 18 once we note that $H_1 = 0$ almost surely $\square$

**Lemma 20.** *Define* $\bar{G}_i := \sum_{j=1}^{N} \frac{\gamma_j e^{-(t_j-t_1)}\zeta\left(t_j, x_{t_j}^{(i)}\right)}{\sigma_{t_j}^2}$, $G_{i,k} := \sum_{j=N-k+2}^{N} \frac{\gamma_j e^{-t_j}\zeta\left(t_j, x_{t_j}^{(i)}\right)}{\sigma_{t_j}^2}$ *and* $R_{i,k}$ *as*

$$R_{i,k} = \begin{cases} 0 & \text{for } k = 0 \\ \langle G_{i,k+1}, \mathbb{E}[x_0^{(i)}|x_{t_{N-k+1}}^{(i)}] - \mathbb{E}[x_0^{(i)}|x_{t_{N-k}}^{(i)}]\rangle & \text{for } k \in \{1, \dots, N-1\}, \\ \langle \bar{G}_i, z_{t_1}^{(i)} - \mathbb{E}\left[z_{t_1}^{(i)}|x_{t_1}^{(i)}\right]\rangle & \text{for } k = N \end{cases} \tag{22}$$

*Let $t_0 = 0$. Consider the filtration defined by the sequence of $\sigma$-algebras, $\mathcal{F}_{i,k} := \sigma(\{x_t^{(j)} : 1 \le j < i, t \in \mathcal{T}\} \cup \{x_t^{(i)} : t \ge t_{N-k}\})$ for $i \in [m]$ and $k \in \{0, \dots, N\}$, satisfying the total ordering $\{(i_1, j_1) < (i_2, j_2)$ iff $i_1 < i_2$ or $i_1 = i_2, j_1 < j_2\}$. Then*

1. *For $k \in [N-1]$, $G_{i,k+1}$ is measurable with respect to $\mathcal{F}_{i,k-1}$ and $\bar{G}_i$ if $\mathcal{F}_{N-1}$ measurable.*

2. *For $i \in [m], k \in \{0\} \cup [N]$, $(R_{i,k})_{i,k}$ forms a martingale difference sequence with respect to the filtration above.*

3. *$H = \sum_{i \in [m]} \sum_{k \in [N]} R_{i,k}$ .*

*Proof.*    1. We first note that for $1 \le k \le N-1$, $\sigma\left\{x_t^i : t \ge t_{N-k+1}\right\} \subseteq \mathcal{F}_{i,k-1}$. Therefore, $G_{i,k+1}$ is measurable with respect to $\mathcal{F}_{i,k-1}$. Furthermore, if $k = N$, then $\bar{G}_i$ is measurable with respect to $\mathcal{F}_{i,k-1}$.

2. First note that $R_{i,k}$ is $\mathcal{F}_{i,k}$ measurable.

$$\mathbb{E}\left[R_{i,k}|\mathcal{F}_{i,k-1}\right] = \begin{cases} \left\langle G_{i,k+1}, \mathbb{E}[x_0^{(i)}|x_{t_{N-k+1}}^{(i)}] - \mathbb{E}\left[\mathbb{E}[x_0^{(i)}|x_{t_{N-k}}^{(i)}]|\mathcal{F}_{i,k-1}\right]\right\rangle = 0, & \text{when } k \in [N-1], \\[2mm] \left\langle \bar{G}_i, \mathbb{E}\left[z_{t_1}^{(i)}|\mathcal{F}_{i,k-1}\right] - \mathbb{E}\left[z_{t_1}^{(i)}|x_{t_1}^{(i)}\right]\right\rangle = 0, & \text{when } k = N \end{cases}$$

The case of $R_{i,0}$ is straightforward.

3. This follows from Lemma 19.

$\square$

**Lemma 21.** *Consider the setting of Lemma 20. Define:*

$$V_{i,k} = \begin{cases} 0 & \text{if } k = 0 \\ \mathbb{E}[x_0^{(i)}|x_{t_{N-k+1}}^{(i)}] - \mathbb{E}[x_0^{(i)}|x_{t_{N-k}}^{(i)}] & \text{if } k \in \{1, \dots, N-1\} \\ z_{t_1}^{(i)} - \mathbb{E}\left[z_{t_1}^{(i)}|x_{t_1}^{(i)}\right] & \text{if } k = N \end{cases} \tag{23}$$

*Let $\Sigma_{i,k} := \mathbb{E}[V_{i,k}V_{i,k}^\top|\mathcal{F}_{i,k-1}]$. Then, we have:*

$$\mathbb{E}\left[R_{i,k}^2|\mathcal{F}_{i,k-1}\right] = \begin{cases} 0 & \text{if } k = 0 \\ G_{i,k+1}^\top \Sigma_{i,k} G_{i,k+1} & \text{if } k \in \{1, \dots, N-1\} \\ \bar{G}_i^\top \Sigma_{i,k} \bar{G}_i & \text{if } k = N \end{cases} \tag{24}$$

*Proof.* This follows from a straightforward application of Lemma 20. $\square$

Let $U$ be any random vector over $\mathbb{R}^d$ independent of $V \sim \mathcal{N}(0, \sigma^2 \mathbf{I}_d)$. Let $W = U + V$ and let $p$ be the density of $W$, $s = \nabla \log p$ and $h = \nabla^2 \log p$. Then, second order Tweedie's formula states (Theorem 1,[31]):
$$\mathbb{E}[VV^\top|W] = \sigma^4 h(W) + \sigma^4 s(W)s^\top(W) + \sigma^2 \mathbb{I}_d \,.$$

**Lemma 22.** *Let $s_\tau : \mathbb{R}^d \to \mathbb{R}^d$ be continuously differentiable for every $\tau > 0$. Let $t' < t$ and $x_t = e^{-(t-t')}x_{t'} + z_{t,t'}$ where $z_{t,t'} \sim \mathcal{N}\left(0, \sigma_{t-t'}^2 \mathbf{I}_d\right)$, as defined in Section 2. Then,*

$$\mathbb{E}\left[z_{t,t'}z_{t,t'}^\top|x_t\right] = \sigma_{t-t'}^4 h_t(x_t) + \sigma_{t-t'}^4 s(t, x_t) s(t, x_t)^\top + \sigma_{t-t'}^2 \mathbf{I}_d$$

$$\mathbb{E}\left[s(t', x_{t'}) s(t', x_{t'})^\top |x_t\right] = e^{2(t'-t)}s(t, x_t)s(t, x_t)^\top + e^{2(t'-t)}h_t(x_t) - \mathbb{E}[h_{t'}(x_{t'})|x_t]$$

*where $h_t(x_t) := \nabla^2 \log(p_t(x_t))$.*

*Proof.* Applying second order Tweedie's formula:

$$\mathbb{E}\left[z_{t,t'}z_{t,t'}^\top | x_t\right] = \sigma_{t-t'}^4 h_t\left(x_t\right) + \sigma_{t-t'}^4 s\left(t, x_t\right) s\left(t, x_t\right)^\top + \sigma_{t-t'}^2 \mathbf{I}_d, \text{ and },\tag{25}$$

$$\mathbb{E}[z_{t'}z_{t'}^\top | x_{t'}] - \sigma_{t'}^4 s(t', x_{t'})s^\top(t', x_{t'}) = \sigma_{t'}^2 \mathbf{I} + \sigma_{t'}^4 h_{t'}(x_{t'})\tag{26}$$

By Markov property, we must have for any measurable function $g$:

$$\mathbb{E}[g(z_{t'})|x_t] = \mathbb{E}[\mathbb{E}[g(z_{t'})|x_t, x_{t'}]|x_t] = \mathbb{E}[\mathbb{E}[g(z_{t'})|x_{t'}]|x_t]$$

Applying this to (26):

$$\sigma_{t'}^4 \mathbb{E}[s(t', x_{t'})s^\top(t', x_{t'})|x_t] = \mathbb{E}[z_{t'}z_{t'}^\top|x_t] - \sigma_{t'}^2 \mathbf{I} - \sigma_{t'}^4 \mathbb{E}[h_{t'}(x_{t'})|x_t]\tag{27}$$

Now, note that $x_t = e^{-t}x_0 + e^{t'-t}z_{t'} + z_{t,t'}$. Taking $y_0 = e^{-t}x_0 + z_{t,t'}$, we have: $x_t = y_0 + e^{t'-t}z_{t'}$. Therefore, applying the second order Tweedie's formula again, we must have:

$$e^{2(t'-t)}\mathbb{E}[z_{t'}z_{t'}^\top|x_t] = e^{4(t'-t)}\sigma_{t'}^4 s(t, x_t)s(t, x_t)^\top + e^{4(t'-t)}\sigma_{t'}^4 h_t(x_t) + e^{2(t'-t)}\sigma_{t'}^2 \mathbf{I}$$

That is : $\mathbb{E}[z_{t'}z_{t'}^\top|x_t] = e^{2(t'-t)}\sigma_{t'}^4 s(t, x_t)s(t, x_t)^\top + e^{2(t'-t)}\sigma_{t'}^4 h_t(x_t) + \sigma_{t'}^2 \mathbf{I}$. Substituting this in Equation (27), we have:

$$\mathbb{E}[s(t', x_{t'})s^\top(t', x_{t'})|x_t] = e^{2(t'-t)}s(t, x_t)s(t, x_t)^\top + e^{2(t'-t)}h_t(x_t) - \mathbb{E}[h_{t'}(x_{t'})|x_t]$$

$\square$

**Lemma 23.** *Let $s_\tau : \mathbb{R}^d \to \mathbb{R}^d$ be continuously differentiable for every $\tau > 0$. For $t > t' > 0$, let $v_{t,t'} := \mathbb{E}\left[x_0|x_t\right] - \mathbb{E}\left[x_0|x_{t'}\right]$, then,*

$$\mathbb{E}\left[v_{t,t'}v_{t,t'}^\top|x_t\right] \preceq$$

$$2e^{2t}\left(\sigma_{t-t'}^4 h_t\left(x_t\right) + \sigma_{t-t'}^2 \mathbf{I}_d\right) + 2e^{2t'}\sigma_{t'}^4 \mathbb{E}\left[\left(s\left(t', x_{t'}\right) - e^{-\left(t-t'\right)}s\left(t, x_t\right)\right)\left(s\left(t', x_{t'}\right) - e^{-\left(t-t'\right)}s\left(t, x_t\right)\right)^\top|x_t\right]$$

*where $h_t\left(x_t\right) := \nabla^2 \log\left(p_t\left(x_t\right)\right)$ is the hessian of the log-density function.*

*Proof.* Using Tweedie's formula, for all $t > 0$,

$$\mathbb{E}\left[x_0|x_t\right] = \mathbb{E}\left[e^t\left(x_t - z_t\right)|x_t\right] = e^t x_t + e^t \mathbb{E}\left[-z_t|x_t\right] = e^t\left(x_t + \sigma_t^2 s\left(t, x_t\right)\right)$$

Note that $x_{t'} = e^{t-t'}\left(x_t - z_{t,t'}\right)$. Furthermore, note from Tweedie's formula and Corollary 2.4 [7] that:

$$\mathbb{E}\left[z_{t,t'}|x_t\right] = -\sigma_{t-t'}^2 s\left(t, x_t\right), \quad \mathbb{E}\left[s\left(t', x_{t'}\right)|x_t\right] = e^{-\left(t-t'\right)}s\left(t, x_t\right)$$

Therefore, we have

$$v_{t,t'} = e^t\left(z_{t,t'} + \sigma_{t-t'}^2 s\left(t, x_t\right)\right) - e^{t'}\sigma_{t'}^2\left(s\left(t', x_{t'}\right) - e^{-\left(t-t'\right)}s\left(t, x_t\right)\right)$$

Then, using Lemma 22 and the fact that $(a + b)(a + b)^\top \preceq 2aa^\top + 2bb^\top$:

$$\mathbb{E}\left[v_{t,t'}v_{t,t'}^\top|x_t\right]$$

$$\preceq 2e^{2t}\mathbb{E}\left[\left(z_{t,t'} + \sigma_{t-t'}^2 s\left(t, x_t\right)\right)\left(z_{t,t'} + \sigma_{t-t'}^2 s\left(t, x_t\right)\right)^\top|x_t\right]$$

$$+ 2e^{2t'}\sigma_{t'}^4 \mathbb{E}\left[\left(s\left(t', x_{t'}\right) - e^{-\left(t-t'\right)}s\left(t, x_t\right)\right)\left(s\left(t', x_{t'}\right) - e^{-\left(t-t'\right)}s\left(t, x_t\right)\right)^\top|x_t\right]$$

$$= 2e^{2t}\left(\sigma_{t-t'}^4 h_t\left(x_t\right) + \sigma_{t-t'}^2 \mathbf{I}_d\right)$$

$$+ 2e^{2t'}\sigma_{t'}^4 \mathbb{E}\left[\left(s\left(t', x_{t'}\right) - e^{-\left(t-t'\right)}s\left(t, x_t\right)\right)\left(s\left(t', x_{t'}\right) - e^{-\left(t-t'\right)}s\left(t, x_t\right)\right)^\top|x_t\right]$$

$\square$

To derive an upper bound for $\left\| \mathbb{E}\left[ \left( s\left(t', x_{t'}\right) - e^{-\left(t-t'\right)} s\left(t, x_t\right) \right) \left( s\left(t', x_{t'}\right) - e^{-\left(t-t'\right)} s\left(t, x_t\right) \right)^\top \mid x_t \right] \right\|_{\text{op}}$,

we adopt a strategy of partitioning the interval $[t', t]$ into smaller subintervals. Specifically, we divide $[t', t]$ as $t' = \tau_0 < \tau_1 < \cdots < \tau_{B-1} < t = \tau_B$, where $B \geq 1$. By leveraging the smoothness of the score function $s_\tau(x)$ over each subinterval $[\tau_i, \tau_{i+1}]$, we express the deviations between $s_{\tau_i}$ and $s_{\tau_{i+1}}$ in terms of the Hessian, $h_\tau(x) := \nabla^2 \log p_\tau(x)$. This decomposition allows us to quantify the overall deviation of the score function across the interval $[t', t]$ in terms of contributions from each subinterval, controlled by the Hessian, $h_\tau(x)$. The following lemma formalizes this approach, establishing an upper bound for the given operator norm in terms of the Hessian and a carefully constructed decomposition. This result will serve as the foundation for subsequent analysis.

**Lemma 24.** *Let $s_\tau : \mathbb{R}^d \to \mathbb{R}^d$ be continuously differentiable for every $\tau > 0$. Let $B \in \mathbb{N}$ and let $\tau_0 := t' < \tau_1 < \tau_2 < \cdots < \tau_{B-1} < t := \tau_B$ for $B \geq 1$ and define $\forall t, h_t(x_t) := \nabla^2 \log\left(p_t(x_t)\right)$. Then,*

$$\left\| \mathbb{E}\left[ \left( s\left(t', x_{t'}\right) - e^{-\left(t-t'\right)} s\left(t, x_t\right) \right) \left( s\left(t', x_{t'}\right) - e^{-\left(t-t'\right)} s\left(t, x_t\right) \right)^\top \mid x_t \right] \right\|_{\text{op}}$$

$$\leq \left\| \mathbb{E}\left[ \sum_{i=0}^{B-1} \mathbb{E}_{\lambda_i, x_{\tau_i}, \tilde{x}_{\tau,i}} \left[ h_{\tau_i}\left(x_{\tau_i, \lambda_i}\right) \left(x_{\tau_i} - \tilde{x}_{\tau_i}\right) \left(x_{\tau_i} - \tilde{x}_{\tau_i}\right)^\top h_{\tau_i}\left(x_{\tau_i, \lambda_i}\right)^\top \mid x_{\tau_{i+1}} \right] \right] \mid x_t \right\|_{\text{op}}$$

*where $\tilde{x}_{\tau_i}$ is an independent copy of $x_{\tau_i}$ when conditioned on $x_{\tau_{i+1}}$. $\lambda_i$ is uniformly distributed over $[0, 1]$ independent of the random variables defined above and $x_{\tau_i, \lambda_i} := \lambda_i x_{\tau_i} + (1 - \lambda_i) \tilde{x}_{\tau_i}$.*

*Proof.* Let $\forall i \in [0, B-1]$, $\Delta_i := \tau_{i+1} - \tau_i$. Then,

$$s\left(t', x_{t'}\right) - e^{-\left(t-t'\right)} s\left(t, x_t\right) = \sum_{i=0}^{B-1} c_i \left( s\left(\tau_i, x_{\tau_i}\right) - e^{-\left(\tau_{i+1} - \tau_i\right)} s\left(\tau_{i+1}, x_{\tau_{i+1}}\right) \right), \quad c_0 = 1, \ c_{i+1} = e^{-\left(\tau_{i+1} - \tau_i\right)} c_i$$

Therefore,

$$\left\| \mathbb{E}\left[ \left( s\left(t', x_{t'}\right) - e^{-\left(t-t'\right)} s\left(t, x_t\right) \right) \left( s\left(t', x_{t'}\right) - e^{-\left(t-t'\right)} s\left(t, x_t\right) \right)^\top \mid x_t \right] \right\|_{\text{op}}$$

$$= \left\| \mathbb{E}\left[ \sum_{0 \leq i, j \leq B-1} c_i c_j \left( s\left(\tau_i, x_{\tau_i}\right) - e^{-\left(\tau_{i+1} - \tau_i\right)} s\left(\tau_{i+1}, x_{\tau_{i+1}}\right) \right) \left( s\left(\tau_j, x_{\tau_j}\right) - e^{-\left(\tau_{j+1} - \tau_i\right)} s\left(\tau_{j+1}, x_{\tau_{j+1}}\right) \right)^\top \mid x_t \right] \right\|_{\text{op}}$$

For $i \neq j$, assuming $i < j$ WLOG, using the Markovian property,

$$\mathbb{E}\left[ \left( s\left(\tau_i, x_{\tau_i}\right) - e^{-\left(\tau_{i+1} - \tau_i\right)} s\left(\tau_{i+1}, x_{\tau_{i+1}}\right) \right) \left( s\left(\tau_j, x_{\tau_j}\right) - e^{-\left(\tau_{j+1} - \tau_i\right)} s\left(\tau_{j+1}, x_{\tau_{j+1}}\right) \right)^\top \mid x_t \right]$$

$$= \mathbb{E}\left[ \mathbb{E}\left[ \left( s\left(\tau_i, x_{\tau_i}\right) - e^{-\left(\tau_{i+1} - \tau_i\right)} s\left(\tau_{i+1}, x_{\tau_{i+1}}\right) \right) \left( s\left(\tau_j, x_{\tau_j}\right) - e^{-\left(\tau_{j+1} - \tau_i\right)} s\left(\tau_{j+1}, x_{\tau_{j+1}}\right) \right)^\top \mid x_{\tau_j}, x_{\tau_{j+1}} \right] \mid x_t \right]$$

$$= \mathbb{E}\left[ \mathbb{E}\left[ s\left(\tau_i, x_{\tau_i}\right) - e^{-\left(\tau_{i+1} - \tau_i\right)} s\left(\tau_{i+1}, x_{\tau_{i+1}}\right) \mid x_{\tau_j}, x_{\tau_{j+1}} \right] \left( s\left(\tau_j, x_{\tau_j}\right) - e^{-\left(\tau_{j+1} - \tau_i\right)} s\left(\tau_{j+1}, x_{\tau_{j+1}}\right) \right)^\top \mid x_t \right]$$

$$= \mathbb{E}\left[ \mathbb{E}\left[ \mathbb{E}\left[ s\left(\tau_i, x_{\tau_i}\right) - e^{-\left(\tau_{i+1} - \tau_i\right)} s\left(\tau_{i+1}, x_{\tau_{i+1}}\right) \mid x_{\tau_i} \right] \mid x_{\tau_j}, x_{\tau_{j+1}} \right] \left( s\left(\tau_j, x_{\tau_j}\right) - e^{-\left(\tau_{j+1} - \tau_i\right)} s\left(\tau_{j+1}, x_{\tau_{j+1}}\right) \right)^\top \mid x_t \right]$$

$$= 0$$

Therefore,

$$\left\| \mathbb{E}\left[ \left( s\left(t', x_{t'}\right) - e^{-\left(t-t'\right)} s\left(t, x_t\right) \right) \left( s\left(t', x_{t'}\right) - e^{-\left(t-t'\right)} s\left(t, x_t\right) \right)^\top \mid x_t \right] \right\|_{\text{op}}$$

$$= \left\| \mathbb{E}\left[ \sum_{i=0}^{B-1} c_i^2 \left( s\left(\tau_i, x_{\tau_i}\right) - e^{-\left(\tau_{i+1} - \tau_i\right)} s\left(\tau_{i+1}, x_{\tau_{i+1}}\right) \right) \left( s\left(\tau_i, x_{\tau_i}\right) - e^{-\left(\tau_{i+1} - \tau_i\right)} s\left(\tau_{i+1}, x_{\tau_{i+1}}\right) \right)^\top \mid x_t \right] \right\|_{\text{op}}$$

$$= \left\| \mathbb{E}\left[ \sum_{i=0}^{B-1} c_i^2 \mathbb{E}\left[ \left( s\left(\tau_i, x_{\tau_i}\right) - e^{-\left(\tau_{i+1} - \tau_i\right)} s\left(\tau_{i+1}, x_{\tau_{i+1}}\right) \right) \left( s\left(\tau_i, x_{\tau_i}\right) - e^{-\left(\tau_{i+1} - \tau_i\right)} s\left(\tau_{i+1}, x_{\tau_{i+1}}\right) \right)^\top \mid x_{\tau_{i+1}} \right] \mid x_t \right] \right\|_{\text{op}}$$

Note that $\mathbb{E}\left[s\left(\tau_i, x_{\tau_i}\right) | x_{\tau_{i+1}}\right] = e^{-(\tau_{i+1} - \tau_i)} s\left(\tau_{i+1}, x_{\tau_{i+1}}\right)$. Therefore,

$$\left\| \mathbb{E}\left[ \left(s\left(t', x_{t'}\right) - e^{-(t-t')} s\left(t, x_t\right)\right) \left(s\left(t', x_{t'}\right) - e^{-(t-t')} s\left(t, x_t\right)\right)^\top | x_t \right] \right\|_{\text{op}} \tag{28}$$

$$\leq \left\| \mathbb{E}\left[ \sum_{i=0}^{B-1} c_i^2 \mathbb{E}\left[ \left(s\left(\tau_i, x_{\tau_i}\right) - s_{\tau_i}\left(\tilde{x}_{\tau_i}\right)\right) \left(s\left(\tau_i, x_{\tau_i}\right) - s_{\tau_i}\left(\tilde{x}_{\tau_i}\right)\right)^\top | x_{\tau_{i+1}} \right] | x_t \right] \right\|_{\text{op}} \tag{29}$$

Using the fundamental theorem of calculus, for $x_{\tau_i, \lambda_i} := \lambda_i x_{\tau_i} + (1 - \lambda_i) \tilde{x}_{\tau_i}, \lambda \in (0,1)$, we have,

$$s\left(\tau_i, x_{\tau_i}\right) - s_{\tau_i}\left(\tilde{x}_{\tau_i}\right) = \int_0^1 h_{\tau_i}\left(x_{\tau_i, \lambda_i}\right)\left(x_{\tau_i} - \tilde{x}_{\tau_i}\right) d\lambda$$

$$= \mathbb{E}_{\lambda \sim \mathcal{U}(0,1)}\left[h_{\tau_i}\left(x_{\tau_i, \lambda}\right)\left(x_{\tau_i} - \tilde{x}_{\tau_i}\right)\right]$$

Substituting in (29) and using the fact that $c_i \leq 1$ completes our proof. $\qquad\square$

We aim to derive a sharp bound on the quantities stated in the previous lemma. Since the Hessian is not assumed to be Lipschitz continuous, directly bounding these quantities can be challenging. To address this, we employ a mollification technique. Mollification smooths a function by averaging it over a small neighborhood, effectively regularizing it to ensure desirable continuity properties. This approach is particularly useful when dealing with functions that may not be smooth or Lipschitz continuous, as it allows us to derive meaningful bounds by working with the mollified version of the function.

In our case, the Hessian is mollified by integrating over a uniformly distributed random variable on a small ball of radius $\epsilon$. This process ensures that the mollified Hessian exhibits controlled variation, enabling us to bound the difference between its values at two points $x$ and $y$. The following lemma formalizes this construction and provides a bound on the operator norm of the difference between the mollified Hessians at $x$ and $y$.

**Lemma 25.** *Let $h : \mathbb{R}^d \to \mathbb{R}^{d \times d}$ such that $\forall x \in \mathbb{R}^d$, $\|h(x)\|_{\text{op}} \leq L$. Let $z$ be uniformly distributed over the unit $\mathbb{L}_2$ ball. For $\epsilon > 0$, define $h_\epsilon(x) := E_z[h_\epsilon(x + \epsilon z)]$. Then, for all $x, y \in \mathbb{R}^d$,*

$$\|h_\epsilon(x) - h_\epsilon(y)\|_{\text{op}} \leq \frac{2Ld}{\epsilon} \|x - y\|_2$$

*Proof.* Define $B(a, R)$ be the ball of radius $R$ around $a$. Define the set $B(x, \epsilon) \cap B(y, \epsilon) = S$ and denote $d\mu_\epsilon$ to be the lebesgue measure over $B(0, \epsilon)$. Then,

$$h_\epsilon(x) - h_\epsilon(y) = \int h(x + Z) d\mu_\epsilon(Z) - \int h(y + Z') d\mu_\epsilon(Z')$$

$$= \frac{1}{|B(0, \epsilon)|} \left[ \int_{B(x, \epsilon)} h(w) dw - \int_{B(y, \epsilon)} h(y) dy \right]$$

$$= \frac{1}{|B(0, \epsilon)|} \left[ \int_{B(x, \epsilon) \cap S^{\complement}} h(w) dw - \int_{B(y, \epsilon) \cap S^{\complement}} h(y) dy \right] \tag{30}$$

$$\implies \|h_\epsilon(x) - h_\epsilon(y)\|_{\text{op}} \leq 2L \frac{\text{Vol}(S^{\complement})}{\text{Vol}(B(0, \epsilon))}$$

Using Theorem 1 from [41], we have

$$\text{Vol}(S^{\complement}) \leq \|x - y\|_2 \times \text{Surf}(B(0, \epsilon))$$

Therefore,

$$\|h_\epsilon(x) - h_\epsilon(y)\|_{\text{op}} \leq 2L \frac{\text{Surf}(B(0, \epsilon))}{\text{Vol}(B(0, \epsilon))} \times \|x - y\|_2$$

We have for $B(0, \epsilon)$, $\frac{\text{Surf}(B(0,\epsilon))}{\text{Vol}(B(0,\epsilon))} = d/\epsilon$ which completes our result. $\qquad\square$

Lemma 25 demonstrates that the mollified Hessian $h_\epsilon$ becomes Lipschitz due to the smoothing introduced by the uniform averaging over the ball $z$, even though the original Hessian $h$ does not have this property. This insight is crucial when dealing with expressions such as

$$\mathbb{E}_{\lambda,x_{t'},\tilde{x}_{t'}}\left[h_{t'}\left(x_{t',\lambda}\right)\left(x_{t'}-\tilde{x}_{t'}\right)\left(x_{t'}-\tilde{x}_{t'}\right)^\top h_{t'}\left(x_{t',\lambda}\right)^\top | x_t\right],$$

which arise from Lemma 24.

When $t$ and $t'$ are close, one would hope to exploit the smoothness of the Hessian $h_t$ with respect to time. Specifically, if $h_t$ were smooth in the time parameter, this would allow the expectation to move inside, enabling the use of Tweedie's second-order formula (Lemma 22) to derive variance bounds that are dimension-free and independent of strong assumptions on the Hessian.

However, directly imposing such strong assumptions on the Hessian is restrictive. To address this, we decompose the Hessian $h_{t'}\left(x_{t',\lambda}\right)$ into two components:

$$h_{t'}\left(x_{t',\lambda}\right) = h_{t',\epsilon}\left(x_{t',\lambda}\right) + \left(h_{t'}\left(x_{t',\lambda}\right) - h_{t',\epsilon}\left(x_{t',\lambda}\right)\right).$$

Here, the first term, $h_{t',\epsilon}\left(x_{t',\lambda}\right)$, leverages the Lipschitz continuity of the mollified Hessian and can be analyzed by conditioning on $x_t$. The second term, which represents the deviation between the original and mollified Hessians, requires a finer analysis that draws upon Lusin's theorem, as developed further in Lemma 28.

The decomposition allows us to systematically address each term: - The Lipschitz property of $h_{t',\epsilon}$ helps bound the first term cleanly. - The second term is bounded using probabilistic arguments based on the regularity properties introduced by mollification.

The following lemma formalizes this decomposition and provides the necessary bounds to proceed with the analysis.

**Lemma 26.** *Suppose Assumption 1-(0) and (1) hold. Let $t > t' > 0$ and define the following quantities:*

1. *Let $\tilde{x}_{t'}$ be an independent copy of $x_{t'}$ when conditioned on $x_t$.*

2. *Let $\lambda \sim \mathsf{Unif}\,(0,1)$ independent of the variables above.*

3. *Let $x_{t',\lambda} := \lambda x_{t'} + (1-\lambda)\tilde{x}_{t'}$, $\tilde{z}_{t,t'} := x_t - e^{-(t-t')}\tilde{x}_{t'}$.*

4. *Let $h_{t'}\left(\cdot\right) := \nabla^2 \log\left(p_{t'}\left(\cdot\right)\right)$.*

5. *For $z$ be uniformly distributed over the unit $\mathbb{L}_2$ ball and $\epsilon > 0$, define $h_{t',\epsilon}(x) := E_z[h_{t'}(x + \epsilon z)]$.*

6. *Let $g_{t',\epsilon}\left(x_{t',\lambda}\right) := \left(h_{t'}\left(x_{t',\lambda}\right) - h_{t',\epsilon}(x_{t',\lambda})\right)$.*

*Then, there exists a random $d \times d$ matrix $M$ such that $\|M\|_{\mathrm{op}} \le \frac{2Ld}{\epsilon}\|(1-\lambda)\,z_{t,t'} + \lambda\tilde{z}_{t,t'}\|_2$ and*

$$\left\|\mathbb{E}_{\lambda,x_{t'},\tilde{x}_{t'}}\left[h_{t'}\left(x_{t',\lambda}\right)\left(x_{t'}-\tilde{x}_{t'}\right)\left(x_{t'}-\tilde{x}_{t'}\right)^\top h_{t'}\left(x_{t',\lambda}\right)^\top | x_t\right]\right\|_{\mathrm{op}}$$

$$\le 6e^{2\left(t-t'\right)}\left\|h_{t',\epsilon}\left(e^{t-t'}x_t\right)\left(\sigma_{t-t'}^4 h_t\left(x_t\right) + \sigma_{t-t'}^2\mathbf{I}_d\right)h_{t',\epsilon}\left(e^{t-t'}x_t\right)^\top\right\|_{\mathrm{op}}$$

$$+ 3\left(\mathbb{E}_{\lambda,x_{t'},\tilde{x}_{t'}}\left[\|M\|_{\mathrm{op}}^2\|x_{t'}-\tilde{x}_{t'}\|_{\mathrm{op}}^2 | x_t\right] + \mathbb{E}_{x_{t'},\tilde{x}_{t'}}\left[\|\mathbb{E}_\lambda\left[g_{t',\epsilon}\left(x_{t',\lambda}\right)\right]\|_{\mathrm{op}}^2\|x_{t'}-\tilde{x}_{t'}\|_{\mathrm{op}}^2 | x_t\right]\right)$$

*Proof.* By assumption, we have $\forall x \in \mathbb{R}^d$, $\|h_t\left(x\right)\|_2 \le L$. Note that conditioned on $x_t$, we have

$$x_t = e^{-\left(t-t'\right)}x_{t'} + z_{t,t'} = e^{-\left(t-t'\right)}\tilde{x}_{t'} + \tilde{z}_{t,t'}$$

Where $\tilde{z}_{t,t'} \sim \mathcal{N}(0,\sigma_{t,t'}^2\mathbf{I}_d)$ marginally. Therefore,

$$x_{t',\lambda} = e^{t-t'}x_t - e^{t-t'}\left((1-\lambda)\,z_{t,t'} + \lambda\tilde{z}_{t,t'}\right)$$

Using Lemma 25,

$$h_{t',\epsilon}\left(x_{t',\lambda}\right) = h_{t',\epsilon}\left(e^{t-t'}x_t\right) + M, \quad \text{for } \|M\|_{\mathrm{op}} \le \frac{2Ld}{\epsilon}\|(1-\lambda)\,z_{t,t'} + \lambda\tilde{z}_{t,t'}\|_2$$

Then,

$$h_{t'}\left(x_{t',\lambda}\right) = h_{t',\epsilon}(x_{t',\lambda}) + \left(h_{t'}\left(x_{t',\lambda}\right) - h_{t',\epsilon}(x_{t',\lambda})\right)$$
$$= h_{t',\epsilon}\left(e^{t-t'}x_t\right) + M + \left(h_{t'}\left(x_{t',\lambda}\right) - h_{t',\epsilon}(x_{t',\lambda})\right)$$

Let $q_t := \mathbb{E}_{\lambda,x_{t'},\tilde{x}_{t'}}\left[h_{t'}\left(x_{t',\lambda}\right)\left(x_{t'} - \tilde{x}_{t'}\right)\left(x_{t'} - \tilde{x}_{t'}\right)^\top h_{t'}\left(x_{t',\lambda}\right)^\top |x_t\right]$ and $g_{t',\epsilon}\left(x_{t',\lambda}\right) :=$ $\left(h_{t'}\left(x_{t',\lambda}\right) - h_{t',\epsilon}(x_{t',\lambda})\right)$. Then, using the fact that $(a+b+c)(a+b+c)^\top \preceq 3(aa^\top + bb^\top + cc^\top)$ for arbitrary vectors $a, b, c \in \mathbb{R}^d$, we have:

$$q_t \preceq 3\underbrace{\mathbb{E}_{\lambda,x_{t'},\tilde{x}_{t'}}\left[h_{t',\epsilon}\left(e^{t-t'}x_t\right)\left(x_{t'} - \tilde{x}_{t'}\right)\left(x_{t'} - \tilde{x}_{t'}\right)^\top h_{t',\epsilon}\left(e^{t-t'}x_t\right)^\top |x_t\right]}_{:=T_1}$$

$$+ 3\underbrace{\mathbb{E}_{\lambda,x_{t'},\tilde{x}_{t'}}\left[M\left(x_{t'} - \tilde{x}_{t'}\right)\left(x_{t'} - \tilde{x}_{t'}\right)^\top M^\top |x_t\right]}_{:=T_2}$$

$$+ 3\underbrace{\mathbb{E}_{\lambda,x_{t'},\tilde{x}_{t'}}\left[g_{t',\epsilon}\left(x_{t',\lambda}\right)\left(x_{t'} - \tilde{x}_{t'}\right)\left(x_{t'} - \tilde{x}_{t'}\right)^\top g_{t',\epsilon}\left(x_{t',\lambda}\right)^\top |x_t\right]}_{:=T_3}$$

Let's first deal with $T_1$. We use the fact that $x_t = e^{-(t-t')}x_{t'} + z_{t,t'} = e^{-(t-t')}\tilde{x}_{t'} + \tilde{z}_{t,t'}$ along with first order and second order Tweedie's formula in Lemma 22

$$T_1 = 2e^{2(t-t')}h_{t',\epsilon}\left(e^{t-t'}x_t\right)\left(\sigma_{t-t'}^4 h_t\left(x_t\right) + \sigma_{t-t'}^2\mathbf{I}_d\right)h_{t',\epsilon}\left(e^{t-t'}x_t\right)^\top$$

Now, for $T_2$, we have

$$T_2 = \mathbb{E}_{\lambda,x_{t'},\tilde{x}_{t'}}\left[M\left(x_{t'} - \tilde{x}_{t'}\right)\left(x_{t'} - \tilde{x}_{t'}\right)^\top M^\top |x_t\right]$$
$$\preceq \mathbb{E}_{\lambda,x_{t'},\tilde{x}_{t'}}\left[\|M\|_{op}^2 \|x_{t'} - \tilde{x}_{t'}\|_{op}^2 |x_t\right]\mathbf{I}_d$$

and similarly for $T_3$,

$$T_3 \preceq \mathbb{E}_{x_{t'},\tilde{x}_{t'}}\left[\|\mathbb{E}_\lambda\left[g_{t',\epsilon}\left(x_{t',\lambda}\right)\right]\|_{op}^2 \|x_{t'} - \tilde{x}_{t'}\|_{op}^2 |x_t\right]\mathbf{I}_d$$

which completes our proof. $\qquad\square$

Lemma 27 provides a corollary of Lusin's theorem (see for e.g. [11]) to assert that any measurable function, such as the Hessian $h_t(x) = \nabla^2 \log p_t(x)$, can be approximated uniformly on a compact subset $G_\gamma \subseteq [t', t] \times F$, where the excluded measure is arbitrarily small. This result ensures that $h_t(x)$ is uniformly continuous on $G_\gamma$, with its continuity quantified by a modulus of continuity $\omega_\gamma(\cdot)$ depending only on $\gamma$. See [39] for Heine–Cantor theorem which implies uniform continuity due to compactness.

**Lemma 27** (Corollary of Lusin's Theorem). *Let $F$ be a convex, compact set over $\mathbb{R}^d$ and $\Lambda$ be the Lebesgue measure. Let $h_t(x) = \nabla^2 \log p_t(x)$ be measurable. For any $\gamma > 0$, there exists a compact set $G_\gamma \subseteq [t', t] \times F$ such that $\Lambda([t', t] \times F) \setminus G_\gamma) < \gamma$ and $(t, x) \to h_t(x)$ is uniformly continuous over $G_\gamma$. Let us call the corresponding modulus of continuity as $\omega_\gamma()$, which depends only on $\gamma$.*

Building on Lemma 27, Lemma 28 aims to bound the fourth moment of the operator norm of the difference $h_{\tau_i}(x_{\tau_i,\lambda}) - h_{\tau_i,\epsilon}(x_{\tau_i,\lambda})$, which arises from the deviation between the Hessian and its mollified counterpart. To achieve this, the interval $[t', t]$ is partitioned into smaller subintervals $\tau_0, \tau_1, \ldots, \tau_B$, allowing the analysis to proceed incrementally. The lemma exploits the uniform continuity of $h_t(x)$ on $G_\gamma$ to tightly control this difference using the modulus of continuity $\omega_\gamma(\epsilon)$. Contributions from outside the compact subset $G_\gamma$ are accounted for separately using indicator functions, with their impact controlled by the boundedness of the Hessian, $\|h_t(x)\|_{op} \leq L$. The resulting bound consists of two key terms: a primary term proportional to $B\omega_\gamma(\epsilon)^4$, capturing the uniform continuity of the Hessian on $G_\gamma$, and a residual term proportional to the probability of $h_t(x)$ lying outside $G_\gamma$, which is effectively managed by the boundedness assumption. This decomposition is crucial for controlling the variance of the Hessian and ensuring the residual terms remain small.

**Lemma 28.** *Fix a $B \in \mathbb{N}$. Let $\tau_0 := t' < \tau_1 < \tau_2 < \cdots < \tau_{B-1} < t := \tau_B$. Let Assumption 1-(0),(1) hold. Let $h_t(x), h_{t,\epsilon}(x)$ be defined as in Lemma 26. Let $Z$ be uniformly distributed on the unit $L^2$ ball in $\mathbb{R}^d$, independent of everything else. Then for any $\gamma > 0$:*

$$\sum_{i=0}^{B-1} \mathbb{E}_{x_{\tau_i}, \tilde{x}_{\tau_i}} \left[ \left\| \mathbb{E}_{\lambda \sim \mathsf{Unif}(0,1)} \left[ h_{\tau_i}(x_{\tau_i,\lambda_i}) - h_{\tau_i,\epsilon}(x_{\tau_i,\lambda_i}) \right] \right\|_{\mathrm{op}}^4 \big| x_{\tau_{i+1}} \right] \leq$$

$$B \omega_\gamma(\epsilon)^4 + 16 L^4 \sum_{i=0}^{B-1} \mathbb{E}_{x_{\tau_i}, \tilde{x}_{\tau_i}} \left[ \int_0^1 \mathbb{1}((\tau_i, x_{\tau_i,\lambda}) \notin G_\gamma) + \mathbb{1}((\tau_i, x_{\tau_i,\lambda} + \epsilon Z) \notin G_\gamma) d\lambda \big| x_{\tau_{i+1}} \right]$$

*where $x_{\tau_i}$ is an i.i.d copy of $\tilde{x}_{\tau_i}$ conditioned on $x_{\tau_{i+1}}$ and $x_{\tau_i,\lambda} := \lambda x_{\tau_i} + (1 - \lambda) \tilde{x}_{\tau_i}$ for any given $\lambda \in [0, 1]$ and $\omega_\gamma, G_\gamma$ are as defined in Lemma 27.*

*Proof.* Let us consider Lusin's theorem (Lemma 27) over $[t', t] \times F$ endowed with the Lebesgue measure $\Lambda$. By Assumption 1-(0),(1): we have $\|h_t(x)\| \leq L$ for every $t$ almost everywhere under the Lebesgue measure on $\mathbb{R}^d$. We denote $\mathbb{E}_{\lambda \sim \mathsf{Unif}(0,1)}$ as $\mathbb{E}_\lambda$ and only in the set of equations below, we denote expectation with respect to $x_{\tau_i}, \tilde{x}_{\tau_i}, Z$ conditioned on $x_{\tau_{i+1}}$ by $\bar{\mathbb{E}}$:

$$\bar{\mathbb{E}} \left[ \left\| \mathbb{E}_\lambda \left[ h_{\tau_i}(x_{\tau_i,\lambda}) - h_{\tau_i,\epsilon}(x_{\tau_i,\lambda}) \right] \right\|_{\mathrm{op}}^4 \big| x_{\tau_{i+1}} \right] \tag{31}$$

$$= \bar{\mathbb{E}} \left[ \left\| \int_0^1 h_{\tau_i}(x_{\tau_i,\lambda}) - h_{\tau_i,\epsilon}(x_{\tau_i,\lambda}) d\lambda \right\|_{\mathrm{op}}^4 \big| x_{\tau_{i+1}} \right]$$

$$\leq \bar{\mathbb{E}} \| \int_0^1 h_{\tau_i}(x_{\tau_i,\lambda}) - h_{\tau_i}(x_{\tau_i,\lambda} + \epsilon Z) d\lambda \|_{\mathrm{op}}^4$$

$$\leq \bar{\mathbb{E}} \int_0^1 \mathbb{1}((\tau_i, x_{\tau_i,\lambda}) \in G_\gamma) \mathbb{1}((\tau_i, x_{\tau_i,\lambda} + \epsilon Z) \in G_\gamma) \omega_\gamma(\epsilon)^4 d\lambda$$

$$+ \bar{\mathbb{E}} \int_0^1 \left[ \mathbb{1}((\tau_i, x_{\tau_i,\lambda}) \notin G_\gamma) + \mathbb{1}((\tau_i, x_{\tau_i,\lambda} + \epsilon Z) \notin G_\gamma) \right] 16 L^4 d\lambda$$

$$\leq \omega_\gamma(\epsilon)^4 + \bar{\mathbb{E}} \int_0^1 \left[ \mathbb{1}((\tau_i, x_{\tau_i,\lambda}) \notin G_\gamma) + \mathbb{1}((\tau_i, x_{\tau_i,\lambda} + \epsilon Z) \notin G_\gamma) \right] 16 L^4 d\lambda \tag{32}$$

Therefore, we must have:

$$\sum_{i=0}^{B-1} \mathbb{E} \left[ \| \int_0^1 h_{\tau_i}(x_{\tau_i,\lambda}) - h_{\tau_i,\epsilon}(x_{\tau_i,\lambda}) d\lambda \|^4 \right]$$

$$\leq B \omega_\gamma(\epsilon)^2 + 16 L^4 \sum_{i=0}^{B-1} \mathbb{E} \left[ \int_0^1 \mathbb{1}((\tau_i, x_{\tau_i,\lambda}) \notin G_\gamma) + \mathbb{1}((\tau_i, x_{\tau_i,\lambda} + \epsilon) \notin G_\gamma) d\lambda \right]$$

$\square$

The following lemma consolidates the results and arguments developed so far to provide a variance bound for a martingale difference sequence. Our goal is to bound the variance of the terms in the sequence $R_{i,k}$, which is determined by both the predictable sequence $G_{i,k+1}$ and the smoothness properties of the score function and its Hessian. To achieve this, we build on several key results:

1. Lemma 28, which establishes bounds for the difference between the Hessian and its mollified counterpart by leveraging the compactness provided by Lusin's theorem.

2. Lemma 26, which shows how the mollified Hessian can be used to control variance terms using its Lipschitz properties.

3. Lemma 24, which provides a decomposition of the conditional variance in terms of contributions from smaller subintervals.

The argument proceeds by partitioning the time interval $[t_{N-k}, t_{N-k+1}]$ into smaller subintervals and analyzing the contributions to the variance over each subinterval. Using mollification and uniform

continuity on compact subsets, we control the deviations arising from the lack of Lipschitz continuity in the Hessian. Furthermore, the variance bounds incorporate the contributions from outside the compact subset, which are managed via Lusin's theorem. By carefully summing these contributions and leveraging smoothing techniques, we arrive at a sharp variance bound that scales with the parameters $\Delta$ (the interval size) and $L$ (the bound on the Hessian)

The final result, formalized in Lemma 29, also uses the second-order Tweedie formula to handle the special case of the last time step ($k = N$) in the martingale sequence. This lemma serves as a culmination of our efforts, combining mollification, decomposition, and smoothness assumptions to derive a practical variance bound that is essential for analyzing the concentration of the martingale difference sequence.

**Lemma 29** (Variance bound for martingale difference sequence). *Consider the martingale difference sequence $R_{i,k}$, predictable sequence $G_{i,k+1}$ with respect to the filtration $\mathcal{F}_{i,k}$ as considered in Lemma 21. Define $\Delta := t_{N-k+1} - t_{N-k}$*

$$\mathbb{E}\left[R_{i,k}^2 | \mathcal{F}_{i,k-1}\right] \leq \begin{cases} 0 & \text{if } k = 0 \\ C(L\Delta^2 + \Delta + L^2\Delta)e^{2t_{N-k+1}}\|G_{i,k+1}\|^2 & \text{if } k \in \{1, \ldots, N-1\} \\ C(L\Delta^2 + \Delta)\|\bar{G}_i\|^2 & \text{if } k = N \end{cases} \quad (33)$$

*Proof.* Consider the case $k \in \{1, \ldots, N-1\}$. For the sake of clarity, we let $t = t_{N-k+1}, t' = t_{N-k}$. Then, $\Delta = t - t'$ and let $B \in \mathbb{N}$. We decompose $[t', t]$ as follows:

$$[t', t] = \cup_{i=1}^B I_i \quad ; \quad I_i := [t' + \tfrac{(i-1)\Delta}{B}, t' + \tfrac{i\Delta}{B}].$$

For $\forall i \in [B]$, $\tau_i \sim \mathsf{Unif}(I_i)$, $J \sim \mathsf{Unif}(\{1, \ldots, B\})$. Given $\tau_i$, define the random variables $Z, \lambda, x_{\tau_i,\lambda}, \tilde{x}_{\tau_i,\lambda}, x_{\tau_i}$ as in Lemma 28 and with $Z, \lambda, (x_s)_{s \geq 0}$ indepenent of $(\tau_i)_i, J$. Define the random variable $\tau^* := \tau_J$, $X = x_{\tau^*,\lambda}$, $X_\epsilon = x_{\tau^*,\lambda} + \epsilon Z$. Notice that $T$ is uniformly distributed over $[t', t]$.

Let $r_i := \tau_{i+1} - \tau_i \leq \frac{\Delta}{B}$. Using Lemma 26 along with the Cauchy-Schwarz inequality, we have

$$\left\| \mathbb{E}_{\lambda_i, x_{\tau_i}, \tilde{x}_{\tau_i}} \left[ h_{\tau_i}\left(x_{\tau_i,\lambda_i}\right)\left(x_{\tau_i} - \tilde{x}_{\tau_i}\right)\left(x_{\tau_i} - \tilde{x}_{\tau_i}\right)^\top h_{\tau_i}\left(x_{\tau_i,\lambda_i}\right)^\top | x_{\tau_{i+1}} \right] \right\|_{\text{op}}$$

$$\leq 6 e^{2r_i} \left\| h_{\tau_i,\epsilon}\left(e^{r_i} x_t\right)\left(\sigma_{r_i}^4 h_{\tau_i}\left(x_{\tau_i}\right) + \sigma_{r_i}^2 \mathbf{I}_d\right) h_{\tau_i,\epsilon}\left(e^{r_i} x_{\tau_i}\right)^\top \right\|_{\text{op}}$$

$$+ \frac{12L^2 d^2}{\epsilon^2} \mathbb{E}_{\lambda, x_{\tau_i}, \tilde{x}_{\tau_i}} \left[ \left\| (1-\lambda) z_{\tau_{i+1},\tau_i} + \lambda \tilde{z}_{\tau_{i+1},\tau_i} \right\|_2^4 | x_{\tau_{i+1}} \right]^{\frac{1}{2}} \mathbb{E}_{\lambda, x_{\tau_i}, \tilde{x}_{\tau_i}} \left[ \left\| x_{\tau_i} - \tilde{x}_{\tau_i} \right\|_{\text{op}}^4 | x_{\tau_{i+1}} \right]^{\frac{1}{2}}$$

$$+ 3\mathbb{E}_{x_{\tau_i}, \tilde{x}_{\tau_i}} \left[ \left\| \mathbb{E}_{\lambda_i}\left[ h_{\tau_i}\left(x_{\tau_i,\lambda_i}\right) - h_{\tau_i,\epsilon}(x_{\tau_i,\lambda_i}) \right] \right\|_{\text{op}}^4 | x_{\tau_{i+1}} \right]^{\frac{1}{2}} \mathbb{E}_{\lambda_i, x_{\tau_i}, \tilde{x}_{\tau_i}} \left[ \left\| x_{\tau_i} - \tilde{x}_{\tau_i} \right\|_{\text{op}}^4 | x_{\tau_{i+1}} \right]^{\frac{1}{2}}$$

$$(34)$$

Using Lemma 24 along with (34) and Cauchy Schwarz inequality, we have

$$\left\| \mathbb{E}\left[ \left(s(t', x_{t'}) - e^{-(t-t')} s(t, x_t)\right)\left(s(t', x_{t'}) - e^{-(t-t')} s(t, x_t)\right)^\top | x_t \right] \right\|_{\text{op}}$$

$$\leq \left\| \mathbb{E}\left[ \sum_{i=0}^{B-1} \mathbb{E}_{\lambda_i, x_{\tau_i}, \tilde{x}_{\tau,i}} \left[ h_{\tau_i}\left(x_{\tau_i,\lambda_i}\right)\left(x_{\tau_i} - \tilde{x}_{\tau_i}\right)\left(x_{\tau_i} - \tilde{x}_{\tau_i}\right)^\top h_{\tau_i}\left(x_{\tau_i,\lambda_i}\right)^\top | x_{\tau_{i+1}} \right] \right] | x_t \right\|_{\text{op}}$$

$$\leq 6 \sum_{i=0}^{B-1} e^{2r_i} \mathbb{E}\left[ \left\| h_{\tau_i,\epsilon}\left(e^{r_i} x_t\right)\left(\sigma_{r_i}^4 h_{\tau_i}\left(x_{\tau_i}\right) + \sigma_{r_i}^2 \mathbf{I}_d\right) h_{\tau_i,\epsilon}\left(e^{r_i} x_{\tau_i}\right)^\top \right\|_{\text{op}} | x_t \right]$$

$$+ \frac{12L^2 d^2}{\epsilon^2} \sum_{i=0}^{B-1} \mathbb{E}\left[ \mathbb{E}_{\lambda, x_{\tau_i}, \tilde{x}_{\tau_i}} \left[ \left\| (1-\lambda) z_{\tau_{i+1},\tau_i} + \lambda \tilde{z}_{\tau_{i+1},\tau_i} \right\|_2^4 | x_{\tau_{i+1}} \right]^{\frac{1}{2}} \mathbb{E}_{\lambda, x_{\tau_i}, \tilde{x}_{\tau_i}} \left[ \left\| x_{\tau_i} - \tilde{x}_{\tau_i} \right\|_{\text{op}}^4 | x_{\tau_{i+1}} \right]^{\frac{1}{2}} | x_t \right]$$

$$+ 3 \sum_{i=0}^{B-1} \mathbb{E}\left[ \mathbb{E}_{x_{\tau_i}, \tilde{x}_{\tau_i}} \left[ \left\| \mathbb{E}_{\lambda_i}\left[ h_{\tau_i}\left(x_{\tau_i,\lambda_i}\right) - h_{\tau_i,\epsilon}(x_{\tau_i,\lambda_i}) \right] \right\|_{\text{op}}^4 | x_{\tau_{i+1}} \right]^{\frac{1}{2}} \mathbb{E}_{\lambda_i, x_{\tau_i}, \tilde{x}_{\tau_i}} \left[ \left\| x_{\tau_i} - \tilde{x}_{\tau_i} \right\|_{\text{op}}^4 | x_{\tau_{i+1}} \right]^{\frac{1}{2}} | x_t \right]$$

$$\leq 6 \sum_{i=0}^{B-1} e^{2r_i} \mathbb{E}\left[\left\|h_{\tau_i,\epsilon}\left(e^{r_i} x_t\right)\left(\sigma_{r_i}^4 h_{\tau_i}\left(x_{\tau_i}\right) + \sigma_{r_i}^2 \mathbf{I}_d\right) h_{\tau_i,\epsilon}\left(e^{r_i} x_{\tau_i}\right)^\top\right\|_{\mathrm{op}} |x_t\right]$$

$$+ \frac{12 L^2 d^2}{\epsilon^2} \sum_{i=0}^{B-1} \mathbb{E}\left[\mathbb{E}_{\lambda, x_{\tau_i}, \tilde{x}_{\tau_i}}\left[\left\|(1-\lambda) z_{\tau_{i+1}, \tau_i} + \lambda \tilde{z}_{\tau_{i+1}, \tau_i}\right\|_2^4 |x_{\tau_{i+1}}\right]^{\frac{1}{2}} \mathbb{E}_{\lambda, x_{\tau_i}, \tilde{x}_{\tau_i}}\left[\left\|x_{\tau_i} - \tilde{x}_{\tau_i}\right\|_{\mathrm{op}}^4 |x_{\tau_{i+1}}\right]^{\frac{1}{2}} |x_t\right]$$

$$+ 3\mathbb{E}\left[\left(\sum_{i=0}^{B-1} \mathbb{E}_{x_{\tau_i}, \tilde{x}_{\tau_i}}\left[\left\|\mathbb{E}_{\lambda_i}\left[h_{\tau_i}\left(x_{\tau_i}, \lambda_i\right) - h_{\tau_i,\epsilon}(x_{\tau_i}, \lambda_i)\right]\right\|_{\mathrm{op}}^4 |x_{\tau_{i+1}}\right]\right)^{\frac{1}{2}} \left(\sum_{i=0}^{B-1} \mathbb{E}_{\lambda_i, x_{\tau_i}, \tilde{x}_{\tau_i}}\left[\left\|x_{\tau_i} - \tilde{x}_{\tau_i}\right\|_{\mathrm{op}}^4 |x_{\tau_{i+1}}\right]\right)^{\frac{1}{2}} |x_t\right]$$

(35)

Using (35) and the observation that $\mathbb{E}_{\lambda_i, x_{\tau_i}, \tilde{x}_{\tau_i}}\left[\left\|x_{\tau_i} - \tilde{x}_{\tau_i}\right\|_{\mathrm{op}}^4 |x_{\tau_{i+1}}\right]^{\frac{1}{2}} = O\left(\sigma_{r_i}^2 d\right) = O\left(\frac{\Delta d}{B}\right)$, we have

$$\left\|\mathbb{E}\left[\left(s\left(t', x_{t'}\right) - e^{-(t-t')} s\left(t, x_t\right)\right)\left(s\left(t', x_{t'}\right) - e^{-(t-t')} s\left(t, x_t\right)\right)^\top |x_t\right]\right\|_{\mathrm{op}}$$

$$\leq 3B e^{\frac{2\Delta}{B}}\left(\frac{L^3 \Delta^2}{B^2} + \frac{L^2 \Delta}{B}\right) + \frac{12 \Delta^2 L^2 d^4}{B \epsilon^2} + \frac{3\Delta d}{\sqrt{B}}\left(\sum_{i=0}^{B-1} \mathbb{E}_{x_{\tau_i}, \tilde{x}_{\tau_i}}\left[\left\|\mathbb{E}_{\lambda_i}\left[h_{\tau_i}\left(x_{\tau_i}, \lambda_i\right) - h_{\tau_i,\epsilon}(x_{\tau_i}, \lambda_i)\right]\right\|_{\mathrm{op}}^4 |x_{\tau_{i+1}}\right]\right)^{\frac{1}{2}}$$

Using Lemma 28,

$$\left(\sum_{i=0}^{B-1} \mathbb{E}_{x_{\tau_i}, \tilde{x}_{\tau_i}}\left[\left\|\mathbb{E}_{\lambda_i}\left[h_{\tau_i}\left(x_{\tau_i}, \lambda_i\right) - h_{\tau_i,\epsilon}(x_{\tau_i}, \lambda_i)\right]\right\|_{\mathrm{op}}^4 |x_{\tau_{i+1}}\right]\right)^{\frac{1}{2}}$$

$$\leq \sqrt{B} \omega_\gamma(\epsilon)^2 + 2L^2 \left(\sum_{i=0}^{B-1} \mathbb{E}\left[\int_0^1 \mathbb{1}((\tau_i, x_{\lambda_i, \tau_i}) \notin G_\gamma) + \mathbb{1}((\tau_i, x_{\lambda_i, \tau_i} + \epsilon Z_i) \notin G_\gamma) d\lambda_i\right]\right)^{\frac{1}{2}}$$

$$\leq \sqrt{B} \omega_\gamma(\epsilon)^2 + 2L^2 \left(B\left(\mathbb{P}((T, X) \notin G_\gamma) + \mathbb{P}((T, X_\epsilon) \notin G_\gamma)\right)\right)^{\frac{1}{2}}$$

Therefore,

$$\left\|\mathbb{E}\left[\left(s\left(t', x_{t'}\right) - e^{-(t-t')} s\left(t, x_t\right)\right)\left(s\left(t', x_{t'}\right) - e^{-(t-t')} s\left(t, x_t\right)\right)^\top |x_t\right]\right\|_{\mathrm{op}}$$

$$\leq 6B e^{\frac{2\Delta}{B}}\left(\frac{L^3 \Delta^2}{B^2} + \frac{L^2 \Delta}{B}\right) + \frac{12 \Delta^2 L^2 d^4}{B \epsilon^2} + 6L^2 \Delta d\left(\omega_\gamma(\epsilon)^2 + \left(\mathbb{P}((T, X) \notin G_\gamma) + \mathbb{P}((T, X_\epsilon) \notin G_\gamma)\right)^{\frac{1}{2}}\right)$$

Notice that none of $\omega_\gamma, G_\gamma$, distribution of $T, X$ depend on $B$. Therefore pick $\epsilon \to 0$ and $B \to \infty$ such that $\frac{1}{B\epsilon^2} \to 0$ and $\omega_\gamma(\epsilon) \to 0$. $(T, X_\epsilon) \to (T, X)$ almost surely as $\epsilon \to 0$. Then, we take $\gamma \to 0$ and argue via continuity of the law of $(T, X)$ with respect to Lebesgue measure that $\mathbb{P}((T, X) \notin G_\gamma) \to \mathbb{P}((T, X) \notin [t', t] \times F)$. Since $F$ is arbitrary compact convex set, we let $F \uparrow \mathbb{R}^d$ to conclude the following:

$$\left\|\mathbb{E}\left[\left(s\left(t', x_{t'}\right) - e^{-(t-t')} s\left(t, x_t\right)\right)\left(s\left(t', x_{t'}\right) - e^{-(t-t')} s\left(t, x_t\right)\right)^\top |x_t\right]\right\|_{\mathrm{op}} = O\left(L^2 \Delta\right) \quad (36)$$

Using Lemma 23, we have

$$\left\|\mathbb{E}\left[\left(\mathbb{E}[x_0|x_t] - \mathbb{E}[x_0|x_{t'}]\right)\left(\mathbb{E}[x_0|x_t] - \mathbb{E}[x_0|x_{t'}]\right)^\top |x_t\right]\right\|_{\mathrm{op}}$$

$$\leq 2e^{2t}\left\|\left(\sigma_{t-t'}^4 h_t\left(x_t\right) + \sigma_{t-t'}^2 \mathbf{I}_d\right)\right\|_{\mathrm{op}}$$

$$+ 2e^{2t'} \sigma_{t'}^4 \left\|\mathbb{E}\left[\left(s\left(t', x_{t'}\right) - e^{-(t-t')} s\left(t, x_t\right)\right)\left(s\left(t', x_{t'}\right) - e^{-(t-t')} s\left(t, x_t\right)\right)^\top |x_t\right]\right\|_{\mathrm{op}}$$

$$= O\left(e^{2t}\left(L\Delta^2 + \Delta + L^2 \Delta\right)\right)$$

The result for $k < N$ then follows due to Lemma 21.

Now, consider the case $k = N$. Recall $\Sigma_{i,k}$ defined in Lemma 21. Then by second order Tweedie formula (Lemma 22) we have $\Sigma_{i,k} = \sigma_{t_1}^4 h_{t_1}(x_{t_1}) + \sigma_{t_1}^2 \mathbf{I}_d \lesssim \Delta^2 L + \Delta$. Combining this with Lemma 21, we conclude the result. $\qquad\square$

We state a useful corollary which is subsequently useful for time bootstrapping and is implicit in the above proof.

**Corollary 1.** *Let $t' < t$ and $\Delta := t - t'$. Then, under Assumption 1,*

$$\left\| \mathbb{E}\left[ \left( s\left( t', x_{t'} \right) - e^{-(t-t')} s\left( t, x_t \right) \right) \left( s\left( t', x_{t'} \right) - e^{-(t-t')} s\left( t, x_t \right) \right)^\top | x_t \right] \right\|_{\text{op}} = O\left( L^2 \Delta \right)$$

*Proof.* The proof is implicit due to (36). □

**Lemma 30.** *Fix $\delta \in (0,1)$. Consider $R_{i,k}$ and $\mathcal{F}_{i,k}$ as defined in Lemma 2 and let $\Delta := t_{N-k+1} - t_{N-k}$. Under Assumption 1, following the definition in Definition 1, conditioned on $\mathcal{F}_{i,k-1}$, $R_{i,k}$ is $(\beta_{i,k}^2 \|G_{i,k}\|^2, W_{i,k})$-subGaussian where $\beta_{i,k}, W_{i,k}$ are $\mathcal{F}_{i,k-1}$ measurable random variables such that $W_{i,k} \leq \log\left(\frac{2}{\delta}\right)$ with probability at-least $1 - \delta$ and*

$$\beta_{i,k} := \begin{cases} 8\left(L+1\right) e^{t_{N-k+1}} \sqrt{\Delta d}, & k \in [N-1], \\ 4\sqrt{\Delta d}, & k = N \end{cases}$$

*Proof.* We have,

$$\mathbb{P}\left( \left| \left\langle G_{i,k+1}, \mathbb{E}[x_0^{(i)} | x_{t_{N-k+1}}^{(i)}] - \mathbb{E}[x_0^{(i)} | x_{t_{N-k}}^{(i)}] \right\rangle \right| \geq \alpha | \mathcal{F}_{i,k-1} \right)$$

$$= \mathbb{P}\left( \left| \left\langle G_{i,k+1}, \mathbb{E}[x_0^{(i)} | x_{t_{N-k+1}}^{(i)}] - \mathbb{E}[x_0^{(i)} | x_{t_{N-k}}^{(i)}] \right\rangle \right|^2 \geq \alpha^2 | \mathcal{F}_{i,k-1} \right)$$

$$= \mathbb{P}\left( \exp\left( \lambda \left\langle G_{i,k+1}, \mathbb{E}[x_0^{(i)} | x_{t_{N-k+1}}^{(i)}] - \mathbb{E}[x_0^{(i)} | x_{t_{N-k}}^{(i)}] \right\rangle^2 \right) \geq \exp\left( \lambda \alpha^2 \right) | \mathcal{F}_{i,k-1} \right)$$

$$\leq \exp\left( -\lambda \alpha^2 \right) \mathbb{E}\left[ \exp\left( \lambda \left\langle G_{i,k+1}, \mathbb{E}[x_0^{(i)} | x_{t_{N-k+1}}^{(i)}] - \mathbb{E}[x_0^{(i)} | x_{t_{N-k}}^{(i)}] \right\rangle^2 \right) | \mathcal{F}_{i,k-1} \right]$$

$$= \exp\left( -\lambda \alpha^2 \right) \mathbb{E}\left[ \exp\left( \lambda \left\langle G_{i,k+1}, \mathbb{E}[x_0^{(i)} | x_{t_{N-k+1}}^{(i)}] - \mathbb{E}[x_0^{(i)} | x_{t_{N-k}}^{(i)}] \right\rangle^2 \right) | \mathcal{F}_{i,k-1} \right]$$

$$\leq \exp\left( -\lambda \alpha^2 \right) \mathbb{E}\left[ \exp\left( \lambda \|G_{i,k+1}\|_2^2 \left\| \mathbb{E}[x_0^{(i)} | x_{t_{N-k+1}}^{(i)}] - \mathbb{E}[x_0^{(i)} | x_{t_{N-k}}^{(i)}] \right\|_2^2 \right) | \mathcal{F}_{i,k-1} \right]$$

Since $G_{i,k+1}$ is measurable with respect to $\mathcal{F}_{i,k-1}$, set $\lambda := \frac{1}{\|G_{i,k+1}\|_2^2 \rho_k^2 d}$ for $\rho_k$ defined in Lemma 11,

$$\rho_k := 8\left(L+1\right) e^{t_{N-k+1}} \sigma_{\gamma_k}$$

Therefore,

$$\mathbb{P}\left( \left| \left\langle G_{i,k+1}, \mathbb{E}[x_0^{(i)} | x_{t_{N-k+1}}^{(i)}] - \mathbb{E}[x_0^{(i)} | x_{t_{N-k}}^{(i)}] \right\rangle \right| \geq \alpha | \mathcal{F}_{i,k-1} \right)$$

$$\leq \exp\left( \frac{-\alpha^2}{\|G_{i,k+1}\|_2^2 \rho_k^2 d} \right) \mathbb{E}\left[ \exp\left( \frac{\left\| \mathbb{E}[x_0^{(i)} | x_{t_{N-k+1}}^{(i)}] - \mathbb{E}[x_0^{(i)} | x_{t_{N-k}}^{(i)}] \right\|_2^2}{\rho_k^2 d} \right) | \mathcal{F}_{i,k-1} \right]$$

Note that Lemma 11 shows that $\mathbb{E}[x_0^{(i)} | x_{t_{N-k+1}}^{(i)}] - \mathbb{E}[x_0^{(i)} | x_{t_{N-k}}^{(i)}]$ is $\rho_k \sqrt{d}$ norm subGaussian

$$\mathbb{E}\left[ \exp\left( \frac{\left\| \mathbb{E}[x_0^{(i)} | x_{t_{N-k+1}}^{(i)}] - \mathbb{E}[x_0^{(i)} | x_{t_{N-k}}^{(i)}] \right\|_2^2}{\rho_k^2 d} \right) \right] \leq 2$$

Therefore, using Markov's inequality, with probablity atleast $1 - \delta$,

$$\mathbb{E}\left[ \exp\left( \frac{\left\| \mathbb{E}[x_0^{(i)} | x_{t_{N-k+1}}^{(i)}] - \mathbb{E}[x_0^{(i)} | x_{t_{N-k}}^{(i)}] \right\|_2^2}{\rho_k^2 d} \right) | \mathcal{F}_{i,k-1} \right]$$

$$\leq \frac{1}{\delta} \mathbb{E}\left[ \mathbb{E}\left[ \exp\left( \frac{\left\| \mathbb{E}[x_0^{(i)} | x_{t_{N-k+1}}^{(i)}] - \mathbb{E}[x_0^{(i)} | x_{t_{N-k}}^{(i)}] \right\|_2^2}{\rho_k^2 d} \right) | \mathcal{F}_{i,k-1} \right] \right] \leq \frac{2}{\delta} \tag{37}$$

Pluggint these equations above, we conclude that with probability at-least $1 - \delta$, for every $\alpha > 0$, we have: $\mathbb{P}\left(\left|\left\langle G_{i,k+1}, \mathbb{E}[x_0^{(i)}|x_{t_{N-k+1}}^{(i)}] - \mathbb{E}[x_0^{(i)}|x_{t_{N-k}}^{(i)}]\right\rangle\right| \geq \alpha|\mathcal{F}_{i,k-1}\right) \leq \frac{2}{\delta}\exp(-\lambda\alpha^2)$, which proves the result for $k \in [N-1]$.

For $k = N$, we similarly use the definition of $\nu_k$ Lemma 12,

$$\nu_k := 4\sigma_{\gamma_k}$$

we have,

$$\mathbb{P}\left(\left|\left\langle \bar{G}_i, z_{t_1}^{(i)} - \mathbb{E}\left[z_{t_1}^{(i)}|x_{t_1}^{(i)}\right]\right\rangle\right| \geq \alpha|\mathcal{F}_{i,k-1}\right) \leq \exp\left(\frac{-\alpha^2}{\left\|\bar{G}_i\right\|_2^2 \nu_k^2 d}\right)\mathbb{E}\left[\exp\left(\frac{\left\|z_{t_1}^{(i)} - \mathbb{E}\left[z_{t_1}^{(i)}|x_{t_1}^{(i)}\right]\right\|_2^2}{\nu_k^2 d}\right)\bigg|\mathcal{F}_{i,k-1}\right]$$

The conclusion follows by a similar argument as (37). $\qquad\square$

Based on the bounds established in Lemma 29 and Lemma 30, we establish the following results.

**Lemma 31.** *For $j \in [N]$, Let $t_j := \Delta j$ and $\gamma_j = \Delta$. Then, for some universal constant $C > 0$ the following equations hold:*

$$\sum_{i\in[m],k\in[N]} \mathbb{E}[R_{i,k}^2|\mathcal{F}_{i,k-1}] \leq C\Delta^3(L\Delta + 1 + L^2)\left(\tfrac{N}{1-e^{-2\Delta}} + \tfrac{1}{(1-e^{-2\Delta})^2}\right)\sum_{i\in[m]}\sum_{j=1}^{N}\left\|\zeta(t_j, x_{t_j}^{(i)})\right\|^2$$

*and*

$$\max\left(\sup_{i\in[m]} \beta_{i,N}\sqrt{W_{i,N}}\|\bar{G}_i\|, \sup_{\substack{i\in[m]\\k\in[N-1]}} \beta_{i,k}\sqrt{W_{i,k}}\|\bar{G}_{i,k+1}\|\right) \leq C(L+1)\sqrt{\Delta}\log(\tfrac{1}{\Delta})\sqrt{d\sup_{i,k} W_{i,k}}\sup_{i,k}\|\zeta(t_k, x_{t_k}^{(i)})\|$$

*Proof.* Define $g_0^2 := (L\Delta^2 + \Delta + L^2\Delta)$. Applying Lemma 29, we conclude:

$$\sum_{i\in[m],k\in[N]} \mathbb{E}[R_{i,k}^2|\mathcal{F}_{i,k-1}] \lesssim \sum_{i\in[m]} (L\Delta^2 + \Delta)\|\bar{G}_i\|^2 + \sum_{i\in[m],k\in[N-1]} (L\Delta^2 + \Delta + L^2\Delta)e^{2t_{N-k+1}}\|G_{i,k+1}\|^2$$

$$\lesssim g_0^2\sum_{i\in[m]}\sum_{k=1}^{N}\left\|\sum_{j=k}^{N}\frac{\gamma_j e^{-(t_j-t_k)}\zeta(t_j, x_{t_j}^{(i)})}{\sigma_{t_j}^2}\right\|^2 = \Delta^2 g_0^2\sum_{i\in[m]}\sum_{k=1}^{N}\left\|\sum_{j=k}^{N}\frac{e^{-(t_j-t_k)}\zeta(t_j, x_{t_j}^{(i)})}{\sigma_{t_j}^2}\right\|^2$$

$$= \Delta^2 g_0^2\sum_{i\in[m]}\sum_{k=1}^{N}\left\|\sum_{j=k}^{N}\frac{e^{-(t_j-t_k)}\zeta(t_j, x_{t_j}^{(i)})}{\sigma_{t_j}^2}\right\|^2$$

$$\leq \Delta^2 g_0^2\sum_{i\in[m]}\sum_{k=1}^{N}\left(\sum_{j=k}^{N}\frac{e^{-2(t_j-t_k)}}{\sigma_{t_j}^4}\right)\left(\sum_{j=k}^{N}\|\zeta(t_j, x_{t_j}^{(i)})\|^2\right) \text{, using Cauchy-Schwarz inequality}$$

$$= \Delta^2 g_0^2\sum_{i\in[m]}\sum_{k=1}^{N}\left(\sum_{j=k}^{N}\frac{e^{-2\Delta(j-k)}}{(1-e^{-2j\Delta})^2}\right)\left(\sum_{j=k}^{N}\|\zeta(t_j, x_{t_j}^{(i)})\|^2\right)$$

$$\leq \Delta^2 g_0^2\sum_{i\in[m]}\sum_{k=1}^{N}\left(\sum_{j=k}^{N}\frac{e^{-2\Delta(j-k)}}{(1-e^{-2j\Delta})^2}\right)\left(\sum_{j=1}^{N}\|\zeta(t_j, x_{t_j}^{(i)})\|^2\right)$$

$$\leq \Delta^2 g_0^2\left(\tfrac{N}{1-e^{-2\Delta}} + \tfrac{1}{(1-e^{-2\Delta})^2}\right)\sum_{i\in[m]}\sum_{j=1}^{N}\|\zeta(t_j, x_{t_j}^{(i)})\|^2 \text{ using Lemma 13} \tag{38}$$

Recall $\beta_{i,k}, W_{i,k}$ as defined in Lemma 30. Applying these results along with the union bound we conclude with probability $1 - \delta$, the following holds every $i, k$ simultaneously:

$$\max\left(\sup_{i\in[m]}\beta_{i,N}\sqrt{W_{i,N}}\|\bar{G}_i\|,\ \sup_{\substack{i\in[m]\\k\in[N-1]}}\beta_{i,k}\sqrt{W_{i,k}}\|\bar{G}_{i,k+1}\|\right)$$

$$\le C\sqrt{\Delta}(L+1)\sqrt{d\sup_{i,k}W_{i,k}}\max\left(\sup_{i,k}e^{t_{N-k+1}}\|G_{i,k}\|,\sup_{i}\|\bar{G}_i\|\right)$$

$$\le C\sqrt{\Delta}(L+1)\sqrt{d\sup_{i,k}W_{i,k}}\left(\sum_{j=1}^{N}\frac{e^{-(t_j-t_1)}}{\sigma_{t_j}^2}\right)\sup_{i,k}\gamma_k\|\zeta(t_k,x_{t_k}^{(i)})\|,\text{ using Holder's inequality}$$

$$= C\Delta^{3/2}(L+1)\sqrt{d\sup_{i,k}W_{i,k}}\left(\sum_{j=1}^{N}\frac{e^{-\Delta(j-1)}}{1-e^{-2j\Delta}}\right)\sup_{i,k}\|\zeta(t_k,x_{t_k}^{(i)})\|$$

$$\le C(L+1)\sqrt{\Delta}\log(\tfrac{1}{\Delta})\sqrt{d\sup_{i,k}W_{i,k}}\sup_{i,k}\|\zeta(t_k,x_{t_k}^{(i)})\|,\text{ using Lemma 13}$$

$\square$

We will specialize the setting in Lemma 17 with $M_n$ being given by $H$, the filtration being $\mathcal{F}_{ik}$ and the martingale decomposition given in Lemma 20. Similarly, $\beta_i$ corresponds to $(\beta_{i,k})_{i,k}$, $K_i$ corresponds to $(W_{i,k})_{i,k}$ given in Lemma 30. $\nu_{i,k}^2$ corresponds to the upper bound on $\mathbb{E}[R_{i,k}^2|\mathcal{F}_{i,k}]$ in Lemma 29. Therefore, $J_i$ corresponds to $\max(1,\frac{C}{W_{i,k}}\log(\frac{\beta_{i,k}^2 W_{i,k}}{\nu_{i,k}^2}))$ satisfies $J_i \le C\log(2d)$ for some constant $C$. In this case, the quantity $\sum_{i=1}^{n}\nu_i^2\|G_i\|^2$ as given in Lemma 17 corresponds to $\sum_{i,k}\mathbb{E}[R_{i,k}^2|\mathcal{F}_{i,k-1}]$ and it can be bound using Lemma 31:

$$\sum_{i\in[m],k\in[N]}\mathbb{E}[R_{i,k}^2|\mathcal{F}_{i,k-1}]\le C\Delta^3(L\Delta+1+L^2)\left(\tfrac{N}{1-e^{-2\Delta}}+\tfrac{1}{(1-e^{-2\Delta})^2}\right)\sum_{i\in[m]}\sum_{j=1}^{N}\|\zeta(t_j,x_{t_j}^{(i)})\|^2$$

Similarly, we adapt $\lambda_{\min},\lambda^*$ be the random variables defined in Lemma 17 to our case for some arbitrary $B\in\mathbb{N},\alpha>1$. This lemma demonstrates the concentration of the quantity $H$ conditioned on the event $\mathcal{B}:=\{\lambda_{\min},\lambda^*\in[e^{-B},e^B]\}$. It remains to deal with the following cases:

1. $\max(\lambda_{\min},\lambda^*)>e^B$
2. $\min(\lambda_{\min},\lambda^*)<e^{-B}$

First, consider the case $\max(\lambda_{\min},\lambda^*)>e^B$.

**Lemma 32.** *Assume $\gamma_t=\Delta$, $\Delta<c_0$ for some universal constant $c_0$. Then $\max(\lambda_{\min},\lambda^*)>e^B$ implies*

$$\sum_{i\in[m],t\in\mathcal{T}}\|\zeta(t,x_t^{(i)})\|^2\le\frac{CNm\alpha}{\Delta}e^{-2B}$$

*Proof.* Using the fact that $\alpha>1$, we note that

$$\max(\lambda_{\min},\lambda^*)>e^B$$
$$\implies\max\left(\sup_{\substack{i\in[m]\\k\in[N-1]}}\sqrt{\Delta}e^{t_{N-k+1}}\|G_{i,k+1}\|,\ \sup_{i\in[m]}\sqrt{\Delta}\|\bar{G}_i\|\right)\le C\sqrt{\alpha}e^{-B}\text{ for some universal constant }C$$

$$(39)$$

By defining $G_{i,0}=0$ and , we note that $\sigma_{t_{N-k+1}}^2 e^{t_{N-k+1}}(G_{i,k+1}-G_{i,k})=\zeta(t_{N-k+1},x_{t_{N-k+1}}^{(i)})$ for $k<N$ and $\sigma_{t_1}^2(\bar{G}_i-e^{t_1}G_{i,N-1})=\zeta(t_1,x_{t_1})$. Using the fact that $\sigma_{t_k}^2\le1$ for some universal constant $c_0$, we conclude that

$$\max(\lambda_{\min}, \lambda^*) > e^B$$

$$\implies \sup_{i \in [m], t \in \mathcal{T}} \|\zeta(t, x_t^{(i)})\| \le C\sqrt{\frac{\alpha}{\Delta}} e^{-B}$$

$$\implies \sum_{i \in [m], t \in \mathcal{T}} \|\zeta(t, x_t^{(i)})\|^2 \le \frac{CNm\alpha}{\Delta} e^{-2B} \tag{40}$$

$\square$

We now consider the event $\min(\lambda^*, \lambda_{\min}) < e^{-B}$.

**Lemma 33.** *Assume* $\gamma_t = \Delta$, $t_j = j\Delta$, $\Delta < c_0$ *for some universal constant* $c_0$, $\alpha > 1$. $\min(\lambda^*, \lambda_{\min}) < e^{-B}$ *implies:*

$$\sum_{i \in [m], t \in \mathcal{T}} \|\zeta(t, x_t^{(i)})\|^2 \ge e^{2B} \frac{\Delta}{mdN^2 \log^2(2d)(L+1)^2 \sup_{i,k} W_{i,k}}$$

*Proof.* It is easy to show that $\min(\lambda^*, \lambda_{\min}) < e^{-B}$ implies:

$$\max(\sup_{i,k} e^{2t_{N-k+1}} \|G_{i,k+1}\|, \sup_i \|\bar{G}_i\|) \ge C \frac{e^B}{\log(2d)(L+1)\sqrt{m\Delta d} \sup_{i,k} \sqrt{W_{i,k}}}$$

This implies that there exists $i, k$ such that

$$\frac{\|\zeta(t_k, x_{t_k}^{(i)})\|}{\sigma_{t_k}^2} \ge C \frac{e^B}{N \log(2d)(L+1)\sqrt{m\Delta d} \sup_{i,k} \sqrt{W_{i,k}}}$$

We then conclude the result using the fact that $\sigma_{t_k}^2 \ge c_0 \Delta$ $\square$

**Lemma 34.** *Assume* $N\Delta > 1$, $\Delta < c_0$ *for some universal constant* $c_0$. *Assume* $t_j = \Delta j$ *and* $\gamma_j = \Delta$. *Let* $\alpha > 1$ *and* $B \in \mathbb{N}$. *Let* $\mathbb{L}_2^2(\zeta) := \sum_{i \in [m], t \in \mathcal{T}} \|\zeta(t, x_t^{(i)})\|^2$, $\mathbb{L}_\infty(\zeta) := \sup_{i \in [m], t \in \mathcal{T}} \|\zeta(t, x_t^{(i)})\|$. *Let* $\sigma_{\max} := \log(\frac{1}{\Delta}) \log(2d) \sqrt{d\Delta \sup_{i,k} W_{i,k}}$. *Then with probability* $1 - (2B+1)e^{-\alpha}$, *at least one of the following inequalities hold:*

    *1.*

$$\frac{H}{L+1} \le C\sqrt{\alpha N \Delta^2 \mathbb{L}_2^2(\zeta)} + C\alpha \mathbb{L}_\infty(\zeta)\sigma_{\max}$$

    *2.*

$$\mathbb{L}_2^2(\zeta) \le \frac{CNm\alpha}{\Delta} e^{-2B}$$

    *3.*

$$\mathbb{L}_2^2(\zeta) \ge c_0 \frac{\Delta e^{2B}}{mdN^2 \log^2(2d)(L+1)^2 \sup_{i,k} W_{i,k}}$$

*Proof.* As considered in Lemma 17, define the event $\mathcal{B} := \{\lambda_{\min}, \lambda^* \in [e^{-B}, e^B]\}$. Applying Lemma 17 to our case with the martingale increments as defined in the discussion above, along with bounds for the quantities $\sum_{i=1}^n \nu_i^2 \|G_i\|^2$ and $\sup_i J_i \beta_i \sqrt{K_i} \|G_i\|$ as developed in Lemma 31, we conclude that:

    1. Almost surely

$$\sum_{i=1}^n \nu_i^2 \|G_i\|^2 \le CN\Delta^2(L+1)^2 \sum_{i \in [m], t \in \mathcal{T}} \|\zeta(t, x_t^{(i)})\|^2$$

2. Almost surely

$$\sup_i J_i \beta_i \|G_i\| \sqrt{K_i} \leq C(L+1) \log(2d) \log(\tfrac{1}{\Delta}) \sqrt{d\Delta \sup_{i,k} W_{i,k}} \sup_{i\in[m], t\in\mathcal{T}} \|\zeta(t, x_t^{(i)})\|$$

$$\mathbb{P}\left(\left\{\frac{H}{L+1} > C\sqrt{\alpha N\Delta^2 \mathbb{L}_2^2(\zeta)} + C\alpha\mathbb{L}_\infty(\zeta)\sigma_{\max}\right\} \cap \mathcal{B}\right) \leq (2B+1)e^{-\alpha} \qquad (41)$$

Define the events $\mathcal{B}_1 := \{\max(\lambda_{\min}, \lambda^*) > e^B\}$, $\mathcal{B}_2 := \{\min(\lambda_{\min}, \lambda^*) < e^{-B}\}$, $\mathcal{A} = \left\{\frac{H}{L+1} > C\sqrt{\alpha N\Delta^2 \mathbb{L}_2^2(\zeta)} + C\alpha\mathbb{L}_\infty(\zeta)\sigma_{\max}\right\}$. By Lemma 32, the event $\{\mathbb{L}_2^2(\zeta) > \frac{CNm\alpha}{\Delta}e^{-2B}\} \subseteq \mathcal{B}_1^\complement$. By Lemma 33, the event:

$$\left\{\mathbb{L}_2^2(\zeta) \geq \frac{\Delta e^{2B}}{mdN^2 \log^2(2d)(L+1)^2 \sup_{i,k} W_{i,k}}\right\} \subseteq \mathcal{B}_2^\complement$$

Therefore consider complement of the event of interest in the statement of the lemma:

$$\mathcal{A} \cap \left\{\mathbb{L}_2^2(\zeta) > \frac{CNm\alpha}{\Delta}e^{-2B}\right\} \cap \left\{\mathbb{L}_2^2(\zeta) \geq \frac{\Delta e^{2B}}{mdN^2 \log^2(2d)(L+1)^2 \sup_{i,k} W_{i,k}}\right\}$$
$$\subseteq \mathcal{A} \cap \mathcal{B}_1^\complement \cap \mathcal{B}_2^\complement$$
$$= \left(\mathcal{A} \cap \mathcal{B} \cap \mathcal{B}_1^\complement \cap \mathcal{B}_2^\complement\right) \cup \left(\mathcal{A} \cap \mathcal{B}^\complement \cap \mathcal{B}_1^\complement \cap \mathcal{B}_2^\complement\right)$$

Clearly, $\mathbb{P}(\mathcal{B} \cup \mathcal{B}_1 \cup \mathcal{B}_2) = 1$. This implies $\mathbb{P}(\mathcal{B}^\complement \cap \mathcal{B}_1^\complement \cap \mathcal{B}_2^\complement) = 0$. Therefore, using the above inclusions along with Equation (41) we conclude:

$$\mathbb{P}\left(\mathcal{A} \cap \left\{\mathbb{L}_2^2(\zeta) > \frac{CNm\alpha}{\Delta}e^{-2B}\right\} \cap \mathcal{B}_2^\complement\right) \leq \mathbb{P}(\mathcal{A} \cap \mathcal{B}) \leq (2B+1)e^{-\alpha}$$

$\square$

# D   Convergence of Empirical Risk Minimization

**Lemma 35.** *For $f \in \mathcal{H}$, let $y_t^{(i)} := \frac{-z_t^{(i)}}{\sigma_t^2}$ and*

$$\mathcal{L}(f) := \sum_{i\in[m], j\in[N]} \frac{\gamma_j \left\|f(t_j, x_{t_j}^{(i)}) - s(t_j, x_{t_j}^{(i)})\right\|_2^2}{m},$$

$$H^f := \sum_{i\in[m], j\in[N]} \frac{\gamma_j}{m} \langle f(t_j, x_{t_j}^{(i)}) - s(t_j, x_{t_j}^{(i)}), y_{t_j}^{(i)} - s(t_j, x_{t_j}^{(i)})\rangle.$$

*If $s \in \mathcal{H}$ then for $\hat{f} = \arg\inf_{f\in\mathcal{H}} \hat{\mathcal{L}}(f)$, we have*

$$\mathcal{L}(\hat{f}) \leq H^{\hat{f}}, \qquad (42)$$

*where $\hat{\mathcal{L}}$ is defined in* (4).

*Proof.* Let $y_t^{(i)} := -\frac{z_t^{(i)}}{\sigma_t^2}$. We have, for any $f \in \mathcal{H}$,

$$\hat{\mathcal{L}}(f) = \hat{\mathcal{L}}(s) + \mathcal{L}(f) + \sum_{i\in[m], j\in[N]} \frac{\gamma_j \left\langle f\left(t_j, x_{t_j}^{(i)}\right) - s\left(t_j, x_{t_j}^{(i)}\right), s\left(t_j, x_{t_j}^{(i)}\right) - y_{t_j}^{(i)}\right\rangle}{m} \qquad (43)$$

where $\widehat{\mathcal{L}}(s) := \sum_{i\in[m],j\in[N]} \frac{\gamma_j \left\| s\left(t_j, x_{t_j}^{(i)}\right) - y_{t_j}^{(i)} \right\|_2^2}{m}$. Since $\hat{f}$ is the minimizer, $\widehat{\mathcal{L}}\left(\hat{f}\right) \le \widehat{\mathcal{L}}(s)$. Therefore,

$$\mathcal{L}\left(\hat{f}\right) \le \sum_{i\in[m],j\in[N]} \frac{\gamma_j \left\langle \hat{f}\left(t_j, x_{t_j}^{(i)}\right) - s\left(t_j, x_{t_j}^{(i)}\right), y_{t_j}^{(i)} - s\left(t_j, x_{t_j}^{(i)}\right) \right\rangle}{m}$$

which completes our proof. $\qquad\square$

We will first demonstrate a very crude bound, which will be of use later to derive a finer bound based on Martingale concentration developed in previous sections.

**Lemma 36.** *Fix $\delta \in (0,1)$ and let $y_t := \frac{-z_t}{\sigma_t^2}$, $\forall t \in \mathcal{T}$, $\gamma_t := \Delta < 1$. Furthermore, assume a linear discretization, i.e, $t_j = \Delta j$. For $\mathcal{L}, \widehat{\mathcal{L}}$ as defined in Lemma 35 and $\hat{f} := \text{argmin}_{f\in\mathcal{F}} \widehat{\mathcal{L}}(f)$, we have almost surely:*

$$\mathcal{L}\left(\hat{f}\right) \le \widehat{\mathcal{L}}(s)$$

*we have with probability atleast $1 - \delta$,*

$$\widehat{\mathcal{L}}(s) \le C(N\Delta + \log(\tfrac{1}{\Delta}))d\log(\tfrac{mN}{\delta})$$

*Proof.* Using Lemma 35 and the Cauchy-Schwarz inequality,

$$\mathcal{L}\left(\hat{f}\right) \le \sum_{i\in[m],j\in[N]} \frac{\gamma_j \left\langle \hat{f}\left(t_j, x_{t_j}^{(i)}\right) - s\left(t_j, x_{t_j}^{(i)}\right), y_{t_j}^{(i)} - s\left(t_j, x_{t_j}^{(i)}\right) \right\rangle}{m} \le \sqrt{\mathcal{L}\left(\hat{f}\right)\widehat{\mathcal{L}}(s)}$$

which completes the first part of the proof. Next, we have

$$\widehat{\mathcal{L}}(s) = \sum_{i\in[m],j\in[N]} \frac{\gamma_j \left\| s\left(t_j, x_{t_j}^{(i)}\right) - y_{t_j}^{(i)} \right\|_2^2}{m}$$

Clearly, since $y_t^{(i)}$ is marginally Gaussian , we conclude that it is $\frac{4\sqrt{d}}{\sigma_t}$ norm subGaussian (see Definition 1). Using the fact that $s(t, x_t)$ is the conditional expectation of $y_t^{(i)}$, Lemma F.3. in [14] shows that $s(t, x_t)$ is $4\sqrt{d}/\sigma_t$-norm subGaussian. Therefore applying a union bound over all $\|s(t, x_t^{(i)})\|, \|y_t^{(i)}\| \gtrsim \frac{\sqrt{d\log(\frac{|\mathcal{T}|m}{\delta})}}{\sigma_t}$, with probability at-least $1 - \delta$ the following holds:

$$\sum_{i\in[m]}\sum_{t\in\mathcal{T}} \frac{\left\| s\left(t, x_t^{(i)}\right) - y_t^{(i)} \right\|_2^2}{m} \lesssim \Delta d\log(\tfrac{Nm}{\delta})\sum_{t\in\mathcal{T}} \frac{1}{\sigma_t^2}$$

Now, note the fact that $\sigma_t \ge c_0 \min(1, t)$ for some universal constant $c_0$. Therefore, $\sum_{t\in\mathcal{T}} \frac{1}{\sigma_t^2} \lesssim N + \frac{\log(\frac{1}{\Delta})}{\Delta}$. Plugging this into the equation above, we conclude the result.

$\qquad\square$

**Lemma 37.** *Recall $y_t^{(i)} := \frac{-z_t^{(i)}}{\sigma_t^2}$ for all $t \in \mathcal{T}$. Let for $f \in \mathcal{H}$,*

$$H^f := \sum_{i\in[m],j\in[N]} \frac{\gamma_j \left\langle f\left(t_j, x_{t_j}^{(i)}\right) - s\left(t_j, x_{t_j}^{(i)}\right), y_{t_j}^{(i)} - s\left(t_j, x_{t_j}^{(i)}\right) \right\rangle}{m}$$

*Then, for $\epsilon > 0$,*

$$\mathbb{P}\left(H^{\hat{f}} \ge \epsilon\right) \le \mathbb{P}\left(\bigcup_{f\in\mathcal{F}} \{H^f \ge \epsilon\} \bigcap \left\{\mathcal{L}(f) \le \widehat{\mathcal{L}}(s)\right\}\right)$$

*where $\mathcal{L}, \widehat{\mathcal{L}}, \hat{f}$ are defined in Lemma 35.*

*Proof.* From Lemma 36, we must have $\mathcal{L}(\hat{f}) \leq \hat{\mathcal{L}}(s)$. Therefore:

$$\mathbb{P}\left(H^{\hat{f}} \geq \epsilon\right) \leq \mathbb{P}\left(\bigcup_{f \in \mathcal{H}} \{H^f \geq \epsilon\} \bigcap \{f \text{ is a minimizer of } \hat{\mathcal{L}}\}\right)$$

$$\leq \mathbb{P}\left(\bigcup_{f \in \mathcal{H}} \{H^f \geq \epsilon\} \bigcap \{\mathcal{L}(f) \leq \hat{\mathcal{L}}(s)\}\right)$$

$\square$

**Lemma 38.** *Let $f \in \mathcal{H}$ and suppose Assumption 1 holds. For any fixed $\tau_0 > 0$, with probability $1 - \delta$, the following holds for every $f \in \mathcal{H}$:*

$$\|f(t+\tau_0, x_{t+\tau_0}) - s(t+\tau_0, x_{t+\tau_0})\| \geq e^{\tau_0}\|f(t, x_t) - s(t, x_t)\| - O(e^{2\tau_0}L\sqrt{d\tau_0}) - 2e^{2\tau_0}L\|z_{t,t+\tau_0}\|$$

$$\|f(t, x_t) - s(t, x_t)\| \geq e^{-\tau_0}\|f(t+\tau_0, x_{t+\tau_0}) - s(t+\tau_0, x_{t+\tau_0})\| - O(e^{\tau_0}L\sqrt{d\tau_0}) - 2e^{\tau_0}L\|z_{t,t+\tau_0}\|$$

*Proof.* Let $g(t, x) := f(t, x) - s(t, x)$. Note that $x_{t+\tau_0} = e^{-\tau_0}x_t + z_{t,t+\tau_0}$. By Assumption 1, $g$ is $2L$ Lipschitz in $x$ and with probability $1 - \delta$ over $x_{t+\tau_0}$, and every $f \in \mathcal{H}$:

$$\|g(t+\tau_0, x_{t+\tau_0})\| \geq e^{\tau_0}\|g(t, x_t)\| - \|g(t+\tau_0, x_{t+\tau_0}) - e^{\tau_0}g(t, x_t)\|$$

$$\geq e^{\tau_0}\|g(t, x_t)\| - \|g(t+\tau_0, x_{t+\tau_0}) - e^{\tau_0}g(t, e^{\tau_0}x_{t+\tau_0})\| - 2e^{\tau_0}L\|e^{\tau_0}x_{t+\tau_0} - x_t\|$$

$$\geq e^{\tau_0}\|g(t, x_t)\| - O(e^{2\tau_0}L\sqrt{d\tau_0 \log(\tfrac{2}{\delta})}) - 2e^{2\tau_0}L\|z_{t,t+\tau_0}\| \tag{44}$$

We conclude the second inequality with a similar proof. $\square$

**Lemma 39.** *Under Assumption 1, with probability $1 - \delta$, for a universal constant $C > 0$ the following holds uniformly for every $f \in \mathcal{H}$:*

$$\left(\sup_{\substack{i \in [m] \\ j \in [N]}} \left\|f\left(t_j, x_{t_j}\right) - s\left(t_j, x_{t_j}\right)\right\|_2\right)^2 \leq C\Delta^{\frac{1}{3}}\left(\sum_{\substack{i \in [m] \\ j \in [N]}} \left\|f\left(t_j, x_{t_j}\right) - s\left(t_j, x_{t_j}\right)\right\|_2^2\right) + CL^2 d\Delta^{\frac{2}{3}}\log(\frac{Nm}{\delta})$$

*Proof.* For the sake of clarity, we will denote $g = f - s$. Using Lemma 38, via the union bound for every $t = t_j$, $\tau_0 = |t_j - t_k|$ along with Gaussian concentration for $z^{(i)}_{t,t+\tau_0}$, we conclude that with probability $1 - \delta$ the following holds uniformly for every $f \in \mathcal{F}$, $i \in [m]$ and $j, k \in \mathcal{T}$ with $|j - k|\Delta \leq 1$ for some universal constant $C, c_0 > 0$:

$$\left\|f\left(t_j, x^{(i)}_{t_j}\right) - s\left(t_j, x^{(i)}_{t_j}\right)\right\|_2 \geq c_0 \left\|f\left(t_k, x^{(i)}_{t_k}\right) - s\left(t_k, x^{(i)}_{t_k}\right)\right\|_2 - CL\sqrt{d|j-k|\Delta \log(\frac{Nm}{\delta})}$$

Squaring both sides and using the AM-GM inequality,

$$\left\|f\left(t_j, x^{(i)}_{t_j}\right) - s\left(t_j, x^{(i)}_{t_j}\right)\right\|_2^2 \geq \frac{c_0^2}{2}\left\|f\left(t_k, x^{(i)}_{t_k}\right) - s\left(t_k, x^{(i)}_{t_k}\right)\right\|_2^2 - C^2 L^2 d|j-k|\Delta \log(\frac{Nm}{\delta}) \tag{45}$$

Now, let $(i^*, k^*) \in \arg\sup_{i \in [m], k \in [N]} \|f(t_k, x^{(i)}_{t_k}) - s(t_k, x^{(i)}_{t_k})\|_2$. Now, for any $j$ such that $|(j - k^*)|\Delta \leq 1$, the Equation (45) implies:

$$\sum_{i \in [m], j \in [N]} \left\|f\left(t_j, x^{(i)}_{t_j}\right) - s\left(t_j, x^{(i)}_{t_j}\right)\right\|_2^2 \geq \sum_{j:|j-k^*|\Delta \leq \Delta^{2/3}} \left\|f\left(t_j, x^{(i^*)}_{t_j}\right) - s\left(t_j, x^{(i^*)}_{t_j}\right)\right\|_2^2$$

$$\geq \sum_{j:|j-k^*|\Delta \leq \Delta^{2/3}} \left(\frac{c_0^2}{2}\left\|f\left(t_{k^*}, x^{(i^*)}_{t_{k^*}}\right) - s\left(t_{k^*}, x^{(i^*)}_{t_{k^*}}\right)\right\|_2^2 - C^2 L^2 d|j-k^*|\Delta \log(\frac{Nm}{\delta})\right)$$

This implies the following inequality from which we can conclude the result.

$$\sum_{i\in[m],j\in[N]}\left\|f\left(t_j,x_{t_j}^{(i)}\right)-s\left(t_j,x_{t_j}^{(i)}\right)\right\|_2^2\geq\frac{c_0^2}{2\Delta^{1/3}}\left\|f\left(t_{k^*},x_{t_{k^*}}^{(i^*)}\right)-s\left(t_{k^*},x_{t_{k^*}}^{(i^*)}\right)\right\|_2^2-2C^2L^2d\Delta^{1/3}\log(\tfrac{Nm}{\delta})$$

$\square$

**Theorem 1** (Empirical $L_2$ Bound). *Let Assumption 1 hold. Fix $\delta\in(0,1)$. For all $j\in[N]$, let $t_j:=\Delta j$ and $\gamma_j:=\Delta$. Let $B:=C\log\left((L+1)\,dmN\log\left(\tfrac{1}{\delta}\right)/\Delta\right)$ for an absolute constant $C>0$, and let $\Delta\log^3(\tfrac{1}{\Delta})d^3\log^3(2d)\log^3\left(\tfrac{2Nm}{\delta}\right)\log^3\left(\tfrac{B|\mathcal{H}|}{\delta}\right)\leq1$ and $N\Delta\leq C\log(\tfrac{1}{\Delta})$. Then for*

$$m\gtrsim\frac{(L+1)^2}{\epsilon^2}\log\left(\frac{B|\mathcal{H}|}{\delta}\right)N\Delta$$

*with probability at least $1-\delta$,*

$$\sum_{i\in[m],j\in[N]}\frac{\gamma_j\left\|\hat{f}\left(t_j,x_{t_j}^{(i)}\right)-s\left(t_j,x_{t_j}^{(i)}\right)\right\|_2^2}{m}\lesssim\epsilon^2$$

*Proof.* Consider $\mathcal{L}(f)$ defined in Lemma 35, $H^f$ as defined in Lemma 2. Let $\hat{f}$ be the empirical risk minimizer. Then, by Lemma 35, we have: $\mathcal{L}(\hat{f})\leq H^{\hat{f}}$ almost surely. Then, using Lemma 37, we have: $\mathcal{L}(\hat{f})\leq\hat{\mathcal{L}}(s)$ almost surely.

As per Lemma 36, we pick $\mathsf{UB}=C(N\Delta+\log(\tfrac{1}{\Delta}))d\log(\tfrac{mN}{\delta})$ for some large enough constant $C$ and conclude that

$$\mathbb{P}\left(\mathcal{L}(\hat{f})>\mathsf{UB}\right)\leq\frac{\delta}{4}\tag{46}$$

Let $f\in\mathcal{F}$ be arbitrary. We consider the martingale $H$ developed in Appendix C with $\zeta=\frac{s-f}{m}$. In this case we can identify $H^f=H$. Considering the notation given in Lemma 34, we have: $\mathbb{L}_2^2(\zeta)=\frac{1}{m\Delta}\mathcal{L}(f)$. Let $\alpha=\log(\frac{10|\mathcal{H}|(2B+1)}{\delta})$. By Lemma 34, we conclude $\mathbb{P}(\mathcal{A}_1(f)\cup\mathcal{A}_3(f)\cup\mathcal{A}_3(f))\geq1-(2B+1)e^{-\alpha}$ where:

1.

$$\mathcal{A}_1(f):=\left\{\frac{H^f}{L+1}\leq C\sqrt{\frac{\alpha N\Delta\mathcal{L}(f)}{m}}+C\frac{\alpha\sigma_{\max}}{m}\sup_{i,t\in\mathcal{T}}|f(t,x_t^{(i)})-s(t,x_t^{(i)})|\right\}$$

2.

$$\mathcal{A}_2(f):=\left\{\mathcal{L}(f)\leq CNm^2\alpha e^{-2B}\right\}$$

3.

$$\mathcal{A}_3(f):=\left\{\mathcal{L}(f)\geq c_0\frac{\Delta^2e^{2B}}{dN^2\log^2(2d)(L+1)^2\sup_{i,k}W_{i,k}}\right\}$$

Taking a union bound over all $f\in\mathcal{H}$, we conclude that $\hat{f}$ satisfies:

$$\mathbb{P}(\mathcal{A}_1(\hat{f})\cup\mathcal{A}_2(\hat{f})\cup\mathcal{A}_3(\hat{f}))\geq1-(2B+1)|\mathcal{H}|e^{-\alpha}.$$

Since $\alpha=\log(\frac{10|\mathcal{H}|(2B+1)}{\delta})$, we conclude:

$$\mathbb{P}(\mathcal{A}_1(\hat{f})\cup\mathcal{A}_2(\hat{f})\cup\mathcal{A}_3(\hat{f}))\geq1-\frac{\delta}{4}.\tag{47}$$

By Lemma 30, we conclude that with probability $1 - \frac{\delta}{4}$, $\sup_{i,k} W_{i,k} \leq \log(\frac{8Nm}{\delta})$. Now, consider

$$
\begin{aligned}
\mathbb{P}(\mathcal{A}_3(\hat{f})) &\leq \mathbb{P}(\mathcal{A}_3(\hat{f}) \cap \{\sup_{i,k} W_{i,k} \leq \log(\tfrac{8Nm}{\delta})\}) + \mathbb{P}(\{\sup_{i,k} W_{i,k} > \log(\tfrac{8Nm}{\delta})\}) \\
&\leq \mathbb{P}(\mathcal{A}_3(\hat{f}) \cap \{\sup_{i,k} W_{i,k} \leq \log(\tfrac{8Nm}{\delta})\}) + \frac{\delta}{4} \\
&\leq \mathbb{P}\left(\left\{\mathcal{L}(f) \geq c_0 \frac{\Delta^2 e^{2B}}{dN^2 \log^2(2d)(L+1)^2 \log(\frac{8Nm}{\delta})}\right\}\right) + \frac{\delta}{4} \\
&\leq \mathbb{P}\left(\{\mathcal{L}(f) \geq \mathsf{UB}\}\right) + \frac{\delta}{4}, \quad \text{(by using the definition of } B\text{)} \\
&\leq \frac{\delta}{2}, \quad \text{(by using Equation (46))}
\end{aligned}
\tag{48}
$$

Now, consider the event $\mathcal{A}_2(\hat{f})$. It is clear from our choice of $B$ that following inclusion holds:

$$
\{\mathcal{L}(\hat{f}) \leq \frac{1}{m}\} \subseteq \mathcal{A}_2(\hat{f})
\tag{49}
$$

Now, consider the event $\mathcal{A}_1(\hat{f})$. Define the following events for some large enough constant $C$.

$$
\begin{aligned}
\mathcal{C} := \cap_{f \in \mathcal{F}} \Bigg\{ &\left( \sup_{\substack{i \in [m] \\ j \in [N]}} \left\| f\left(t_j, x_{t_j}\right) - s\left(t_j, x_{t_j}\right) \right\|_2 \right)^2 \\
&\leq C\Delta^{\frac{1}{3}} \left( \sum_{\substack{i \in [m] \\ j \in [N]}} \left\| f\left(t_j, x_{t_j}\right) - s\left(t_j, x_{t_j}\right) \right\|_2^2 \right) + CL^2 d\Delta^{\frac{1}{3}} \log(\tfrac{Nm}{\delta}) \Bigg\}
\end{aligned}
$$

$$
\mathcal{D} := \left\{ \sigma_{\max} \leq C \log(\tfrac{1}{\Delta}) \log(2d) \sqrt{d\Delta \log(\tfrac{Nm}{\delta})} \right\}
$$

Lemma 39, we have $\mathbb{P}(\mathcal{C}) \geq 1 - \frac{\delta}{8}$. By Lemma 30, and union bound we have $\sup_{i,k} W_{i,k} \leq \log(\frac{8Nm}{\delta})$ with probability $1 - \frac{\delta}{8}$. Therefore, $\mathbb{P}(\mathcal{D}) \geq 1 - \frac{\delta}{8}$. Under the event $\mathcal{A}_1(\hat{f}) \cap \mathcal{C} \cap \mathcal{D}$ we have:

1. $\mathcal{L}(\hat{f}) \leq H^{\hat{f}}$ (This holds almost surely by Lemma 35)

2.

$$
H^{\hat{f}} \leq C(L+1)\sqrt{\frac{\alpha N \Delta \mathcal{L}(\hat{f})}{m}} + C(L+1)\frac{\alpha \sigma_{\max}}{m}\left[\Delta^{-1/3}\sqrt{m\mathcal{L}(\hat{f})} + L\sqrt{d}\Delta^{1/6}\sqrt{\log(\tfrac{Nm}{\delta})}\right]
$$

3.

$$
\sigma_{\max} \leq C \log(\tfrac{1}{\Delta}) \log(2d) \sqrt{d\Delta \log(\tfrac{Nm}{\delta})}
$$

Using the choice of $\Delta$ being small enough as stated in the Theorem, as well as our choice of $\alpha$, we conclude that under the event $\mathcal{A}_1(\hat{f}) \cap \mathcal{C} \cap \mathcal{D}$, for some large enough constant $C'$:

$$
\mathcal{L}(\hat{f}) \leq C'(L+1)\sqrt{\frac{\alpha N \Delta \mathcal{L}(\hat{f})}{m}} + C'\frac{(L+1)}{m}
$$

$$
\implies \mathcal{L}(\hat{f}) \leq \frac{(L+1)^2 \log(1/\Delta) \log(\frac{|\mathcal{F}|B}{\delta})}{m}
$$

Therefore, under the events $(\mathcal{A}_1(\hat{f}) \cap \mathcal{D} \cap \mathcal{C}) \cup \mathcal{A}_2(\hat{f})$, the guarantee for $\mathcal{L}(\hat{f})$ stated in the theorem holds. It now remains to show that $\mathbb{P}\left((\mathcal{A}_1(\hat{f}) \cap \mathcal{D} \cap \mathcal{C}) \cup \mathcal{A}_2(\hat{f})\right) \geq 1 - \delta$. We begin with Equation (47):

$$
\begin{aligned}
1 - \frac{\delta}{4} &\leq \mathbb{P}(\mathcal{A}_1(\hat{f}) \cup \mathcal{A}_2(\hat{f}) \cup \mathcal{A}_3(\hat{f})) \\
&\leq \mathbb{P}(\mathcal{A}_1(\hat{f}) \cup \mathcal{A}_2(\hat{f})) + \mathbb{P}(\mathcal{A}_3(\hat{f})) \leq \mathbb{P}(\mathcal{A}_1(\hat{f}) \cup \mathcal{A}_2(\hat{f})) + \frac{\delta}{2}, \quad \text{by applying Equation (48)} \\
&= \mathbb{P}((\mathcal{A}_1(\hat{f}) \cup \mathcal{A}_2(\hat{f})) \cap \mathcal{C} \cap \mathcal{D}) + \mathbb{P}((\mathcal{A}_1(\hat{f}) \cup \mathcal{A}_2(\hat{f})) \cap (\mathcal{C} \cap \mathcal{D})^{\complement}) + \frac{\delta}{2} \\
&\leq \mathbb{P}((\mathcal{A}_1(\hat{f}) \cup \mathcal{A}_2(\hat{f})) \cap \mathcal{C} \cap \mathcal{D}) + \mathbb{P}(\mathcal{C}^{\complement}) + \mathbb{P}(\mathcal{D}^{\complement}) + \frac{\delta}{2} \\
&\leq \mathbb{P}((\mathcal{A}_1(\hat{f}) \cup \mathcal{A}_2(\hat{f})) \cap \mathcal{C} \cap \mathcal{D}) + \frac{3\delta}{4}, \quad \text{by bound on } \mathbb{P}(\mathcal{C}), \mathbb{P}(\mathcal{D}) \text{ given above} \\
&= \mathbb{P}((\mathcal{A}_1(\hat{f}) \cap \mathcal{C} \cap \mathcal{D}) \cup (\mathcal{A}_2(\hat{f}) \cap \mathcal{C} \cap \mathcal{D})) + \frac{3\delta}{4} \\
&\leq \mathbb{P}((\mathcal{A}_1(\hat{f}) \cap \mathcal{C} \cap \mathcal{D}) \cup \mathcal{A}_2(\hat{f})) + \frac{3\delta}{4} \quad\quad\quad\quad\quad\quad\quad\quad\quad\quad (50)
\end{aligned}
$$

This demonstrates the desired result. $\qquad\square$

## E    Generalization error bounds

**Lemma 40.** *Let all $f(t, x) \in \mathcal{H}$, be parameterized as $g(t, x; \theta)$ for $\theta \in \Theta \subseteq \mathbb{R}^D$ and $\theta_*$ be such that $h(t, x_t; \theta_*) = s(t, x_t)$. Suppose $\exists \lambda, \mu \geq 0$ such that $\forall \theta \in \Theta$,*

$$
\mathbb{E}\left[\|g(t, x_t; \theta) - g(t, x_t, \theta_*)\|_2^4\right] \leq \lambda^2 \|\theta - \theta_*\|^4, \text{ and}
$$

$$
\mathbb{E}\left[\|g(t, x_t; \theta) - g(t, x_t, \theta_*)\|_2^2\right] \geq \mu \|\theta - \theta_*\|^2
$$

*Then, all $f \in \mathcal{H}$ satisfy Assumption 2 with $\kappa = \frac{\lambda}{\mu}$.*

*Proof.* The proof follows by squaring the second inequality and comparing with the first inequality.
$\qquad\square$

**Lemma 41.** *For timestep $t \geq 0$, let $x_t$ be defined as in (1). Consider function $f : \mathbb{R} \times \mathbb{R}^d \to \mathbb{R}^d$ such that $\exists \kappa \geq 1$ satisfying,*

$$
\left(\mathbb{E}_{x_t}\left[\|f(t, x_t) - s(t, x_t)\|_2^4\right]\right)^{\frac{1}{4}} \leq \kappa \left(\mathbb{E}_{x_t}\left[\|f(t, x_t) - s(t, x_t)\|_2^2\right]\right)^{\frac{1}{2}}
$$

*Let $\mathcal{X} = \left\{x_t^{(i)}\right\}_{i \in [m]}$ be iid samples. Then, with probability atleast $1 - \exp\left(-\frac{m}{8\kappa^2}\right)$ there exists a set $\mathcal{G} \subseteq [m]$ such that $|G| \geq \frac{m}{8\kappa^2}$ and*

$$
\forall i \in \mathcal{G}, \quad \left\|f\left(t, x_t^{(i)}\right) - s\left(t, x_t^{(i)}\right)\right\|_2^2 \geq \frac{1}{2}\mathbb{E}_{x_t}\left[\|f(t, x_t) - s(t, x_t)\|_2^2\right]
$$

*Proof.* Using the Payley-Zygmund inequality, for any $i \in [m]$, $\forall \theta \in [0, 1]$,

$$
\mathbb{P}\left(\left\|f\left(t, x_t^{(i)}\right) - s\left(t, x_t^{(i)}\right)\right\|_2^2 \geq \theta\mathbb{E}_{x_t}\left[\|f(t, x_t) - s(t, x_t)\|_2^2\right]\right) \geq (1 - \theta)^2 \frac{\mathbb{E}_{x_t}\left[\|f(t, x_t) - s(t, x_t)\|_2^2\right]^2}{\mathbb{E}_{x_t}\left[\|f(t, x_t) - s(t, x_t)\|_2^4\right]}
$$
(51)

Define the iid indicator random variable $\{\chi_i\}_{i \in [m]}$ as,

$$
\chi_i := \mathbb{1}\left(\left\|f\left(t, x_t^{(i)}\right) - s\left(t, x_t^{(i)}\right)\right\|_2^2 \geq \frac{1}{2}\mathbb{E}_{x_t}\left[\|f(t, x_t) - s(t, x_t)\|_2^2\right]\right)
$$

Then, using (51), $\mathbb{P}\left(\chi_i = 1\right) \geq \frac{1}{4\kappa^2}$. Let $\mu := \sum_{i=1}^{m} \mathbb{E}\left[\chi_i\right] \geq \frac{m}{4\kappa^2}$. Using standard chernoff bounds for Bernoulli random variables,

$$\forall \epsilon \in (0, 1) , \ \mathbb{P}\left(\sum_{i=1}^{m} \chi_i \leq (1 - \epsilon) \mu\right) \leq \exp\left(-\frac{\epsilon^2 \mu}{2}\right)$$

The result then follows by setting $\epsilon := \frac{1}{2}$. $\qquad\square$

**Theorem 2** ($L_2$ Error Bound). *Let Assumptions 1 and 2 hold. Fix $\delta \in (0, 1)$. For all $j \in [N]$, let $t_j := \Delta j$ and $\gamma_j := \Delta$. Let $B := C \log\left((L + 1) dmN \log\left(\frac{1}{\delta}\right) / \Delta\right)$ for an absolute constant $C > 0$, and let $\Delta \log^3(\frac{1}{\Delta})d^3 \log^3(2d) \log^3\left(\frac{2Nm}{\delta}\right) \log^3\left(\frac{B|\mathcal{H}|}{\delta}\right) \leq 1$ and $N\Delta \leq C \log(\frac{1}{\Delta})$. If*

$$m \gtrsim \kappa^2 \max\left\{\log\left(\frac{N}{\delta}\right), \frac{(L + 1)^2 N\Delta}{\epsilon^2} \log\left(\frac{B|\mathcal{H}|}{\delta}\right)\right\}$$

*then with probability at least $1 - \delta$,*

$$\sum_{j\in[N]} \gamma_j \mathbb{E}_{x_{t_j}}\left[\left\|\hat{f}\left(t_j, x_{t_j}\right) - s\left(t_j, x_{t_j}\right)\right\|_2^2\right] \lesssim \epsilon^2$$

*Proof.* Using Theorem 1, we have with probability at least $1 - \delta$,

$$\sum_{i\in[m], j\in[N]} \frac{\gamma_{t_j} \left\|\hat{f}\left(t_j, x_{t_j}\right) - s\left(t_j, x_{t_j}\right)\right\|_2^2}{m} \lesssim \frac{(L + 1)^2 \log\left(\frac{B|\mathcal{H}|}{\delta}\right)}{m} \tag{52}$$

Using Lemma 40 and 41, if $m \gtrsim \kappa^2 \log\left(\frac{N}{\delta}\right)$ then, using a union-bound, for all particular timesteps $\{t_j\}_{j\in[N]}$ with probability at least $1 - \delta$,

$$\frac{1}{\kappa^2} \gamma_{t_j} \mathbb{E}_{x_{t_j}}\left[\left\|\hat{f}\left(t_j, x_{t_j}\right) - s\left(t_j, x_{t_j}\right)\right\|_2^2\right] \lesssim \sum_{i\in[m]} \frac{\gamma_{t_j} \left\|\hat{f}\left(t_j, x_{t_j}^{(i)}\right) - s\left(t_j, x_{t_j}^{(i)}\right)\right\|_2^2}{m} \tag{53}$$

Adding over all timesteps $\{t_j\}_{j\in[N]}$,

$$\sum_{j\in[N]} \gamma_{t_j} \mathbb{E}_{x_{t_j}}\left[\left\|\hat{f}\left(t_j, x_{t_j}\right) - s\left(t_j, x_{t_j}\right)\right\|_2^2\right] \lesssim \kappa^2 \sum_{i\in[m], j\in[N]} \frac{\gamma_{t_j} \left\|\hat{f}\left(t_j, x_{t_j}\right) - s\left(t_j, x_{t_j}\right)\right\|_2^2}{m}$$

$$\lesssim \frac{\kappa^2 (L + 1)^2 \log\left(\frac{B|\mathcal{H}|}{\delta}\right)}{m}$$

The result then follows by setting the RHS smaller by $\epsilon^2$. $\qquad\square$

**Theorem 5** (Accelerated Inference). *Under the same assumptions as Theorem 2, partition the timesteps $\{t_j = \Delta j\}_{j\in[N]}$ into $k$ disjoint subsets $S_1, S_2, \ldots, S_k$, where each subset $S_i$ contains timesteps of the form $t_j = \Delta(i + mk)$ for $m \in \mathbb{N}$. Define $\gamma'_j := k\Delta$ for all $j$ in any subset $S_i$. Then, there exists at least one subset $S_i$ such that:*

$$\sum_{j\in S_i} \gamma'_j \mathbb{E}_{x_{t_j}}\left[\left\|\hat{f}(t_j, x_{t_j}) - s(t_j, x_{t_j})\right\|_2^2\right] \lesssim \epsilon^2,$$

*with probability at least $1 - \delta$.*

*Proof.* From Theorem 2, we have with probability $1 - \delta$:

$$\sum_{j\in[N]} \gamma_j \mathbb{E}_{x_{t_j}}\left[\left\|\hat{f}(t_j, x_{t_j}) - s(t_j, x_{t_j})\right\|_2^2\right] \lesssim \epsilon^2,$$

where $\gamma_j = \Delta$. Partition the $N$ timesteps into $k$ disjoint subsets $S_1, \ldots, S_k$ as described. Each subset $S_i$ contributes:

$$\sum_{j \in S_i} \gamma_j \mathbb{E}_{x_{t_j}} \left[ \left\| \hat{f}(t_j, x_{t_j}) - s(t_j, x_{t_j}) \right\|_2^2 \right] = \sum_{j \in S_i} \Delta \mathbb{E}_{x_{t_j}} \left[ \left\| \hat{f}(t_j, x_{t_j}) - s(t_j, x_{t_j}) \right\|_2^2 \right].$$

Summing over all $k$ subsets gives the original total:

$$\sum_{i=1}^{k} \sum_{j \in S_i} \Delta \mathbb{E}_{x_{t_j}} \left[ \left\| \hat{f}(t_j, x_{t_j}) - s(t_j, x_{t_j}) \right\|_2^2 \right] \lesssim \epsilon^2.$$

Now scale each subset's step size by $k$ (i.e., $\gamma_j' = k\Delta$). The contribution of subset $S_i$ becomes:

$$\sum_{j \in S_i} \gamma_j' \mathbb{E}_{x_{t_j}} \left[ \left\| \hat{f}(t_j, x_{t_j}) - s(t_j, x_{t_j}) \right\|_2^2 \right] = k \sum_{j \in S_i} \Delta \mathbb{E}_{x_{t_j}} \left[ \left\| \hat{f}(t_j, x_{t_j}) - s(t_j, x_{t_j}) \right\|_2^2 \right].$$

Summing over all subsets with the scaled $\gamma_j'$, we get:

$$\sum_{i=1}^{k} \sum_{j \in S_i} \gamma_j' \mathbb{E}_{x_{t_j}} \left[ \left\| \hat{f}(t_j, x_{t_j}) - s(t_j, x_{t_j}) \right\|_2^2 \right] = k \sum_{i=1}^{k} \sum_{j \in S_i} \Delta \mathbb{E}_{x_{t_j}} \left[ \left\| \hat{f}(t_j, x_{t_j}) - s(t_j, x_{t_j}) \right\|_2^2 \right] \lesssim k\epsilon^2.$$

We conclude that at least one subset $S_i$ must satisfy:

$$\sum_{j \in S_i} \gamma_j' \mathbb{E}_{x_{t_j}} \left[ \left\| \hat{f}(t_j, x_{t_j}) - s(t_j, x_{t_j}) \right\|_2^2 \right] \lesssim \epsilon^2,$$

since otherwise all $k$ subsets would contribute more than $\epsilon^2$, leading to a total exceeding $k\epsilon^2$, which contradicts the scaled bound $k\epsilon^2$. $\qquad\square$

## F  Dimension Free Experiments

In this section, we describe our experimental setup for the experiments conducted in Figure 1. We implement DSM on samples drawn from $\mathcal{N}(0, \Sigma)$ using an Ornstein–Uhlenbeck schedule with $\bar{\alpha}_j = \exp(-2\theta t_j)$ ($\theta = 1.0$, $T = 5.0$) and a two-layer MLP of hidden size $H = 1000$ that concatenates the noisy sample $x_t \in \mathbb{R}^d$ with a scalar time embedding $t_j/(N-1) \in [0, 1]$ to predict noise $z_{\text{pred}}$. A random covariance $\Sigma = Q\Lambda Q^\top$ is generated from a GOE matrix with eigenvalues $\Lambda_{ii} \sim \text{Uniform}(1, 2)$. We train for $E = 200$ epochs on $m_{\text{train}} = 1000$ samples (batch size 1000, learning rate $\eta = 10^{-3}$) and evaluate on $m_{\text{test}} = 1000$ held-out samples over $N = 100$ timesteps. For each dimension $d \in \{10, 20, 30, 40, 75, 100, 125, 150, 175, 200\}$ (each averaged over $R = 5$ runs) we compute the per-step MSE $E_j = (1/m_{\text{test}}) \sum_{i=1}^{m_{\text{test}}} \| - \Sigma_{t_j}^{-1} x_t^{(i)} - z_{\text{pred}}^{(i)} / \sqrt{1 - \bar{\alpha}_j} \|^2$, average $E_j$ over $j = 2, \ldots, N$ and runs to obtain a time-averaged error, define the scaled error $\widetilde{E}(d) = (\text{mean time-averaged error})/(\#\text{params}(d) \cdot \log\log d)$, and plot $\widetilde{E}(d)$ versus $d$ on a log–log axis alongside a best-fit linear curve. Our experiments were performed on a single Google Colab CPU.

# G  Bootstrapped Score Matching

---

**Algorithm 1** BSM $\left( \left\{ x_0^{(i)} \right\}_{i \in [m]}, T, N, \{\mathcal{H}_i\}_{i \in [N]}, k_0 \right)$

---

**Input:** Dataset $D := \left\{ x_0^{(i)} \right\}_{i \in [m]}$, Initial Sample Size $m$, Number of discretized timesteps $N$
   labelled as $0 < t_0 < t_1 < \cdots < t_N = T$, Sequence of Function classes $\{\mathcal{H}_i\}_{i \in [N]}$, $k_0 \in \mathbb{N}$

**Output:** Estimated Score Functions $\{\hat{s}_{t_k}\}_{k \in [N]}$ to optimize $\mathbb{E}_{x_{t_k}} \left[ \|\hat{s}(t, x_{t_k}) - s(t_k, x_{t_k})\|_2^2 \right]$

1  **for** $k \in [N]$ **do**
2  $\quad$ Let $\forall i \in [m]$, $x_{t_k}^{(i)} = x_0^{(i)} e^{-t_k} + z_{t_k}^{(i)}$
$\quad$ **if** $k \leq k_0$ **then**
3  $\qquad \hat{s}_{t_k} \leftarrow \arg\min_{f \in \mathcal{H}_k} \frac{1}{m} \sum_{i \in [m]} \left\| f(t_k, x_{t_k}^{(i)}) - \frac{-z_{t_k}^{(i)}}{\sigma_{t_k}^2} \right\|_2^2$ $\triangleright$ *Denoising Score Matching (DSM)*
4  $\quad$ **end**
5  $\quad$ **else**
6  $\qquad \gamma_k \leftarrow t_k - t_{k-1}$
$\qquad \alpha_k \leftarrow e^{-\gamma_k} \frac{1 - e^{-2t_{k-1}}}{1 - e^{-2t_k}}$ $\triangleright$ *Bootstrapped Score Matching (BSM)*
$\qquad \tilde{y}_{t_k}^{(i)} \leftarrow (1 - \alpha_k) \frac{-z_{t_k}^{(i)}}{\sigma_{t_k}^2} + \alpha_k \left( \frac{-z_{t_k}^{(i)}}{\sigma_{t_k}^2} + \left( \hat{s}_{t_{k-1}}(x_{t_{k-1}}^{(i)}) - \frac{-z_{t_{k-1}}^{(i)}}{\sigma_{t_{k-1}}^2} \right) \right)$ $\triangleright$ *Bootstrapped Targets*
$\qquad \hat{s}_{t_k} \leftarrow \arg\min_{f \in \mathcal{H}_k} \sum_{i \in [m]} \frac{\left\| f(t_k, x_{t_k}^{(i)}) - \tilde{y}_{t_k}^{(i)} \right\|_2^2}{m}$ $\triangleright$ *Learning with biased targets*
7  $\quad$ **end**
8  **end**

---

**Lemma 42** (Bootstrap Consistency). *For some $\alpha >$, let*

$$\tilde{y}_t := -\frac{z_t}{\sigma_t^2} - \alpha \left( s(t', x_{t'}) - \frac{-z_{t'}}{\sigma_{t'}^2} \right)$$

*Then, $\mathbb{E}[\tilde{y}_t | x_t] = s(t, x_t)$.*

*Proof.* Note that by Tweedie's formula,

$$s(t', x_{t'}) = \mathbb{E}\left[ \frac{-z_{t'}}{\sigma_{t'}^2} \Big| x_{t'} \right]$$

Therefore, using the Markovian property, we have

$$\mathbb{E}\left[ s(t', x_{t'}) - \frac{-z_{t'}}{\sigma_{t'}^2} \Big| x_t \right] = \mathbb{E}\left[ \mathbb{E}\left[ s(t', x_{t'}) - \frac{-z_{t'}}{\sigma_{t'}^2} | x_{t'}, x_t \right] | x_t \right],$$

$$= \mathbb{E}\left[ \mathbb{E}\left[ s(t', x_{t'}) - \frac{-z_{t'}}{\sigma_{t'}^2} | x_{t'} \right] | x_t \right],$$

$$= 0$$

Finally, the result follows using another application of Tweedie's formula which shows that $s(t, x_t) = \mathbb{E}[-z_t / \sigma_t^2 | x_t]$. $\square$

**Lemma 43** (Bootstrap Variance). *For $\Delta := t - t'$ and $\alpha := e^{-\Delta} \frac{\sigma_{t'}^2}{\sigma_t^2}$, let*

$$\tilde{y}_t := -\frac{z_t}{\sigma_t^2} - \alpha \left( s(t', x_{t'}) - \frac{-z_{t'}}{\sigma_{t'}^2} \right)$$

*Then, under Assumption 1,*

$$\left\| \mathbb{E}\left[ (\tilde{y}_t - s(t, x_t))(\tilde{y}_t - s(t, x_t))^\top | x_t \right] \right\|_{\mathrm{op}} = O\left( \frac{(L^2 + 1)\Delta}{\sigma_t^4} \right)$$

*Proof.* Using Tweedie's formula,

$$s_t(x_t) := \mathbb{E}\left[\frac{-z_t}{\sigma_t^2}\Big|x_t\right], \quad s(t', x_{t'}) := \mathbb{E}\left[\frac{-z_{t'}}{\sigma_{t'}^2}\Big|x_{t'}\right]$$

Using the Markov property,

$$\mathbb{E}\left[s(t', x_{t'}) - \frac{-z_{t'}}{\sigma_{t'}^2}\Big|x_t\right] = \mathbb{E}\left[\mathbb{E}\left[s(t', x_{t'}) - \frac{-z_{t'}}{\sigma_{t'}^2}\Big|x_{t'}, x_t\right]\Big|x_t\right] = \mathbb{E}\left[\mathbb{E}\left[s(t', x_{t'}) - \frac{-z_{t'}}{\sigma_{t'}^2}\Big|x_{t'}\right]\Big|x_t\right] = 0$$

Therefore, $\mathbb{E}[h_{t,t'}|x_t] = 0$. Let $v_{t,t'} := s_t(x_t) - \alpha s(t', x_{t'})$ and $r_{t,t'} := \frac{z_t}{\sigma_t^2} - \alpha\frac{z_{t'}}{\sigma_{t'}^2}$.

First consider $r_{t,t'}$. We have using (1), $z_t = e^{-(t-t')}z_{t'} + z_{t,t'}$ where $z_{t,t'} \sim \mathcal{N}(0, \sigma_{t-t'}^2)$. Then,

$$r_{t,t'} = \frac{z_t}{\sigma_t^2} - \alpha\frac{z_{t'}}{\sigma_{t'}^2} = \frac{e^{-\Delta}z_{t'} + z_{t,t'}}{\sigma_t^2} - \alpha\frac{z_{t'}}{\sigma_{t'}^2} = \left(\frac{e^{-\Delta}}{\sigma_t^2} - \frac{\alpha}{\sigma_{t'}^2}\right)z_{t'} + \frac{z_{t,t'}}{\sigma_t^2} \tag{54}$$

Next, for $v_{t,t'}$ again using Tweedie's formula,

$$\begin{aligned}
v_{t,t'} &= \mathbb{E}\left[\frac{-z_t}{\sigma_t^2}\Big|x_t\right] - \alpha s(t', x_{t'}) = \mathbb{E}\left[\frac{-z_t}{\sigma_t^2}\Big|x_t\right] - \alpha s(t', x_{t'}) \\
&= \mathbb{E}\left[\frac{-e^{-\Delta}z_{t'} - z_{t,t'}}{\sigma_t^2}\Big|x_t\right] - \alpha s(t', x_{t'}) = \mathbb{E}\left[\frac{-e^{-\Delta}z_{t'}}{\sigma_t^2}\Big|x_t\right] - \mathbb{E}\left[\frac{z_{t,t'}}{\sigma_t^2}\Big|x_t\right] - \alpha s(t', x_{t'}) \\
&= \mathbb{E}\left[\mathbb{E}\left[\frac{-e^{-\Delta}z_{t'}}{\sigma_t^2}\Big|x_{t'}, x_t\right]\Big|x_t\right] - \mathbb{E}\left[\frac{z_{t,t'}}{\sigma_t^2}\Big|x_t\right] - \rho_{t,t'}s(t', x_{t'}) \\
&= \mathbb{E}\left[\mathbb{E}\left[\frac{-e^{-\Delta}z_{t'}}{\sigma_t^2}\Big|x_{t'}\right]\Big|x_t\right] - \mathbb{E}\left[\frac{z_{t,t'}}{\sigma_t^2}\Big|x_t\right] - \alpha s(t', x_{t'}), \text{ using the Markov property} \\
&= \alpha\mathbb{E}\left[\mathbb{E}\left[\frac{-z_{t'}}{\sigma_{t'}^2}\Big|x_{t'}\right]\Big|x_t\right] - \alpha s(t', x_{t'}) - \mathbb{E}\left[\frac{z_{t,t'}}{\sigma_t^2}\Big|x_t\right] + \left(\frac{\alpha}{\sigma_{t'}^2} - \frac{e^{-\Delta}}{\sigma_t^2}\right)\mathbb{E}[z_{t'}|x_t] \\
&= \alpha\left(\mathbb{E}[s(t', x_{t'})|x_t] - s(t', x_{t'})\right) - \mathbb{E}\left[\frac{z_{t,t'}}{\sigma_t^2}\Big|x_t\right] + \left(\frac{\alpha}{\sigma_{t'}^2} - \frac{e^{-\Delta}}{\sigma_t^2}\right)\mathbb{E}[z_{t'}|x_t] \tag{55}
\end{aligned}$$

Therefore, using (55) and (54),

$$\begin{aligned}
\tilde{y}_t - s(t, x_t) &= v_{t,t'} + r_{t,t'} \\
&= \alpha\left(\mathbb{E}[s(t', x_{t'})|x_t] - s(t', x_{t'})\right) + \frac{1}{\sigma_t^2}\left(z_{t,t'} - \mathbb{E}[z_{t,t'}|x_t]\right) + \left(\frac{\alpha}{\sigma_{t'}^2} - \frac{e^{-\Delta}}{\sigma_t^2}\right)\left(z_{t'} - \mathbb{E}[z_{t'}|x_t]\right) \\
&= \alpha\left(\mathbb{E}[s(t', x_{t'})|x_t] - s(t', x_{t'})\right) + \frac{1}{\sigma_t^2}\left(z_{t,t'} - \mathbb{E}[z_{t,t'}|x_t]\right), \text{ using the value of } p \\
&= \alpha\left(e^{-(t-t')}s(t, x_t) - s(t', x_{t'})\right) + \frac{1}{\sigma_t^2}\left(z_{t,t'} + \sigma_{t-t'}^2 s(t, x_t)\right), \text{ using Theorem 1 from [7]}
\end{aligned}$$

Therefore,

$$\begin{aligned}
&\mathbb{E}\left[(\tilde{y}_t - s(t, x_t))(\tilde{y}_t - s(t, x_t))^\top|x_t\right] \\
&\preceq 2\alpha^2\mathbb{E}\left[\left(e^{-(t-t')}s(t, x_t) - s(t', x_{t'})\right)\left(e^{-(t-t')}s(t, x_t) - s(t', x_{t'})\right)^\top|x_t\right] \\
&\quad + \frac{2}{\sigma_t^4}\mathbb{E}\left[\left(z_{t,t'} + \sigma_{t-t'}^2 s(t, x_t)\right)\left(z_{t,t'} + \sigma_{t-t'}^2 s(t, x_t)\right)^\top|x_t\right] \\
&= 2\alpha^2\mathbb{E}\left[\left(e^{-(t-t')}s(t, x_t) - s(t', x_{t'})\right)\left(e^{-(t-t')}s(t, x_t) - s(t', x_{t'})\right)^\top|x_t\right] \\
&\quad + \frac{2}{\sigma_t^4}(\sigma_{t-t'}^4 h_t(x_t) + \sigma_{t-t'}^2\mathbf{I}_d) \text{ using Lemma 22, where } h_t(x_t) := \nabla^2\log(p_t(x_t))
\end{aligned}$$

which implies,

$$\left\| \mathbb{E}\left[ (\tilde{y}_t - s(t, x_t))(\tilde{y}_t - s(t, x_t))^\top | x_t \right] \right\|_{\text{op}}$$

$$\leq 2\alpha^2 \left\| \mathbb{E}\left[ \left( e^{-(t-t')}s(t, x_t) - s\left(t', x_{t'}\right) \right) \left( e^{-(t-t')}s(t, x_t) - s\left(t', x_{t'}\right) \right)^\top | x_t \right] \right\|_{\text{op}}$$

$$+ \frac{2}{\sigma_t^4} \left\| \sigma_{t-t'}^4 h_t(x_t) + \sigma_{t-t'}^2 \mathbf{I}_d \right\|_{\text{op}}$$

$$= O\left( \frac{L\Delta^2 + \Delta}{\sigma_t^4} + \alpha^2 L^2 \Delta \right), \text{ using Assumption 1 and Corollary 1}$$

$$= O\left( \frac{(L^2 + 1)\Delta}{\sigma_t^4} \right)$$

$\square$

### G.1 Experimental Details

In this section, we provide some preliminary experiments (Figure 3) with the Bootstrapped Score Matching algorithm described in Section 5. The formal pseudocode has been provided in Algorithm 1.

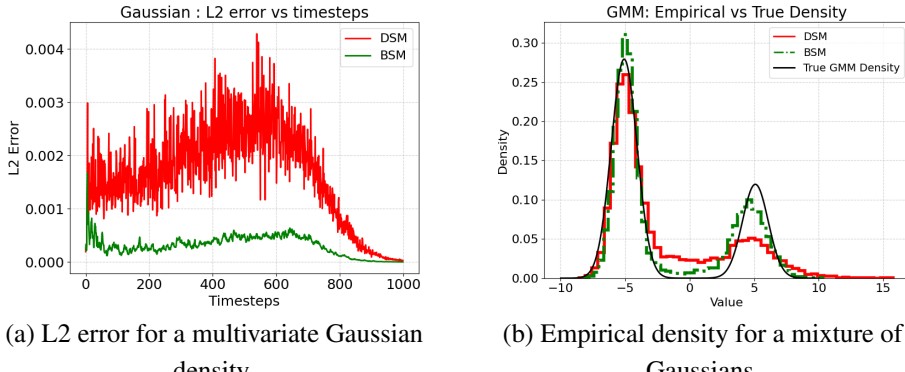

(a) L2 error for a multivariate Gaussian density

(b) Empirical density for a mixture of Gaussians

Figure 3: Experiments with Bootstrapped Score Matching. (a) represents the L2 error at each timestep while performing score estimation for a multivariate Gaussian density. In this case, since the score function is linear, (4) can be solved exactly without a neural network. We note that BSM significantly enhances the quality of the score function. (b) explores multimodal densities, specifically a mixture of Gaussians. Here, we use a 3-layer neural network to represent the score function and plot the empirical density learned by using (2) with different score estimation algorithms. We note that using score bootstrapping significantly enhances the proportional representation of the minor mode, leading to a fair output. We provide details of the experimental setup in the Appendix Section G.

In the first experiment, we study the accuracy of different score estimation methods in the context of learning the score function of a Gaussian distribution under the variance-reduced Bootstrapped Score Matching (BSM) objective. We compare BSM with DSM to evaluate their relative performance in estimating the true score function across different timesteps. Our target distribution is a $d$-dimensional Gaussian distribution with covariance matrix $\Sigma \in \mathbb{R}^{d \times d}$, constructed as $\Sigma = 5MM^T + 5vv^T$ where $M \in \mathbb{R}^{d \times d}$ and $v \in \mathbb{R}^{d \times 1}$ are sampled from a standard normal distribution. We generate $m = 10000$ samples from the target distribution. Note that since the target density is gaussian, the density at all intermediate timesteps, $p_t$, also follows a gaussian distribution. The time evolution follows an non-linear decay model, with $N = 1000$ discrete timesteps sampled as: $t_i = \text{linspace}(0.001, t_{\text{max}}, N)^2$, where $t_{\text{max}} = \sqrt{5}$. The noise covariance scaling factor follows $\sigma_t = \sqrt{1 - e^{-2t}}$. The bootstrap ratio for BSM is adaptively chosen as $1 - (\sigma_t/(\sigma_{t-t'} + \sigma_t))$, where $t'$ represents the previous timestep. The score function is estimated using the standard least-squares regression solution on account of the simple target distribution which implies a linear score function of the form $s(t, x) := A_t x$ for some matrix $A_t$. We run 5 training epochs for the first few timesteps ($t \leq 3$) and 1 epoch thereafter. We plot the squared error of the learned score matrix, $\hat{A}_t$ against the true score matrix, $A_t$ at all timesteps.

In the second experiment, we move away from the Gaussian density, which is unimodal, to a Gaussian Mixture model (GMM), which is multimodal. We fix the dimensionality of the data as $d = 1$ for ease of visualization, and generate a mixture of two gaussians with means $\pm 5$ and mixture weights $0.7$ and $0.3$ respectively. We generate $m = 10000$ samples from the GMM. The time evolution is linear with $N = 1000$ timesteps. We train a 3 layer neural network with hidden layer dimensions of 10 each, separately for DSM and BSM. We train the neural network for 100 epochs, with an initial learning rate of $0.05$, using the *AdamW* optimizer, along with a cosine scheduler to manage the learning rate schedule. The number of warmup steps of the scheduler are chosen to be $10\%$ of the total training steps. When training the BSM network, we start bootstrapping after $k_0 = 250$ timesteps and 90 epochs. The bootstrap ratio is fixed at $0.9$. Once training is completed, we sample 10000 points using the learned score functions to plot and compare the empirical density. Our experiments were performed on a single Google Colab CPU.

