# OpenReview forum: "Dimension-free Score Matching and Time Bootstrapping for Diffusion Models"
_NeurIPS.cc/2025/Conference — NeurIPS 2025 poster_

### Official Review · Reviewer_NwC3 · 2025-07-02

**Clarity:** 3
**Significance:** 3
**Originality:** 2
**Rating:** 4
**Confidence:** 2

**Summary:**

This paper investigates the sample complexity of score matching and establishes the first nearly dimension-free bounds under certain regularity conditions. Specifically, assuming Gaussian noise and a Lipschitz condition on the score function, the authors show that the number of samples required to achieve an $\varepsilon$-accurate score estimate is $O(L^2 \log(B|H|/\delta)/ \varepsilon^2)$. This bound holds for both empirical error and the expected $L_2$-norm error. The paper also introduces a bootstrapped score matching (BSM) method to improve empirical performance.

**Questions:**

1. Could the authors provide concrete application scenarios where the function class H is explicitly available?

2. If the function class H is defined by a neural network class, what is the relationship between $\log(H)$ and the number of network parameters?

3. Even if the function class is parameterized by a one-dimensional contious parameter in $R$, will $
\log(H)$ tend to infinity?

4. Do previous works [3] and [14] also rely on Assumption 2 to establish generalization bounds for score function learning?

5. Definition 1 seems to be used only in the proof. The authors may consider moving it to Section 4.

6. In Theorem 1, the bound is said to hold with high probability. Is the randomness here due to both the Gaussian noise in $x_t$ and the randomness in the samples $x^{(i)}$? Or conditioned on the data $x_t$?

**Ethical Concerns:**

["NO or VERY MINOR ethics concerns only"]

**Final Justification:**

I will keep my positive score for potential acceptance.

**Limitations:**

Yes

**Quality:**

3

**Strengths And Weaknesses:**

**Strengths:**

The primary contribution of this work lies in establishing a sample complexity bound for score matching that is almost independent of the data dimension. This may address a key challenge in high-dimensional generative modeling. The paper also verify the theory with numerical experiments, and demonstrate the performance of the proposed BSM method.

**Weakness:**

1. The analysis assumes optimization over a specific function class, which may not be available in practical settings.

2. The analysis relies on a hypercontractivity assumption, which may not hold in real-world data distributions.

3. The theoretical results are derived only for a fixed step size $t_j = \Delta j$, which limits their generality.

4. There is a minor typo in Remark 1: "at least" should replace "atleast."

---

> ### Author Rebuttal · Authors · 2025-07-30
>
> We thank the reviewer for the great review. The main concern raised by the reviewer is regarding the function class H, and the means to optimize over this function class and how large this can be. We clarify these points below.
>
> ## The function class $H$ and the dependence on complexity
>
> **Motivation for function class in practice**: We follow a standard learning theoretic setup where optimization is considered over a general function class $H$ with suitable properties. The function class itself is characterized in practice by the type of optimization algorithm used. Our assumptions on the function class were motivated by the practical scenarios where both $t$ and $x_t$ are provided as input to neural networks for score learning, enabling smoothness across time and space. We would like to note that our work presents statistical sample complexity bounds for learning the score function, but to optimize the score learning objective, **explicit access to the function class is not needed in practice** since optimization constraints implicitly define a function class.
>
> **$\log(|H|)$ and dependence on parameters**: Given a class of functions $H$, $\log(|H|)$ is the standard dependency for statistical rates in learning theory (see for e.g. Corollary 4.6 [Ref 1], Section 4.4.3 [Ref 2], Theorem 19.5 [Ref 3]). This could be the set of neural networks with fp32 quantization and bounded weights, for instance. In general, a class of functions with M free parameters has $\log(|H|) = \Theta(M)$. We will clarify this aspect in the paper. Prior works on score matching analysis ([3] and [14] in the paper) also incur such factors, where [14] explicitly upper bounds this quantity with the number of parameters in the neural networks times the diameter.  In particular, for the class represented by fully connected neural network
> with ReLU activations with $P$ parameters, each bounded by $R$ and depth $D$, they use $\log(|H|) \approx PD\log(R)$ (see Setting 1.1 [14])
>
> **Improvements over**  $\log(|H|)$: A large body of papers on Deep Learning Theory explore the high dimensional loss landscape and implicit regularization, a phenomenon where the optimization algorithm only effectively optimizes over a small class of weights, making the effective complexity $\log(|H_0|)$ which is much smaller than $\log(|H|)$, and the bounds should replace $\log|H|$ with $\log|H_0|$. However, this smaller class usually contains a local minimum which is almost as good as the global minimum. See [Ref 1] for a short survey. We believe that further exploration on this topic is beyond the score of our present work, since it involves deep unsolved problems in this field.
>
> ## Assumption 2 and comparison to [3] and [14]
>
> In short, assumption 2 is not exactly present in [3] and [14], these have a different setting allowing them to bypass this assumption. The setting can be more stringent than our setting (see Lines 174-185). We expound more on the differences below:
>
> [14] does not consider the $L^2$ error as we do, but a different “high probability” version which does not involve expectations. This can be bounded without using fourth moments. However, this also incurs a polynomial dependence in the high-probability parameters $\delta$ instead of the logarithmic dependence in our case.
>
> [3] Corollary 9 assumes that the function class has bounded outputs almost surely, which can be more stringent than the control of  4th moments. This also considers the score function and its estimate to be dissipative (which roughly gives quadratic growth at infinity along with sub-Gaussianity as shown in Lemma 45), which allows sharper concentration results.
>
> We want to note that Assumption 2 is very mild since we require control only over the $4^{th}$ moments. This holds for most popular distributions, including heavy tailed distributions which have finite fourth moments such as the pareto distribution. This assumption is also common in the robust statistics literature as shown in the literature review.
>
> ## The randomness in Theorem 1
>
> The randomness in theorem 1 arises from both the datapoints as well as the gaussian noise.
>
> We show that this cannot hold uniformly for arbitrary datapoints $(x_0^{(i)})$ via contradiction because of the following LeCam’s 2 point method style argument below.
>
> Assume that the concentration in Theorem 1 holds uniformly for all choices of $(x_0^{(i)})$ over the support of $P$. Consider two probability densities $P,Q$ such that $P \ll Q$ and $Q \ll P$ (i.e, they are mutually absolutely continuous). Let the result in Theorem 1 holds uniformly for every $x_0^{(1)},\dots, x_0^{(m)}$ over the support of P (and support of Q). Define the events $A_P$ and $A_Q$ as follows:
>
> $A_P (x):=\\{ \frac{1}{m}\sum_{i,j}\gamma_j\|\hat{f}(t,x_{t}^{i})-\nabla\log P(t,x_t^{i})\|^2\leq\epsilon^2 \\}$, $A_Q (x):= \\{ \frac{1}{m}\sum_{i,j}\gamma_j\|\hat{f}(t,x_{t}^{i})-\nabla\log Q(t,x_t^{i})\|^2\leq\epsilon^2\\}$
>
> By theorem 1 and our assumption, this would imply that $\mathbb{P}(A_P(x)) \geq 1-\delta$ and $\mathbb{P}(A_Q(x)) \geq 1-\delta$. Therefore, with probability $1-2\delta$, both $A_P(x)$ and $A_Q(x)$ must hold simultaneously which implies that $\frac{1}{m} \sum_{i,j}\gamma_j\|\nabla \log P(t,x_t^{(i)})-\nabla\log Q(t,x_t^{i})\|^2 \lesssim  \epsilon^2$.
>
> This leads to a contradiction when $P$ and $Q$ are different distributions (say Gaussians with same mean but different variance).
>
> _[Ref 1]: Why Do Local Methods Solve Nonconvex Problems? Tengyu Ma arxiv:2103.13462_

---

> > ### Comment · Reviewer_NwC3 · 2025-08-05
> >
> > Thank for the rebuttal, and I will keep my positive score for potential acceptance.

---

### Official Review · Reviewer_HNsM · 2025-07-03

**Clarity:** 3
**Significance:** 3
**Originality:** 3
**Rating:** 5
**Confidence:** 3

**Summary:**

This paper establishes dimension-free sample complexity bounds (modulo dependence due to the hypothesis class) for diffusion models. A single function approximator is used to jointly estimate the score for different noise levels. The proof techniques include marginal-based error decomposition and sharp variance bounds. Additionally, the paper proposes bootstrapped score matching with variance reduction techniques to improve the performance.

**Questions:**

In this paper, a uniform time discretization is considered. When $T$ is large enough, the marginal distribution $p_T$ approaches the Gaussian distribution $N(0,I)$, implying that the score function should have a simpler form. Could the authors clarify how this affects the sample efficiency? And how to optimize the time discretization?

**Ethical Concerns:**

["NO or VERY MINOR ethics concerns only"]

**Final Justification:**

Overall, I believe this paper provides a good contribution to the community. I skimmed through the other reviews and discussion, looks like the major concerns have been resolved. I will keep my current positive rating.

**Limitations:**

Yes

**Quality:**

3

**Strengths And Weaknesses:**

**Strengths**:
This paper achieves a double-exponential reduction in dimension dependence by leveraging a single function approximator to estimate the score across noise levels. The proposed martingale error decomposition is theoretically novel, offering a clear path toward tighter guarantees. Given the widespread adoption of diffusion models in modern generative modeling, addressing the issue of dimension dependence is impactful, especially for understanding and improving their performance on high-dimensional data.

**Weaknesses**:
I am not aware of significant weaknesses in the current version.

---

> ### Author Rebuttal · Authors · 2025-07-30
>
> We thank the reviewer for the great review, showcasing the technical novelty in our work and the importance of our results.
>
> **Time-variation of the Score function**:  We considered uniform discretization for the sake of clarity of exposition and simpler calculations. We believe our techniques readily extend to non-uniform time discretization.
>
> Thank you for your insightful question. We believe the question regarding the best time discretization is meaningful when the statistical learning error and the discretization error are considered together to analyze sample complexity. This requires additional non-trivial results as noted below, which we hope to pursue in future work.
>
> a) We need to formalize the reviewer’s observation that the score functions at later times should be easier to estimate. This may involve rigorously characterizing the evolution of the smoothness of the score function, $L(t)$, from the target distribution to the Gaussian distribution. While this is intuitively clear, the details are not mathematically obvious.
>
> All our proofs remain intact when the drift satisfies the variable condition $ ||f(t,x) - f(t,y)|| \leq L(t) ||x-y|| $  with a measurable $L(t)\ge 0$. Wherever a fixed $L$ formerly appeared, we
>
> (i) replace the quadratic factor $(L+1)^{2}$ in martingale‐variance terms by the time-weighted average $ \bar L_{2}^{\,2}=\frac{1}{N\Delta}\sum_{j=1}^{N}\gamma_{j}\bigl(L(t_{j})+1\bigr)^{2}$,
>
> (iii) bound each sub-Gaussian tail parameter by the worst-case value $L_{\infty}=\sup_{t\in[0,T]}L(t)$, which therefore enters only inside logarithmic terms.
>
> With these substitutions, the sample-complexity statement of Theorem 1 becomes $m \gtrsim \frac{\bar L_{2}^{\,2}}{\varepsilon^{2}}\log\bigl(B|H|/\delta\bigr)$. Consequently, the dimension-free guarantees are robust to \emph{time-varying smoothness}, depending chiefly on the _average_ $\bar L_{2}^{\,2}$ rather than the worst-case $L_{\infty}$. This can be **especially useful if there is a considerable gap between the $\sup_t L(t)$ and $\sqrt{\sum_t L(t)^2/T}$**.
>
> b) Extend Theorem 1 in [Ref 1] to a fine-grained decomposition depending on the properties of the discretization.
>
> _[Ref 1]: Nearly d-Linear Convergence Bounds for Diffusion Models via Stochastic Localization, Benton et al,  arxiv:2308.03686_

---

> > ### Comment · Reviewer_HNsM · 2025-08-08
> >
> > Thank you for the explanation. I will keep my current rating.

---

### Official Review · Reviewer_Q2sL · 2025-07-04

**Clarity:** 2
**Significance:** 2
**Originality:** 2
**Rating:** 4
**Confidence:** 2

**Summary:**

The paper studies the sampling error of the score function estimated by a score-matching objective in the context of a diffusion model. The  authors provide high-probability bounds for the error. The claimed contributions are 1) the bounds are nearly dimension-free, 2) the use of a novel martingale-based decomposition, and 3) a bootstrapped version of the score estimate with lower variance.

**Questions:**

Most of my questions have been asked in the weakness part of the above section.

**Ethical Concerns:**

["NO or VERY MINOR ethics concerns only"]

**Final Justification:**

My main concern about the paper was that it claims to offer dimension-free bounds whereas there are unexplained dimension dependence in the results. During the rebuttal, the authors have clarified to me how the stated conditions were acceptable in view of the literature and some standard assumptions, which make the results "dimension independent" in view of the literature framework. In view of this, I am willing to raise my score to a weak accept and I lower my confidence score as I'm not as familiar with the literature context the authors mention. However, as the authors' responses have indicated, the "dimension-free" claim in the title and in the paper actually has quite a number of caveats that were not draft in the original draft. I think the authors should pay attention to not overclaiming this contribution and should try to include a proper discussion about these caveats in the revision. I also think that this limits the originally claimed novelty, which prevents me from raising the score up to an Accept.

**Limitations:**

Very restrictive limitations were provided in the Conclusion. I think there are technical limitations that are unclear from the current exposition, and I have included them in the Weakness section.

**Quality:**

2

**Strengths And Weaknesses:**

Strengths:
- The bootstrapped score estimator looks interesting and I am slightly surprised by its performance.


Weaknesses:

- I am not convinced by the claim that the provided results are dimension-free:
1) line 128 as well as subsequent theoretical results involve a condition that is very hard to intrepet, and notably has d^3 log^3(2d) dependence. As Remark 2 says, for this to be truly dimension free, this would require the discretisation step to be Delta = O(1/d^3), which is hard to reason about in practice. I'm not sure I understand why randomly sampled times, as argued in Remark 2, would address this theoretical issue.
2) The log |H| dependence together with s in H feels prohibitively strong as to introduce in the conditions. One would expect H to also have dimension dependence (often exponential in d), unless a very informative class of H is chosen.  It would be helpful for the authors to provide a few concrete examples of score functions that satisfy Assumption 1 with concrete examples of H, to convince the readers that these are actually dimension-free results.

- I am also not convinced that Theorem 3 provides a way to obtain faster inference, as I take that k is fixed in Theorem 3. Can the authors clarify whether k is allowed to grow, and whether the bound in Theorem 3 has hidden implicit dependence on k? In particular, I am not convinced about the claim in Remark 2 that Theorem 3 is a way to answer the Delta = O(1/d^3) requirement.

- While I find the bootstrapped score matching section interesting, not enough details are given to illustrate why the bootstrapped version would be better. Both variance computations are upper bounds and not exact. Also, some limited experiments are included in the appendix but not in the main text. To illustrate the validity of this method, and also in view of the context of this paper, I think the authors need to perform experiments with a larger variety of settings, e.g. multimodal distributions beyond d=1 and non-Gaussian data.

- Section 5: Can the authors clarify whether, the additional term in tilde y_t, should be interpreted as some sort of estimation of the noise? If so, some interpretation around this will be nice. There also seems to be a tradeoff in terms of decreasing Delta to improve the bound in Lemma 7 and the need to get good estimation of hat s in an earlier time step -- it would be great if some more interpretations can be given for this tradeoff. At the moment, the discussion mostly focuses on an explanation of how the method works but not really on why it works and what might be potential barriers for the method. I think a better illustration of the latter would enhance the message quite a lot.

---

> ### Author Rebuttal · Authors · 2025-07-30
>
> We thank the reviewer for their detailed questions and address the main technical concerns below.
>
> ## The condition $\Delta = O(1/d^3)$ and dimension independence
>
> We thank the reviewer for raising the subtle yet important point regarding the effect of setting $\Delta = O(1/d^3))$ on the overall dimension-free rates claimed. Below we explain why this still results in dimension free rates. We begin with an explanation of statistical error and optimization error in standard formulations of diffusion models.
>
> **The Statistical Formulation**: The standard loss formulation in both theory and practice has time discretization $\Delta =0$, corresponding to an integral/expectation over time. See Equation (17) in [Ref1] and Equation (7) in [Ref 2].
>
> We consider a fine approximation of the integral with $\Delta = O(1/d^3)$ due to purely technical reasons since a $\log(\log(1/\Delta))$ term appears in the error bound. Note that Delta can be made much smaller than $O(1/d^3)$ due to $\log\log$ dependence.
>
> In fact, due to assumption 1, the difference between the integral (i.e, $\Delta = 0$) and our loss is $O(\sqrt{\Delta})$. Therefore, we can consider the standard integral formulation of the loss as well with a few additional lines of proof.
>
> **The Optimization Question**: The integral or the large sum in the loss function is computationally intractable to optimize directly. Hence stochastic optimization algorithms such as SGD or Adam are used. Here random batches of datapoints $(x_0, x_t, t)$ are drawn to evaluate the stochastic gradients. When the times are sampled randomly, we obtain unbiased estimators for the gradients of this loss, even when $\Delta = 0$ (or when it is very small). **Therefore, practical training of diffusion models can be performed efficiently even when $\Delta = O(1/\Delta^3)$**.
>
> Consider the widely used github repository in this field, corresponding to [Ref 2] (user:yang-song, project:score_sde in github):  In the file “losses . py”, the times during training are sampled uniformly from a continuous distribution when _training.continuous = True_ in config and they are sampled from discrete grid when it is set to _False_.
>
> **Our Contributions**: In learning theory, statistical convergence rates and optimization complexity are usually studied separately. In our work, we study how well the minimizer of this loss function behaves by **obtaining statistical error rates**, and do not consider the optimization question – which tries to understand how fast we can reach this minimizer. There are several theoretical works which show that stochastic optimization algorithms converge to the minimizers efficiently under certain conditions even when empirical/population risk cannot be efficiently evaluated.
>
> ## $\log(|H|)$ dependence and its meaning
>
> This requires an extended and subtle argument to interpret clearly. First, as noted by reviewer 6emV, log(|H|) dependence is standard in learning theory (see for e.g. Corollary 4.6 [Ref 3], Section 4.4.3 [Ref 4]). Prior works on score matching analysis ([3] and [14] in the paper) also incur such factors. For instance [14] explicitly upper bounds this quantity with the number of parameters in the neural network. Therefore, in comparison **we bring down the complexity from $poly(d)\log(|H|)$ to $\log(|H|)$ which is meaningful in high dimensions**, as emphasized in Remark 1.
>
> **Improvement over prior works**: To put our results in context (see line 190), prior work [12] shows that score matching suffers from exponential dependence on $d$ in the worst case. In contrast, we show that when the target distribution admits a suitable class of estimators with mild smoothness assumptions, score estimation becomes sample-efficient. This closes the gap between theoretical results and empirical findings in diffusion models, where global optimization reliably learns accurate score functions for natural images, despite the worst-case hardness of training neural networks.
>
> **Special cases for improving over** $\log(|H|)$: Many deep learning theory papers study the high-dimensional loss landscape and implicit regularization, where optimization effectively operates over a smaller set of weights. This reduces the effective complexity from $\log⁡|H|$ to $\log⁡|H_0|$ where $|H_0| \ll |H_1|$, leading to tighter bounds. Typically, $H_0$ still contains a local minimum nearly as good as the global one (see [Ref 5]).
>
> We believe that further exploration on this topic is beyond the scope of our present work, since it involves deep unsolved problems in this field.
>
> ## Theorem 3 and faster inference
>
> We thank the reviewer for raising this important technical point. We first want to note that $k$ can be any integer smaller than $N$. We will make this clear in the statement of Theorem 3. **Theorem 3 has no implicit dependence on $k$**. The proof is a straightforward consequence of Markov’s inequality, which can be easily verified by the reviewer. In addition to the explanation of statistical rates vs optimization complexity above, we now discuss the role of discretization error to further explain Remark 2 and its relationship to Theorem 3. Consider the sources of error in diffusion model sampling based on the results in [Theorem 1, Ref 6].
>
> As noted above, during training, the loss function is considered to be an integral (with $\Delta = 0$) or a fine discretization as in our case with $\Delta = O(1/d^3)$. For the sake of clarity, we specialize to the case where $\Delta = O(1/d^3)$.  During inference, the neural network is queried at times $\tau_1,\dots,\tau_A$ which can be a subset of the times $\{t_1,t_2,..,t_N\}$ with $A \ll N$.
>
> Referring to [Theorem 1, Ref 6], there are two sources of error:
>
> 1. Statistical error: $\epsilon_{score}^2(\tau_1,\dots,\tau_A) = \sum_{i=1}^{A} (\tau_{i}-\tau_{i-1}) \mathbb{E}[\|\|\hat{f}(x_{\tau_i},\tau_i) - s(x_{\tau_i}, \tau_i)\|\|\^2]$
> 2. Discretization error: which is related to the quantity: $\max_i |\tau_i - \tau_{i-1}|$
>
> Theorem 1 in our work obtains the statistical error as defined above for the sequence of times $t_1,\dots,t_N$ as $\epsilon_{score}^2(t_1,\dots,t_N) \leq \epsilon^2$. Theorem 3 shows that then shows that we can take a much coarser discretization with $\tau_i - \tau_{i-1} = k \Delta$, and $A =\lceil N/k \rceil$ such that $\epsilon_{score}^2(\tau_1,\dots,\tau_A) \leq \epsilon^2$ is also true. Therefore, **using a coarser discretization during inference does not affect the statistical error**.
>
> ## Questions regarding bootstrapped score matching
>
> The reviewer is correct that the extra term in $\hat{y}_t$ estimates the noise to reduce its variance. We want to note that the main contribution of our work are the theoretical results on score estimation with DSM. Bootstrapped score matching was inspired by some of the analysis techniques used in the process, and we believe this gives an interesting variance reduction paradigm. We present some upper bounds and basic empirical results as evidence of its efficacy, and hope to inspire extensive and rigorous future work on this topic. We state this explicitly in lines 330-331.
>
> Line 314 describes the target which has an unbiased but high variance component and a biased but low variance component where the bias is coming from the estimation of the score at the previous timestep. DSM considers $\alpha_k = 0$ to obtain a completely unbiased estimate with high variance and our aim with BSM is to choose $\alpha_k$ for the optimal bias-variance tradeoff and achieve overall lower sample complexity.
>
> We are exploring this as part of ongoing work and will clarify some of these points in the revised manuscript. We included this in our paper because this is a clean algorithmic takeaway from our analysis which we feel will be interesting for the community to build upon. Therefore we request the reviewer to not consider this as a significant weakness of our work.
>
> _[Ref 1]: Denoising Diffusion Probabilistic Models, Ho et al, arxiv: 2006.11239_
>
> _[Ref 2]: Score-Based Generative Modeling Through Stochastic Differential Equations, Sohl-Dickstein et al, arxiv:2011.13456_
>
> _[Ref 3]: Shalev-Shwartz, S. and Ben-David, S., 2014. Understanding machine learning: From theory to algorithms. Cambridge university press._
>
> _[Ref 4]: Bach, F., 2024. Learning theory from first principles. MIT press._
>
> _[Ref 5]: Why Do Local Methods Solve Nonconvex Problems? Tengyu Ma arxiv:2103.13462_
>
> _[Ref 6]: Nearly d-Linear Convergence Bounds for Diffusion Models via Stochastic Localization, Benton et al,  arxiv:2308.03686_

---

> > ### Comment · Reviewer_Q2sL · 2025-08-05
> >
> > I thank the authors for the detailed replies. The responses do clarify my main questions, which are regarding dimension and |H| dependence. I urge the authors to include such a discussion in the main paper, as there seems to be much more nuances than the authors' claim of "dimension-free bounds". I also think that the phrase "dimension-free" should be used more carefully in the paper. I will adjust my score and confidence accordingly.

---

> > > ### Author Response · Authors · 2025-08-05
> > >
> > > We thank you for your consideration. We will include the above necessary discussions in our revision. We will also further clarify our usage of dimension-free.

---

### Official Review · Reviewer_6emv · 2025-07-04

**Clarity:** 1
**Significance:** 2
**Originality:** 3
**Rating:** 4
**Confidence:** 2

**Summary:**

This work obtains sample (and time) complexity bounds for estimating the score. Assuming that the score may be approximated in $L^2$ and that the score is both spatially and temporally Lipschitz with sufficiently nice parameters, they are able to show that the dependence in dimension is only doubly logarithmic. Namely, the best score estimator in a sufficiently regular and expressive class has $\varepsilon^2$ error if the number of samples used is $O(\log \log d)$.

**Questions:**

**Major**

- The step-size condition in Theorem 1 is unreadable. The factor $d^3$ appears to be outside the logarithm, but this is very unclear. Remark 2 appears to clarify that this does not affect the statistical rate.

**Minor**

- In related work, Markov should always be capitalized
- subGaussian should be sub-Gaussian
- Ito should be \\^Ito.
- Equation (3) is strange, the $-$ sign should be outside the fraction.
- L. 114: gaussian -> Gaussian
- L. 115: datapoints -> data points
- Figure 1 caption: atleast -> at least, similarly elsewhere.
- L. 232: F.3. has an extra period.
- L. 250: tweedie -> Tweedie. Also appears elsewhere.
- L. 258: hessian -> Hessian
- L. 265: proof Freedman -> proof of Freedman

**Ethical Concerns:**

["NO or VERY MINOR ethics concerns only"]

**Final Justification:**

My primary concerns were addressed in the author response, although I still have reservations about how some of the details were communicated in the text.

Nonetheless, I think the paper is sufficiently novel to merit acceptance, although I would not be opposed to rejecting it.

**Limitations:**

Yes.

**Paper Formatting Concerns:**

None.

**Quality:**

3

**Strengths And Weaknesses:**

**Strengths**

- The work purports to obtain dimension-free statistical rates for learning a Lipschitz score under relatively standard hypotheses,
- There are significant technical contributions: leveraging the temporal regularity via a martingale argument, which could prove useful in other domains.
- The BSM algorithm is interesting as a method for variance reduction.

**Weakness**

- It seems that a large price ought to be paid in time complexity ($N \sim d^{3}$) in training, compared to the standard analysis. Is this trade-off reasonable? It is not really clear to me how Theorem 3 fits into this picture in terms of the concrete guarantees.
- The paper is very difficult to parse and should be thoroughly checked for typos. I have highlighted some in the main text below, but have not thoroughly checked the appendix.
- Spatial Lipschitzness appears in a major way and also justifies the temporal regularity. However, a more reasonable hypothesis might be time-varying regularity (in both space and time).
- The dependence on $\log \lvert{\mathcal H}\rvert$, while standard, means that it is difficult to achieve these dimension-free rates for practical classes.

---

> ### Author Rebuttal · Authors · 2025-07-30
>
> We thank the reviewer for  acknowledging our technical contributions and providing helpful technical feedback to improve our work. We will thoroughly check for typos and improve the exposition as suggested. We address the main technical concerns below.
>
> ## Training time computational complexity
>
> The main concern raised by the reviewer is that setting $\Delta = O(1/d^3)$ can make training inefficient. This is an important and subtle technical point. We explain why this does not affect training complexity with a short exposition on statistical error vs optimization error in diffusion models.
>
> **The Statistical Formulation**: The standard loss formulation in both theory and practice has time discretization $\Delta =0$, corresponding to an integral/expectation over time. See Equation (17) in [Ref1] and Equation (7) in [Ref 2].
>
> We consider a fine approximation of the integral with $\Delta = O(1/d^3)$ due to purely technical reasons since a $\log(\log(1/\Delta))$ term appears in the error bound. Note that $\Delta$ can be made much smaller than $O(1/d^3)$ due to $\log\log$ dependence.
>
> In fact, due to assumption 1, the difference between the integral (i.e, $\Delta = 0$) and our loss is $O(\sqrt{\Delta})$. Therefore, we can consider the standard integral formulation of the loss as well with a few additional lines of proof.
>
> **The Optimization Question**: The integral or the large sum in the loss function is computationally intractable to optimize directly. Hence stochastic optimization algorithms such as SGD or Adam are used. Here random batches of datapoints $(x_0, x_t, t)$ are drawn to evaluate the stochastic gradients. When the times are sampled randomly, we obtain unbiased estimators for the gradients of this loss, even when $\Delta = 0$ (or when it is very small). **Therefore, practical training of diffusion models can be performed efficiently even when $\Delta = O(1/d^3)$.**
>
> Consider the widely used github repository in this field, corresponding to [Ref 2] (user:yang-song, project:score_sde in github):  In the file “losses . py”, the times during training are sampled uniformly from a continuous distribution when _training.continuous = True_ in config and they are sampled from discrete grid when it is set to _False_.
>
> **Our Contributions**: In learning theory, statistical convergence rates and optimization complexity are usually studied separately. In our work, we study how well the minimizer of this loss function behaves by **obtaining statistical error rates, and do not consider the optimization question** – which tries to understand how fast we can reach this minimizer. There are several theoretical works which show that stochastic optimization algorithms converge to the minimizers efficiently under certain conditions even when empirical/population risk cannot be efficiently evaluated.
>
> ## Theorem 3 and faster inference
>
> We now discuss the role of discretization error to further explain Remark 2 and its relationship to Theorem 3. Consider the sources of error in diffusion model sampling based on the results in [Theorem 1, Ref 3].
>
> As noted above, during training, the loss function is considered to be an integral (with $\Delta = 0$) or a fine discretization as in our case with $\Delta = O(1/d^3)$. For the sake of clarity, we specialize to the case where $\Delta = O(1/d^3)$.  During inference, the neural network is queried at times $\tau_1,\dots,\tau_A$ which can be a subset of the times $\{t_1,t_2,..,t_N\}$ with $A \ll N$.
>
> Referring to [Theorem 1, Ref 3], there are two sources of error:
>
> 1. Statistical error: $\epsilon_{score}^2(\tau_1,\dots,\tau_A) = \sum_{i=1}^{A} (\tau_{i}-\tau_{i-1}) \mathbb{E}[\|\|\hat{f}(x_{\tau_i},\tau_i) - s(x_{\tau_i}, \tau_i)\|\|\^2]$
> 2. Discretization error: which is related to the quantity: $\max_i |\tau_i - \tau_{i-1}|$
>
> Theorem 1 obtains the statistical error as defined above for the sequence of times $t_1,\dots,t_N$ as $\epsilon_{score}^2(t_1,\dots,t_N) \leq \epsilon^2$. Theorem 3 shows shows that we can take a much coarser discretization with $\tau_i - \tau_{i-1} = k \Delta$, and $A =\lceil N/k \rceil$ such that $\epsilon_{score}^2(\tau_1,\dots,\tau_A) \leq \epsilon^2$ is also true. Therefore, **using a coarser discretization during inference does not affect the statistical error**.
>
> ## Time-varying Lipschitz constant
>
> Thank you for the great suggestion on the assumption. This can be especially useful if the score function of the target distribution has a large Lipschitz constant which gets **smoothed out during the forward process**. We refer to our rebuttal to reviewer HNsM, where we point out that this question is more meaningful when explored with the question of understanding the best time discretization, which requires further technical work (For example, it might need to be finer when there is high Lipschitz constant).
>
> We note that our proof techniques allow for time varying Lipschitz constant, but we chose to have a uniform upper bound for the sake of clarity. This assumption is standard in the Markov Chain Monte Carlo literature as well as analyses of iteration complexity of diffusion models (see [Ref 4]). We plan to explore time varying regularity in future work and sketch the primary modifications below.
>
> All proofs remain intact when the drift satisfies the variable condition $ ||f(t,x) - f(t,y)|| \leq L(t) ||x-y|| $  with a measurable $L(t)\ge 0$. Wherever a fixed $L$ formerly appeared, we
>
> (i) replace the quadratic factor $(L+1)^{2}$ in martingale‐variance terms by the time-weighted average $ \bar L_{2}^{\,2}=\frac{1}{N\Delta}\sum_{j=1}^{N}\gamma_{j}\bigl(L(t_{j})+1\bigr)^{2}$,
>
> (iii) bound each sub-Gaussian tail parameter by the worst-case value $L_{\infty}=\sup_{t\in[0,T]}L(t)$, which therefore enters only inside logarithmic terms.
>
> With these substitutions, the sample-complexity statement of Theorem 1 becomes $m \gtrsim \frac{\bar L_{2}^{\,2}}{\varepsilon^{2}}\log\bigl(B|H|/\delta\bigr)$. Consequently, the dimension-free guarantees are robust to \emph{time-varying smoothness}, depending chiefly on the _average_ $\bar L_{2}^{\,2}$ rather than the worst-case $L_{\infty}$. This can be **especially useful if there is a considerable gap between the $\sup_t L(t)$ and $\sqrt{\sum_t L(t)^2/T}$**.
>
> ## $\log(|H|)$ dependence
>
> As noted by the reviewer, log(|H|) dependence is standard in learning theory (see for e.g. Corollary 4.6 [Ref 5], Section 4.4.3 [Ref 6]). Prior works on score matching analysis ([3] and [14] in the paper) also incur such factors. For instance [14] explicitly upper bounds this quantity with the number of parameters in the neural network. Therefore, in comparison **we bring down the complexity from $poly(d)\log(|H|)$ to $\log(|H|)$** which is meaningful in high dimensions, as emphasized in Remark 1.
>
> **Improvement over prior works**: To put our results in context (see line 190), prior work [12] shows that score matching suffers from exponential dependence on $d$ in the worst case. In contrast, we show that when the target distribution admits a suitable class of estimators with mild smoothness assumptions, score estimation becomes sample-efficient. This closes the gap between theoretical results and empirical findings in diffusion models, where global optimization reliably learns accurate score functions for natural images, despite the worst-case hardness of training neural networks.
>
> **Special cases for improving over** $\log(|H|)$: Many deep learning theory papers study the high-dimensional loss landscape and implicit regularization, where optimization effectively operates over a smaller set of weights. This reduces the effective complexity from $\log⁡|H|$ to $\log⁡|H_0|$ where $|H_0| \ll |H_1|$, leading to tighter bounds. Typically, $H_0$ still contains a local minimum nearly as good as the global one (see [Ref 7]).
>
> We believe that further exploration on this topic is beyond the scope of our present work, since it involves deep unsolved problems in this field.
>
> _[Ref 1]: Denoising Diffusion Probabilistic Models, Ho et al, arxiv: 2006.11239_
>
> _[Ref 2]: Score-Based Generative Modeling Through Stochastic Differential Equations, Sohl-Dickstein et al, arxiv:2011.13456_
>
> _[Ref 3]: Nearly d-Linear Convergence Bounds for Diffusion Models via Stochastic Localization, Benton et al,  arxiv:2308.03686_
>
> _[Ref 4]: Chen, S., Chewi, S., Li, J., Li, Y., Salim, A. and Zhang, A.R., 2022. Sampling is as easy as learning the score: theory for diffusion models with minimal data assumptions. ICLR 2023._
>
> _[Ref 5]: Shalev-Shwartz, S. and Ben-David, S., 2014. Understanding machine learning: From theory to algorithms. Cambridge university press._
>
> _[Ref 6]: Bach, F., 2024. Learning theory from first principles. MIT press._
>
> _[Ref 7]: Why Do Local Methods Solve Nonconvex Problems? Tengyu Ma arxiv:2103.13462_

---

> > ### Comment · Reviewer_6emv · 2025-08-06
> > **Response**
> >
> > I thank the authors for their thorough response. I agree that my concerns (computational complexity, dependence on size of the hypothesis class) do not really involve the key story of this paper, which is about the statistical questions.
> >
> > Instead, the main issue I have with this paper is its lack of readability. Particularly, it would be helpful if the discussion around computational complexity was stated more clearly in the paper. Furthermore, efforts should be taken to improve readability of key theorems.
> >
> > As my score was already positive, I do not see any reason to raise it further. Nonetheless I commend the authors for the effort they have put into their response.

---

> > > ### Author Response · Authors · 2025-08-06
> > >
> > > Thank you for your response and kind words about our explanations. We will incorporate the above discussion on computational complexity in our revised manuscript and will incorporate all feedback to enhance the readability of our results.
> > >
> > > We wanted to respectfully request that if you feel positively about our work and our responses were satisfactory towards your queries, to please reconsider your score since a score of 3 counts as a "borderline reject", not a positive score.
> > >
> > > If there is any other question or concerns that you might have, please let us know and we would be happy to clarify further. Thank you again for your time and consideration!

---

### Note · Authors · 2025-08-12

We thank the reviewers for their thoughtful feedback and insightful discussions. We summarize some salient points of our discussion here in our final remarks.

*(1) Time-varying smoothness.*

We appreciate the suggestion to allow variable Lipschitz constants. Our proofs extend directly to $L(t)$ by replacing $(L+1)^2$ with the time-averaged $\bar L_{2}^{\,2}=\frac{1}{N\Delta}\sum_{j=1}^{N}\gamma_j\bigl(L(t_j)+1\bigr)^2$, while only logarithmic factors depend on $L_\infty=\sup_t L(t)$. We will integrate this formulation and its implications into the revision.

*(2) Clarifications on sample complexity.*

Our main contribution is improving the sample complexity from $d \log |H|$ to the standard $\log |H|$ term. During the discussion, we clarified that the $1/d^{3}$ step-size appears only in an auxiliary analysis for discretization and does not enter either the statistical error bound or the training cost. We will make this explicit in the paper together with concrete examples of $H$.

*(3) Computational complexity of optimization.*

We will add a dedicated subsection separating statistical risk from computational complexity of the optimization algorithm: in practice, unbiased stochastic gradients for the expected loss are obtained by sampling times $t$ directly, so there is no need to enumerate an $O(1/\Delta)$ grid. Moreover, our results show that coarser inference schedules preserve the statistical term of the sampling error. Finally, we will incorporate all typographical and readability edits suggested by the reviewers.

---

### Decision · Program_Chairs · 2025-09-17

**Decision:**

Accept (poster)

**Comment:**

This work establishes a dimension independent bound for score-function learning. The key to achieve the improved statistical rate is to consider joint learning of score functions for all time steps with time regularity smoothness assumption on the hypothesis class. To study such learning problem with dependent data along the trajectory, the authors developed a technique based on Martingale decomposition of the error. Overall, the result presented in the paper improves the generalization analysis of score function learning in the current literature. The reviewers unanimously recommend acceptance, and the meta-review agrees with their assessment.